# CAN WE FIND NASH EQUILIBRIA AT A LINEAR RATE IN MARKOV GAMES?

**Zhuoqing Song**
Fudan University
zqsong19@fudan.edu.cn

**Jason D. Lee**
Princeton University
jasonlee@princeton.edu

**Zhuoran Yang**
Yale University
zhuoran.yang@yale.edu

## ABSTRACT

We study decentralized learning in two-player zero-sum discounted Markov games where the goal is to design a policy optimization algorithm for either agent satisfying two properties. First, the player does not need to know the policy of the opponent to update its policy. Second, when both players adopt the algorithm, their joint policy converges to a Nash equilibrium of the game. To this end, we construct a meta algorithm, dubbed as `Homotopy-PO`, which provably finds a Nash equilibrium at a *global linear rate*. In particular, `Homotopy-PO` interweaves two base algorithms `Local-Fast` and `Global-Slow` via homotopy continuation. `Local-Fast` is an algorithm that enjoys local linear convergence while `Global-Slow` is an algorithm that converges globally but at a slower sublinear rate. By switching between these two base algorithms, `Global-Slow` essentially serves as a "guide" which identifies a benign neighborhood where `Local-Fast` enjoys fast convergence. However, since the exact size of such a neighborhood is unknown, we apply a doubling trick to switch between these two base algorithms. The switching scheme is delicately designed so that the aggregated performance of the algorithm is driven by `Local-Fast`. Furthermore, we prove that `Local-Fast` and `Global-Slow` can both be instantiated by variants of optimistic gradient descent/ascent (OGDA) method, which is of independent interest.

## 1    INTRODUCTION

Multi-agent reinforcement learning (MARL) which studies how a group of agents interact with each other and make decisions in a shared environment (Zhang et al., 2021a) has received much attention in recent years due to its wide applications in games (Lanctot et al., 2019; Silver et al., 2017; Vinyals et al., 2019), robust reinforcement learning (Pinto et al., 2017; Tessler et al., 2019; Zhang et al., 2021b), robotics (Shalev-Shwartz et al., 2016; Matignon et al., 2012), among many others. Problems in MARL are frequently formulated as Markov Games (Littman, 1994; Shapley, 1953). In this paper, we focus on one important class of Markov games: two-player zero-sum Markov games. In such a game, the two players compete against each other in an environment where state transition and reward depend on both players' actions.

Our goal is to design efficient policy optimization methods to find Nash equilibria in zero-sum Markov games. This task is usually formulated as a nonconvex-nonconcave minimax optimization problem. There have been works showing that Nash equilibria in matrix games, which are a special kind of zero-sum Markov games with convex-concave structures, can be found at a linear rate (Gilpin et al., 2012; Wei et al., 2020). However, due to the nonconvexity-nonconcavity, theoretical understanding of zero-sum Markov games is sparser. Existing methods have either sublinear rates for finding Nash equilibria, or linear rates for finding regularized Nash equiliria such as quantal response equilibria which are approximations for Nash equilibria (Alacaoglu et al., 2022; Cen et al., 2021; Daskalakis et al., 2020; Pattathil et al., 2022; Perolat et al., 2015; Wei et al., 2021; Yang & Ma, 2022; Zeng et al., 2022; Zhang et al., 2022; Zhao et al., 2022). A natural question is:

*Q1: Can we find Nash equilibria for two-player zero-sum Markov games at a linear rate?*

Furthermore, in Markov games, it is desirable to design decentralized algorithms. That is, when a player updates its policy, it does not need to know the policy of other agents, as such information is usually unavailable especially when the game is competitive in nature. Meanwhile, other desiderata in MARL include *symmetric updates* and *rationality*. Here symmetry means that the algorithm employed by each player is the same/symmetric, and their updates differ only through using the different local information possessed by each player. Rationality means that if other players adopt stationary policy, the algorithm will converge to the best-response policy (Sayin et al., 2021; Wei et al., 2021). In other words, the algorithm finds the optimal policy of the player.

In decentralized learning, each player observes dynamic local information due to the changes in other players' policy, which makes it more challenging to design efficient algorithms (Daskalakis et al., 2020; Hernandez-Leal et al., 2017; Sayin et al., 2021). Symmetric update also poses challenges for the convergence. Condon (1990) shows multiple variants of value iteration with symmetric updates can cycle and fail to find NEs. Gradient descent/ascent (GDA) with symmetric update can cycle even in matrix games (Daskalakis et al., 2018; Mertikopoulos et al., 2018). Thus, an even more challenging question to pose is:

> *Q2: Can we further answer Q1 with a decentralized algorithm that is symmetric and rational?*

In this paper, we give the first affirmative answers to *Q1* and *Q2*. In specific, we propose a meta algorithm `Homotopy-PO` which provably converges to a Nash equilibrium (NE) with two base algorithms `Local-Fast` and `Global-Slow`. `Homotopy-PO` is a homotopy continuation style algorithm that switches between `Local-Fast` and `Global-Slow`, where `Global-Slow` behaves as a "guide" which identifies a benign neighborhood for `Local-Fast` to enjoy linear convergence. A novel switching scheme is designed to achieve global linear convergence without knowing the size of such a neighborhood. Next, we propose the averaging independent optimistic gradient descent/ascent (Averaging OGDA) method and the independent optimistic policy gradient descent/ascent (OGDA) method. Then, we instantiate `Homotopy-PO` by proving that Averaging OGDA and OGDA satisfy the conditions of `Global-Slow` and `Local-Fast`, respectively. This yields the first algorithm which provably finds Nash equilibria in zero-sum Markov games at a global linear rate. In addition, `Homotopy-PO` is decentralized, symmetric, rational and last-iterate convergent.

**Our contribution.** Our contribution is two-fold. First, we propose a meta algorithm `Homotopy-PO` which is shown to converge to Nash equilibria of two-player zero-sum Markov games with global linear convergence, when the two base algorithms satisfy certain benign properties. Moreover, `Homotopy-PO` is a decentralized algorithm and enjoys additional desiderata in MARL including symmetric update, rationality and last-iterate convergence. Second, we instantiate `Homotopy-PO` by designing two base algorithms based on variants of GDA methods, which are proved to satisfy the conditions required by `Homotopy-PO`. In particular, we prove that the example base algorithm OGDA enjoys local linear convergence to Nash equilibria, which might be of independent interest.

## 1.1 RELATED WORK

A more comprehensive literature review is moved to Appendix A due to the space limitation. Of particular relevance are two decentralized algorithms Daskalakis et al. (2020) and Wei et al. (2021).

Daskalakis et al. (2020) consider an independent policy gradient descent/ascent algorithm which is a natural extension of single-agent policy gradient descent to two-player zero-sum Markov games. They utilize the two-sided gradient dominance to prove a sub-linear convergence rate of the gradient-descent-ascent (GDA) method. This is the first non-asymptotic convergence result of GDA for finding Nash equilibria in Markov games. However, their method is asymmetric, where one-player takes much smaller steps than its opponent. And their convergence results are base on average policies with no explicit guarantee for last-iterate convergence. Wei et al. (2021) propose an actor-critic optimistic policy gradient descent/ascent algorithm that is simultaneous decentralized, symmetric, rational and has $O(1/\sqrt{t})$ last-iterate convergence rate to the Nash equilibrium set. They use a critic which averages the approximate value functions from past iterations to tame nonstationarity in approximate Q-functions and get better approximations for policy gradients. A classical averaging stepsize from Jin et al. (2018) is utilized by the critic so that the errors accumulate slowly and last-

iterate convergence is obtained. However, the critic using such choice of stepsizes throughout the game also destroys linear convergence since relatively large errors from past iterations cannot decay fast enough and will harm the accuracy of more recent steps.

## 2 NOTATIONS AND PRELIMINARIES

For integers $n \leq n'$, we denote $[n : n'] = \{n, n+1, \cdots, n'\}$ and $[n] = \{1, \cdots, n\}$. We use $\|\cdot\|$ to denote the Euclidean norm, and $\|\cdot\|_p$ denotes the $\ell_p$-norm. For any vector $\boldsymbol{x} \in \mathbb{R}^d$ and closed convex set $\mathcal{C} \subseteq \mathbb{R}^d$, let $\mathcal{P}_{\mathcal{C}}(\boldsymbol{x})$ denote the unique projection point of $\boldsymbol{x}$ onto $\mathcal{C}$. In addition, the distance between $\boldsymbol{x}$ and $\mathcal{C}$ is denoted by $\mathrm{dist}(\boldsymbol{x}, \mathcal{C}) = \|\boldsymbol{x} - \mathcal{P}_{\mathcal{C}}(\boldsymbol{x})\|$.

**Markov game.** A two-player zero-sum discounted Markov game is denoted by a tuple $\mathcal{MG} = (\mathcal{S}, \mathcal{A}, \mathcal{B}, \mathbb{P}, \boldsymbol{R}, \gamma)$, where $\mathcal{S} = [S]$ is the state space; $\mathcal{A} = [A]$ and $\mathcal{B} = [B]$ are the action spaces of the min-player and the max-player respectively; $\mathbb{P} : \mathcal{S} \times \mathcal{A} \times \mathcal{B} \to \Delta_{\mathcal{S}}$ is the transition kernel, $\boldsymbol{R} = \{\boldsymbol{R}_s\}_{s \in \mathcal{S}} \subseteq [0, 1]^{A \times B}$ is the reward function, and $\gamma$ is the discount factor. Specifically, at state $s$, when the min-player takes action $a$ and the max-player takes action $b$ at state $s$, $\mathbb{P}(s'|s, a, b)$ is probability that the next state becomes $s'$, $\boldsymbol{R}_s(a, b)$ is the reward received by the max-player, and the min-player receives an loss $-\boldsymbol{R}_s(a, b)$. We assume that the rewards are bounded in $[0, 1]$ without loss of generality.

Let $\boldsymbol{x} = \{\boldsymbol{x}_s\}_{s \in \mathcal{S}}$ and $\boldsymbol{y} = \{\boldsymbol{y}_s\}_{s \in \mathcal{S}}$ denote the policies of the min-player and the max-player, where $\boldsymbol{x}_s \in \Delta_{\mathcal{A}}$ and $\boldsymbol{y}_s \in \Delta_{\mathcal{B}}$. The policy spaces of the min-player and the max-player are denoted by $\mathcal{X} = (\Delta_{\mathcal{A}})^S$, $\mathcal{Y} = (\Delta_{\mathcal{B}})^S$. Let $\mathcal{Z} = \mathcal{X} \times \mathcal{Y}$ denote the product policy space. The policy $\boldsymbol{x} \in \mathcal{X}$ ($\boldsymbol{y} \in \mathcal{Y}$) is treated as an $AS$-dimensional ($BS$-dimensional) vector, and the policy pair $\boldsymbol{z} = (\boldsymbol{x}, \boldsymbol{y})$ is treated as an $(A + B)S$-dimensional vector where $\boldsymbol{z}_s = (\boldsymbol{x}_s, \boldsymbol{y}_s)$ represents an $(A + B)$-dimensional vector by concatenating $\boldsymbol{x}_s$ and $\boldsymbol{y}_s$.

The value function under the policy pair $(\boldsymbol{x}, \boldsymbol{y})$ is defined as an $S$-dimensional vector with its entries representing the expected cumulative rewards: $V^{\boldsymbol{x},\boldsymbol{y}}(s) = \mathbb{E}_{\boldsymbol{x},\boldsymbol{y}}[\sum_{t=0}^{+\infty} \gamma^t \boldsymbol{R}_{s^t}(a^t, b^t) | s^0 = s]$. Define $V^{\boldsymbol{x},\dagger}$ ($V^{\dagger,\boldsymbol{y}}$) as the value functions of $\boldsymbol{x}$ ($\boldsymbol{y}$) with its best response, i.e., $V^{\boldsymbol{x},\dagger}(s) = \max_{\boldsymbol{y}' \in \mathcal{Y}} V^{\boldsymbol{x},\boldsymbol{y}'}(s)$, $V^{\dagger,\boldsymbol{y}}(s) = \min_{\boldsymbol{x}' \in \mathcal{X}} V^{\boldsymbol{x}',\boldsymbol{y}}(s)$. For state $s \in \mathcal{S}$, define the Bellman target operator $\boldsymbol{Q}_s : \mathbb{R}^S \to \mathbb{R}^{A \times B}$ such for vector $v \in \mathbb{R}^S$,

$$\boldsymbol{Q}_s[v](a, b) = \boldsymbol{R}_s(a, b) + \gamma \sum_{s' \in \mathcal{S}} \mathbb{P}(s'|s, a, b) \, v(s').$$

The Q-function $\boldsymbol{Q}^{\boldsymbol{x},\boldsymbol{y}} = \{\boldsymbol{Q}_s^{\boldsymbol{x},\boldsymbol{y}}\}_{s \in \mathcal{S}}$ is defined as a collection of $A$-by-$B$ matrices with $\boldsymbol{Q}_s^{\boldsymbol{x},\boldsymbol{y}} = \boldsymbol{Q}_s[V^{\boldsymbol{x},\boldsymbol{y}}]$. The (state) visitation distribution is defined as $\boldsymbol{d}_s^{\boldsymbol{x},\boldsymbol{y}}(s') = \sum_{t=0}^{+\infty} \gamma^t \mathrm{Pr}^{\boldsymbol{x},\boldsymbol{y}}[s^t = s'|s^0 = s]$. For any distribution $\boldsymbol{\rho} \in \Delta_{\mathcal{S}}$, we abbreviate $V^{\boldsymbol{x},\boldsymbol{y}}(\boldsymbol{\rho}) = \sum_{s \in \mathcal{S}} \boldsymbol{\rho}(s) V^{\boldsymbol{x},\boldsymbol{y}}(s)$, $\boldsymbol{d}_{\boldsymbol{\rho}}^{\boldsymbol{x},\boldsymbol{y}}(s) = \sum_{s' \in \mathcal{S}} \boldsymbol{\rho}(s') \boldsymbol{d}_{s'}^{\boldsymbol{x},\boldsymbol{y}}(s)$. From Gilpin et al. (2012), there is a problem-dependent constant $c_+ > 0$ such that for any policy pair $\boldsymbol{z} = (\boldsymbol{x}, \boldsymbol{y}) \in \mathcal{Z}$ and $s \in \mathcal{S}$,

$$\max_{\boldsymbol{y}_s' \in \Delta_{\mathcal{B}}} \boldsymbol{x}_s^\top \boldsymbol{Q}_s^* \boldsymbol{y}_s' - \min_{\boldsymbol{x}_s' \in \Delta_{\mathcal{A}}} {\boldsymbol{x}_s'}^\top \boldsymbol{Q}_s^* \boldsymbol{y}_s \geq c_+ \cdot \mathrm{dist}(\boldsymbol{z}_s, \mathcal{Z}_s^*). \tag{1}$$

**Nash equilibrium.** The minimax game value of state $s$ is defined as $v^*(s) = \min_{\boldsymbol{x} \in \mathcal{X}} \max_{\boldsymbol{y} \in \mathcal{Y}} V^{\boldsymbol{x},\boldsymbol{y}}(s) = \max_{\boldsymbol{y} \in \mathcal{Y}} \min_{\boldsymbol{x} \in \mathcal{X}} V^{\boldsymbol{x},\boldsymbol{y}}(s)$. A policy pair $(\boldsymbol{x}, \boldsymbol{y})$ is called a Nash equilibrium (NE) if and only if: for any $s \in \mathcal{S}$,

$$V^{\boldsymbol{x},\dagger}(s) = V^{\dagger,\boldsymbol{y}}(s) = v^*(s).$$

Define the minimax Q-functions as $\boldsymbol{Q}_s^* = \boldsymbol{Q}_s[v^*]$. Define the sets $\mathcal{X}_s^*$ and $\mathcal{Y}_s^*$ as

$$\mathcal{X}_s^* = \arg\min_{\boldsymbol{x}_s' \in \Delta_{\mathcal{A}}} \max_{\boldsymbol{y}_s' \in \Delta_{\mathcal{B}}} \langle \boldsymbol{x}_s', \boldsymbol{Q}_s^* \boldsymbol{y}_s' \rangle, \quad \mathcal{Y}_s^* = \arg\max_{\boldsymbol{y}_s' \in \Delta_{\mathcal{B}}} \min_{\boldsymbol{x}_s' \in \Delta_{\mathcal{A}}} \langle \boldsymbol{x}_s', \boldsymbol{Q}_s^* \boldsymbol{y}_s' \rangle.$$

Then $\mathcal{X}_s^*$ and $\mathcal{Y}_s^*$ are non-empty, closed and convex. Denote $\mathcal{Z}_s^* = \mathcal{X}_s^* \times \mathcal{Y}_s^*$. Let $\mathcal{X}^* = \prod_{s \in \mathcal{S}} \mathcal{X}_s^*$, $\mathcal{Y}^* = \prod_{s \in \mathcal{S}} \mathcal{Y}_s^*$, $\mathcal{Z}^* = \prod_{s \in \mathcal{S}} \mathcal{Z}_s^*$. A policy pair $(\boldsymbol{x}^*, \boldsymbol{y}^*)$ attains Nash equilibrium if and only if $(\boldsymbol{x}^*, \boldsymbol{y}^*) \in \mathcal{Z}^*$, i.e., $(\boldsymbol{x}_s^*, \boldsymbol{y}_s^*) \in \mathcal{Z}_s^*$ for any $s \in \mathcal{S}$ (Başar & Olsder, 1998; Filar & Vrieze, 2012). We denote the closure of the NE set's neighborhood as $\overline{B}(\mathcal{Z}^*, c) = \{\boldsymbol{z} \in \mathcal{Z} : \mathrm{dist}(\boldsymbol{z}, \mathcal{Z}^*) \leq c\}$.

**Interaction protocol.** In each iteration, each player plays a policy and observes the marginal reward function and the marginal transition kernel, i.e., in iteration $t$, the min-player plays $\boldsymbol{x}^t \in \mathcal{X}$, while the max-player plays $\boldsymbol{y}^t \in \mathcal{Y}$. The min-player receives the marginal reward function $\boldsymbol{r}_x^t : \mathcal{S} \times \mathcal{A} \to [0,1]$ with $\boldsymbol{r}_x^t(s,a) = \sum_{b \in \mathcal{B}} \boldsymbol{y}_s^t(b)\boldsymbol{R}_s(a,b)$ and marginal transition kernel $\mathbb{P}_x^t : \mathcal{S} \times \mathcal{A} \to \Delta_\mathcal{S}$ with $\mathbb{P}_x^t(s'|s,a) = \sum_{b \in \mathcal{B}} \boldsymbol{y}_s^t(b)\mathbb{P}(s'|s,a,b)$, while the max-player receives $\boldsymbol{r}_y^t$ and $\mathbb{P}_y^t$ which are defined analogously. Each player is oblivious to its opponent's policy.

# 3 A HOMOTOPY CONTINUATION ALGORITHM WITH GLOBAL LINEAR CONVERGENCE

We propose a decentralized algorithm with global linear convergence by (1) proposing a meta algorithm which can achieve global linear convergence with two base algorithms, (2) providing examples for the base algorithms. The analysis for the example base algorithms are in Section 4 and Section 5.

## 3.1 A HOMOTOPY CONTINUTATION META ALGORITHM

We present a homotopy continuation meta algorithm. It can achieve global linear convergence by switching between two base algorithms: Global-Slow base algorithm (`Global-Slow`) and Local-Fast base algorithm (`Local-Fast`). `Global-Slow` is globally convergent, but only attains a $\widetilde{O}(\frac{1}{T})$ rate. `Local-Fast` is not necessarily globally convergent but attains a linear convergence rate in a neighborhood of the Nash equilibrium set.

**Global-Slow base algorithm**: by calling `Global-Slow`$([T_1 : T_2], \tilde{\boldsymbol{z}}, \eta')$ during time interval $[T_1 : T_2]$ where $\tilde{\boldsymbol{z}} = (\tilde{\boldsymbol{x}}, \tilde{\boldsymbol{y}})$ is the initial policy pair, the players play policy pair $\boldsymbol{z}^t = (\boldsymbol{x}^t, \boldsymbol{y}^t)$ for each iteration $t \in [T_1 : T_2]$, and compute an average policy pair $\widehat{\boldsymbol{z}}^{[T_1:T_2]} = (\widehat{\boldsymbol{x}}^{[T_1:T_2]}, \widehat{\boldsymbol{y}}^{[T_1:T_2]})$ at the end of iteration $T_2$ such that $\boldsymbol{z}^t, \widehat{\boldsymbol{z}}^{[T_1:T_2]}$ satisfy the following two properties:

• **global convergence**: there is a problem-dependent constant $C' > 0$ such that

$$\text{dist}(\widehat{\boldsymbol{z}}^{[T_1:T_2]}, \mathcal{Z}^*) \le \frac{C' \log(T_2 - T_1 + 1)}{\eta'(T_2 - T_1 + 1)}, \tag{2}$$

This property means the average policy produced by `Global-Slow` converges to the NE set at a sublinear $\widetilde{O}(1/T)$ rate.

• **geometric boundedness**: there exists a problem-dependent constant $D_0 > 0$ (possibly $D_0 > 1$) such that if $\eta' \le 1$, then for any $t \in [T_1 : T_2]$,

$$\text{dist}^2(\boldsymbol{z}^t, \mathcal{Z}^*) \le D_0^{t-T_1} \cdot \text{dist}^2(\tilde{\boldsymbol{z}}, \mathcal{Z}^*), \tag{3}$$

$$\text{dist}^2(\widehat{\boldsymbol{z}}^{[T_1:T_2]}, \mathcal{Z}^*) \le D_0^{T_2-T_1} \cdot \text{dist}^2(\tilde{\boldsymbol{z}}, \mathcal{Z}^*). \tag{4}$$

This property ensures that the iterate $\boldsymbol{z}^t$ at any intermediate time $t$ and the average policy $\widehat{\boldsymbol{z}}^{[T_1:T_2]}$ do not diverge faster than geometrically from the NE set. In `Global-Slow`, $\{\boldsymbol{z}^t\}_{t \in [T_1:T_2]}$ are the policy pairs played during $[T_1 : T_2]$, while $\widehat{\boldsymbol{z}}^{[T_1:T_2]}$ will mainly be used as the initial policy in the next switch to `Local-Fast` in the meta algorithm `Homotopy-PO` (Algorithm 1).

**Local-Fast base algorithm**: by calling `Local-Fast`$([T_1 : T_2], \widehat{\boldsymbol{z}}, \eta)$ during time interval $[T_1 : T_2]$ where $\widehat{\boldsymbol{z}} = (\widehat{\boldsymbol{x}}, \widehat{\boldsymbol{y}})$ is the initial policy pair, the players play policy pair $\boldsymbol{z}^t = (\boldsymbol{x}^t, \boldsymbol{y}^t)$ for each iteration $t \in [T_1 : T_2]$ such that $\boldsymbol{z}^t$ satisfies the local linear convergence property:

• **local linear convergence**: there exist problem-dependent constants $c_0 \in (0,1)$ and $\delta_0, \Gamma_0 > 0$ such that if $\text{dist}^2(\widehat{\boldsymbol{z}}, \mathcal{Z}^*) < \delta_0 \eta^4$, then for any $t \in [T_1 : T_2]$

$$\text{dist}^2(\boldsymbol{z}^t, \mathcal{Z}^*) \le \Gamma_0 \cdot (1 - c_0 \eta^2)^{t-T_1} \text{dist}^2(\widehat{\boldsymbol{z}}, \mathcal{Z}^*). \tag{5}$$

In other words, if initialized a neighborhood of $\mathcal{Z}^*$ with radius $\sqrt{\delta_0 \eta^4}$, `Local-Fast` converges to $\mathcal{Z}^*$ at a linear rate.

With these base algorithms, a naive and impractical approach is to run `Global-Slow` first until $\boldsymbol{z}^t$ reaches $\overline{B}(\mathcal{Z}^*, \sqrt{\delta_0 \eta^4})$, and then, run `Local-Fast` to achieve linear convergence. However, the problem is *we do not know the value of* $\delta_0$. That is, when running the algorithm, since

$\mathcal{Z}^*$ is unknown, it is impossible tell whether the algorithm has reached the benign neighborhood for `Local-Fast` to enjoy the linear rate . Thus, we cannot decide when to switch from `Global-Slow` to `Local-Fast`.

---

**Algorithm 1:** `Homotopy-PO`: a meta-algorithm with global linear convergence

---

**Input:** iterations: $[0:T]$, initial policy pair: $\boldsymbol{z}^0 \in \mathcal{Z}$, stepsizes: $\eta, \eta' > 0$

set $k = 1$, $\widetilde{\mathcal{I}}_{\mathrm{lf}}^0 = -1$, $\boldsymbol{z}^{-1} = \boldsymbol{z}^0$

**while** $\widetilde{\mathcal{I}}_{\mathrm{lf}}^{k-1} < T$ **do**

$\quad$ $\mathcal{I}_{\mathrm{gs}}^k = \widetilde{\mathcal{I}}_{\mathrm{lf}}^{k-1} + 1$, $\widetilde{\mathcal{I}}_{\mathrm{gs}}^k = \min\{\mathcal{I}_{\mathrm{gs}}^k + 2^k - 1, T\}$, $\mathcal{I}_{\mathrm{lf}}^k = \widetilde{\mathcal{I}}_{\mathrm{gs}}^k + 1$, $\widetilde{\mathcal{I}}_{\mathrm{lf}}^k = \min\{\mathcal{I}_{\mathrm{lf}}^k + 4^k - 1, T\}$

$\quad$ during time interval $[\mathcal{I}_{\mathrm{gs}}^k : \widetilde{\mathcal{I}}_{\mathrm{gs}}^k]$, run `Global-Slow`$([\mathcal{I}_{\mathrm{gs}}^k : \widetilde{\mathcal{I}}_{\mathrm{gs}}^k], \boldsymbol{z}^{\widetilde{\mathcal{I}}_{\mathrm{lf}}^{k-1}}, \eta')$ and compute an

$\quad\quad$ average policy $\widehat{\boldsymbol{z}}^{[\mathcal{I}_{\mathrm{gs}}^k : \widetilde{\mathcal{I}}_{\mathrm{gs}}^k]}$

$\quad$ during time interval $[\mathcal{I}_{\mathrm{lf}}^k : \widetilde{\mathcal{I}}_{\mathrm{lf}}^k]$, run `Local-Fast`$([\mathcal{I}_{\mathrm{lf}}^k : \widetilde{\mathcal{I}}_{\mathrm{lf}}^k], \widehat{\boldsymbol{z}}^{[\mathcal{I}_{\mathrm{gs}}^k : \widetilde{\mathcal{I}}_{\mathrm{gs}}^k]}, \eta)$

$\quad$ $k \leftarrow k + 1$

**end**

---

To overcome this problem, we propose a homotopy continuation method `Homotopy-PO` which smartly switches between `Global-Slow` and `Local-Fast`. The pseudocode is in Algorithm 1. In `Homotopy-PO`, we split $[0:T]$ into the segments:

$$[0:T] = [\mathcal{I}_{\mathrm{gs}}^1 : \widetilde{\mathcal{I}}_{\mathrm{gs}}^1] \cup [\mathcal{I}_{\mathrm{lf}}^1 : \widetilde{\mathcal{I}}_{\mathrm{lf}}^1] \cup \cdots \cup [\mathcal{I}_{\mathrm{gs}}^k : \widetilde{\mathcal{I}}_{\mathrm{gs}}^k] \cup [\mathcal{I}_{\mathrm{lf}}^k : \widetilde{\mathcal{I}}_{\mathrm{lf}}^k] \cup \cdots$$

where $[\mathcal{I}_{\mathrm{gs}}^k : \widetilde{\mathcal{I}}_{\mathrm{gs}}^k]$ is the time interval of the $k$-th call to `Global-Slow` and $\left|[\mathcal{I}_{\mathrm{gs}}^k : \widetilde{\mathcal{I}}_{\mathrm{gs}}^k]\right| = 2^k$; $[\mathcal{I}_{\mathrm{lf}}^k : \widetilde{\mathcal{I}}_{\mathrm{lf}}^k]$ is the time interval of the $k$-th call to `Local-Fast` and $|[\mathcal{I}_{\mathrm{lf}}^k : \widetilde{\mathcal{I}}_{\mathrm{lf}}^k]| = 4^k$. The switching scheme of `Homotopy-PO` method can be summarized as below: starting from $k = 1$,

• (Step 1) during time interval $[\mathcal{I}_{\mathrm{gs}}^k : \widetilde{\mathcal{I}}_{\mathrm{gs}}^k]$, run `Global-Slow` for $\left|[\mathcal{I}_{\mathrm{gs}}^k : \widetilde{\mathcal{I}}_{\mathrm{gs}}^k]\right| = 2^k$ iterations with the initial policy $\boldsymbol{z}^{\widetilde{\mathcal{I}}_{\mathrm{lf}}^{k-1}}$ (for $k \geq 1$, it is the last-iterate policy of the last call to `Local-Fast`)

• (Step 2) during time interval $[\mathcal{I}_{\mathrm{lf}}^k : \widetilde{\mathcal{I}}_{\mathrm{lf}}^k]$, run `Local-Fast` for $\left|[\mathcal{I}_{\mathrm{lf}}^k : \widetilde{\mathcal{I}}_{\mathrm{lf}}^k]\right| = 4^k$ iterations with the initial policy $\widehat{\boldsymbol{z}}^{[\mathcal{I}_{\mathrm{gs}}^k : \widetilde{\mathcal{I}}_{\mathrm{gs}}^k]}$ that is the average policy of the last call to `Global-Slow`

• (Step 3) $k \leftarrow k + 1$, goto Step 1.

Now, we elaborate on how `Homotopy-PO` achieves global linear convergence given a Global-Slow base algorithm and a Local-Fast base algorithm. Specifically, there are two hidden phases which are oblivious to the players and only used for analysis. The two phases are split by $k^* = \max\{k_1^*, k_2^*\}$, where $2^{k_1^*} = \widetilde{O}(1/(\sqrt{\delta_0}\eta^2))$ and $2^{k_2^*} = O(\frac{1}{c_0\eta^2} \log(D_0\Gamma_0)) = \widetilde{O}(1/(c_0\eta^2))$. The value of $k^*$ is unknown to the players.

**Hidden Phase I.** In the beginning, `Global-Slow` behaves like a "guide" in the sense that its average policy $\widehat{\boldsymbol{z}}^{[\mathcal{I}_{\mathrm{gs}}^k : \widetilde{\mathcal{I}}_{\mathrm{gs}}^k]}$ is getting closer to the NE set as $k$ goes. For small $k$, $\mathrm{dist}(\boldsymbol{z}^t, \mathcal{Z}^*)$ could possibly increase when running `Local-Fast`. However, since the average policy $\widehat{\boldsymbol{z}}^{[\mathcal{I}_{\mathrm{gs}}^k : \widetilde{\mathcal{I}}_{\mathrm{gs}}^k]}$ is the initial policy of the $k$-th call to `Local-Fast`, by the global convergence as in (2), for $k \geq k_1^*$, $\widehat{\boldsymbol{z}}^{[\mathcal{I}_{\mathrm{gs}}^k : \widetilde{\mathcal{I}}_{\mathrm{gs}}^k]}$ will reach $\overline{B}(\mathcal{Z}^*, \sqrt{\delta_0\eta^4})$. Thus, after $k \geq k_1^*$, each time when we switch to `Local-Fast`, it will exhibit linear convergence during time interval $[\mathcal{I}_{\mathrm{lf}}^k : \widetilde{\mathcal{I}}_{\mathrm{lf}}^k]$.

**Hidden Phase II.** After $k \geq k_1^*$, `Local-Fast` enjoys fast linear convergence and becomes the main contributor to the convergence (see segments $\overline{AB}, \overline{CD}$ in Figure 1). Thanks to the fast convergence of `Local-Fast`, in this phase, $\mathrm{dist}(\boldsymbol{z}^t, \mathcal{Z}^*)$ can be much smaller than $C'/t$. Note that we use $\boldsymbol{z}^{\widetilde{\mathcal{I}}_{\mathrm{lf}}^{k-1}}$ as the initial policy of the $k$-th call to `Global-Slow`. Thus, `Global-Slow` could possibly cause $\mathrm{dist}(\boldsymbol{z}^t, \mathcal{Z}^*)$ to increase. However, instead of bounding $\mathrm{dist}(\boldsymbol{z}^t, \mathcal{Z}^*)$ by (2), now (3) can provide a tighter bound for $\mathrm{dist}(\boldsymbol{z}^t, \mathcal{Z}^*)$ when calling `Global-Slow` during Hidden Phase II. (3) implies that $\mathrm{dist}(\boldsymbol{z}^t, \mathcal{Z}^*)$ increases at most geometrically when running `Global-Slow` (see segments $\overline{BC}, \overline{DE}$ in Figure 1). After $2^k \geq O(\frac{1}{c_0\eta^2} \log(D_0\Gamma_0))$ $(k \geq k_2^*)$, the possible increase of $\mathrm{dist}(\boldsymbol{z}^t, \mathcal{Z}^*)$ caused by `Global-Slow` is much less than the decrease caused by

`Local-Fast`, and thus, can be "omitted". More specifically, in $\overline{AB}$, $\text{dist}^2(\boldsymbol{z}^t, \mathcal{Z}^*)$ converges at rate of $1 - c_0\eta^2$ for $|[\mathcal{I}_{\text{lf}}^k : \widetilde{\mathcal{I}}_{\text{lf}}^k]| = 4^k$ iterations, while in $\overline{BC}$, $\text{dist}^2(\boldsymbol{z}^t, \mathcal{Z}^*)$ diverges at rate of $D_0$ for $|[\mathcal{I}_{\text{gs}}^{k+1} : \widetilde{\mathcal{I}}_{\text{gs}}^{k+1}]| = 2^{k+1}$ iterations. Then, since $4^k/2^{k+1} = 2^{k-1}$, if one step increase of `Global-Slow` is much smaller than $2^{k-1}$ steps of decrease of `Local-Fast`, i.e., $D_0(1 - c_0\eta^2/2)^{2^{k-1}} \ll 1$, then, we obtain the global linear convergence (see the line $\overline{AC}$ in Figure 1).

Hidden Phase I has at most $O(4^{k^*})$ steps, where $O(4^{k^*})$ is polynomial in $C', 1/c_0, 1/\delta_0, 1/\eta, 1/\eta'$ and only logarithmic in $D_0, \Gamma_0$. Then, it enters Hidden Phase II and linear convergence begins. This yields the global linear convergence. The formal proof is deferred to Appendix E.

**Theorem 1** *Let* $\{\boldsymbol{z}^t = (\boldsymbol{x}^t, \boldsymbol{y}^t)\}_{t \in [0:T]}$ *be the policy pairs played when running* `Homotopy-PO` *(Algorithm 1). Then, there exists a problem-dependent constant* $D \leq \widetilde{O}(\text{poly}(C', 1/c_0, 1/\delta_0, 1/\eta, 1/\eta'))$ *such that for any* $t \in [0 : T]$*, we have* $\text{dist}^2(\boldsymbol{z}^t, \mathcal{Z}^*) \leq 2S \max\{\Gamma_0, 1\} \cdot \left(1 - \frac{c_0\eta^2}{48}\right)^{t-D}$*, where the value of* $C', c_0, \delta_0, \Gamma_0$ *can be found in the definitions of* `Global-Slow` *and* `Local-Fast`.

As $D$ is independent of $t$, Theorem 1 guarantees the global linear convergence of `Homotopy-PO`.

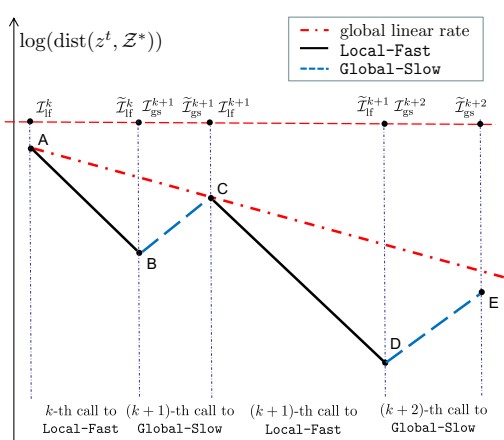

Figure 1: An illustration of upper bound for $\log(\text{dist}(\boldsymbol{z}^t, \mathcal{Z}^*))$ in Hidden Phase II.

## 3.2 EXAMPLES OF BASE ALGORITHMS

We introduce the averaging independent optimistic gradient descent/ascent (Averaging OGDA) method and the independent optimistic policy gradient descent/ascent (OGDA) method which will serve as examples for `Global-Slow` and `Local-Fast` respectively. Both Averaging OGDA and OGDA are symmetric, rational and decentralized algorithms. The pseudocodes are in Appendix G.

**Example of Global-Slow base algorithm (Averaging OGDA).** By running `Averaging-OGDA([T_1 : T_2], z̃, η')` with initial policy $\widetilde{\boldsymbol{z}} = (\widetilde{\boldsymbol{x}}, \widetilde{\boldsymbol{y}})$, the min-player initializes $\widetilde{\boldsymbol{x}}^{T_1} = \boldsymbol{x}^{T_1} = \widetilde{\boldsymbol{x}}$ and $\underline{V}^{T_1}(s) = V^{\dagger,\widetilde{y}}(s)$, the max-player initializes $\widetilde{\boldsymbol{y}}^{T_1} = \boldsymbol{y}^{T_1} = \widetilde{\boldsymbol{y}}$ and $\overline{V}^{T_1}(s) = V^{\widetilde{x},\dagger}(s)$, and they update for $t \in [T_1 + 1 : T_2]$ as follows:

$$
\begin{aligned}
\underline{V}^t(s) &= \min_{a \in \mathcal{A}} \sum_{j=T_1}^{t-1} \alpha_{t-T_1}^{j-T_1+1} \underline{\boldsymbol{q}}_s^j(a), & \overline{V}^t(s) &= \max_{b \in \mathcal{B}} \sum_{j=T_1}^{t-1} \alpha_{t-T_1}^{j-T_1+1} \overline{\boldsymbol{q}}_s^j(b), \\
\boldsymbol{x}_s^t &= \mathcal{P}_{\Delta_{\mathcal{A}}} \left( \widetilde{\boldsymbol{x}}_s^{t-1} - \eta' \underline{\boldsymbol{q}}_s^{t-1} \right), & \boldsymbol{y}_s^t &= \mathcal{P}_{\Delta_{\mathcal{B}}} \left( \widetilde{\boldsymbol{y}}_s^{t-1} + \eta' \overline{\boldsymbol{q}}_s^{t-1} \right), \\
\widetilde{\boldsymbol{x}}_s^t &= \mathcal{P}_{\Delta_{\mathcal{A}}} \left( \widetilde{\boldsymbol{x}}_s^{t-1} - \eta' \underline{\boldsymbol{q}}_s^t \right), & \widetilde{\boldsymbol{y}}_s^t &= \mathcal{P}_{\Delta_{\mathcal{B}}} \left( \widetilde{\boldsymbol{y}}_s^{t-1} + \eta' \overline{\boldsymbol{q}}_s^t \right),
\end{aligned}
\tag{6}
$$

where $\underline{\boldsymbol{q}}_s^j = \boldsymbol{Q}_s[\underline{V}^j]\boldsymbol{y}_s^j$, $\overline{\boldsymbol{q}}_s^j = \boldsymbol{Q}_s[\overline{V}^j]^\top \boldsymbol{x}_s^j$, and $\boldsymbol{Q}_s[\cdot]$ is the Bellman target operator defined in the introduction part. The min-player and the max-player compute the average policies

$$
\widehat{\boldsymbol{x}}^{[T_1:T_2]} = \sum_{t=T_1}^{T_2} \alpha_{T_2-T_1+1}^{t-T_1+1} \boldsymbol{x}^t, \qquad \widehat{\boldsymbol{y}}^{[T_1:T_2]} = \sum_{t=T_1}^{T_2} \alpha_{T_2-T_1+1}^{t-T_1+1} \boldsymbol{y}^t.
\tag{7}
$$

We use the classical averaging stepsizes $\{\alpha_t^j\}$ from Jin et al. (2018): $\alpha_t = \frac{H+1}{H+t}$, $\alpha_t^j = \alpha_j \prod_{k=j+1}^t (1 - \alpha_k)$ $(1 \leq j \leq t-1)$, $\alpha_t^t = \alpha_t$, with $H = \frac{1+\gamma}{1-\gamma}$. In Averaging OGDA, $\boldsymbol{x}^t, \boldsymbol{y}^t$

are the policies played at iteration $t \in [T_1 : T_2]$, and $\widetilde{\boldsymbol{x}}^t, \underline{V}^t, \widetilde{\boldsymbol{y}}^t, \overline{V}^t$ are local auxiliary variables help to generate such sequences of $\boldsymbol{x}^t, \boldsymbol{y}^t$. The global convergence and geometric boundedness of Averaging OGDA are shown in Section 4. The RHS of (2) in the definition of `Global-Slow` can be directly extended to different convergence rates and more algorithms such as Wei et al. (2021) with a different initialization can serve as the generalized `Global-Slow`. More details are in Appendix H.

**Example of Local-Fast base algorithm (OGDA).** By running $\texttt{OGDA}([T_1 : T_2], \widehat{\boldsymbol{z}}, \eta)$ with initial policy $\widehat{\boldsymbol{z}} = (\widehat{\boldsymbol{x}}, \widehat{\boldsymbol{y}})$, the min-player initializes $\widetilde{\boldsymbol{x}}^{T_1} = \boldsymbol{x}^{T_1} = \widehat{\boldsymbol{x}}$, the max-player initializes $\widetilde{\boldsymbol{y}}^{T_1} = \boldsymbol{y}^{T_1} = \widehat{\boldsymbol{y}}$, and they update for $t \in [T_1 + 1 : T_2]$ as follows:

$$
\begin{aligned}
\boldsymbol{x}_s^t &= \mathcal{P}_{\Delta_\mathcal{A}}\left(\widetilde{\boldsymbol{x}}_s^{t-1} - \eta \boldsymbol{Q}_s^{t-1}\boldsymbol{y}_s^{t-1}\right), & \boldsymbol{y}_s^t &= \mathcal{P}_{\Delta_\mathcal{B}}\left(\widetilde{\boldsymbol{x}}_s^{t-1} + \eta\left(\boldsymbol{Q}_s^{t-1}\right)^\top \boldsymbol{x}_s^{t-1}\right), \\
\widetilde{\boldsymbol{x}}_s^t &= \mathcal{P}_{\Delta_\mathcal{A}}\left(\widetilde{\boldsymbol{x}}_s^{t-1} - \eta \boldsymbol{Q}_s^t \boldsymbol{y}_s^t\right), & \widetilde{\boldsymbol{y}}_s^t &= \mathcal{P}_{\Delta_\mathcal{B}}\left(\widetilde{\boldsymbol{y}}_s^{t-1} + \eta\left(\boldsymbol{Q}_s^t\right)^\top \boldsymbol{x}_s^t\right),
\end{aligned} \tag{8}
$$

where we abbreviate $\boldsymbol{Q}_s^t = \boldsymbol{Q}_s^{\boldsymbol{x}^t, \boldsymbol{y}^t}$ for $t \in [T_1 : T_2]$. In OGDA, $\boldsymbol{x}^t, \boldsymbol{y}^t$ are the policies played at iteration $t \in [T_1 : T_2]$, while $\widetilde{\boldsymbol{x}}^t, \widetilde{\boldsymbol{y}}^t$ are local auxiliary variables. OGDA can be considered as a natural extension of the classical optimistic gradient descent/ascent to Markov games because when there is only one state ($S = 1$), it reduces to the classical OGDA method for matrix games.

The proof for local linear convergence of OGDA is of independent interest and shown in Section 5.

### 3.3 GLOBAL LINEAR CONVERGENCE

We can instantiate the meta algorithm `Homotopy-PO` by using OGDA (8) as `Local-Fast` and Averaging OGDA (6) as `Global-Slow`. This gives the provable global linear convergence for zero-sum discounted Markov games as in Theorem 2. In practice, `Homotopy-PO` can also exhibit linear convergence. Our numerical experiments are in Appendix I due to the space limitation.

**Theorem 2** *(Global Linear Convergence) Let $\{\boldsymbol{z}^t = (\boldsymbol{x}^t, \boldsymbol{y}^t)\}_{t \in [0:T]}$ be the policy pairs played when running `Homotopy-PO` (Algorithm 1), where `Local-Fast` uses OGDA with $\eta \leq \frac{(1-\gamma)^{\frac{5}{2}}}{32\sqrt{S}(A+B)}$, and `Global-Slow` uses Averaging OGDA with $\eta' \leq \frac{1-\gamma}{16\max\{A,B\}}$. Then, there exist problem-dependent constants $c \in (0,1)$ and $M > 0$ such that for any $t \in [0:T]$,*

$$
\mathrm{dist}^2(\boldsymbol{z}^t, \mathcal{Z}^*) \leq \frac{16S^2}{1-\gamma} \cdot \left(1 - c\eta^2\right)^{t - \frac{M\log^2(SAB/(c_+\eta\eta'))}{\eta^4\eta'^2}}, \tag{9}
$$

*where $c = \Omega(c_+^2/\mathrm{poly}(S, A, B, 1/(1-\gamma)))$ and $M = \mathrm{poly}(S, A, B, 1/(1-\gamma), 1/c_+)$.*

**Decentralized implementation.** Since both OGDA and Averaging OGDA are symmetric, rational and decentralized, our instantiation of `Homotopy-PO` is naturally a symmetric, rational and decentralized algorithm. Pseudocodes and more details can be found in Appendix G.

**Linear rate comparison with matrix games.** For the convex-concave matrix games, Gilpin et al. (2012) and Wei et al. (2020) propose centralized/decentralized methods with global linear rates of $(1 - O(\varphi(\boldsymbol{G})))^t$ and $(1 - O(\varphi(\boldsymbol{G})^2))^t$ respectively, where $\varphi(\boldsymbol{G})$ is a certain condition measure of matrix $\boldsymbol{G}$. Details of $\varphi(\boldsymbol{G})$ are in Lemma 22. The constant $c_+$ in (1) can be naturally defined as $c_+ = \min_{s \in \mathcal{S}} \varphi(\boldsymbol{Q}_s^*)$ (see Corollary 2). Thus, the global linear convergence rate for zero-sum Markov games in Theorem 2 is comparable to solving matrix games up to polynomials in $S, A, B, 1/(1-\gamma)$.

## 4 GLOBAL CONVERGENCE AND GEOMETRIC BOUNDEDNESS OF AVERAGING OGDA

We show that the Averaging OGDA (6) method has $O(\log T/T)$ global convergence rate and geometric boundedness. Thus, Averaging OGDA can be serve as `Global-Slow` in `Homotopy-PO`.

**Global convergence.** The proof for global convergence of Averaging OGDA adapts several standard techniques from Markov games (Zhang et al., 2022; Wei et al., 2021). We attach its proof in Appendix D.1 for completeness.

**Theorem 3** *(Global Convergence) Let $\widehat{\boldsymbol{z}}^{[T_1:T_2]} = (\widehat{\boldsymbol{x}}^{[T_1:T_2]}, \widehat{\boldsymbol{y}}^{[T_1:T_2]})$ be the average policy (7) generated by running* `Averaging-OGDA`$([T_1:T_2], \tilde{\boldsymbol{z}}, \eta')$ *with* $\eta' \leq \frac{1-\gamma}{16 \max\{A,B\}}$. *There is a problem-dependent constant* $C' = O(\frac{\sqrt{S}(A+B)}{c_+ (1-\gamma)^6})$ *such that* $\widehat{\boldsymbol{z}}^{[T_1:T_2]}$ *satisfies*

$$\text{dist}\left(\widehat{\boldsymbol{z}}^{[T_1:T_2]}, \mathcal{Z}^*\right) \leq \frac{C' \cdot \log\left(T_2 - T_1 + 1\right)}{\eta'\left(T_2 - T_1 + 1\right)}. \tag{10}$$

This gives the $\widetilde{O}(1/T)$ global convergence rate of `Global-Slow`. This property guarantees that `Global-Slow` can serve as a "guide" in Hidden Phase I as described in Section 3.1.

**Geometric boundedness.** The proof of geometric boundedness mainly relies on the stability of projected gradient descent with respect to the NE set (Appendix B). We will prove that the increase of $\text{dist}(\boldsymbol{z}^t, \mathcal{Z}^*)$ is at most geometric by providing mutual bounds among $\{\text{dist}(\boldsymbol{z}^t, \mathcal{Z}^*)\}$, $\{\text{dist}(\widetilde{\boldsymbol{z}}^t, \mathcal{Z}^*)\}$, $\{\|\overline{V}^t - \underline{V}^t\|_\infty\}$, $\{\max_b \overline{\boldsymbol{q}}_s^t(b) - \min_a \underline{\boldsymbol{q}}_s^t(a)\}$ inductively. The formal proof is in Appendix D.2.

**Theorem 4** *(Geometric Boundedness) Let $\{\boldsymbol{z}^t\}_{t \in [T_1:T_2]}$, $\widehat{\boldsymbol{z}}^{[T_1:T_2]}$ be the policy pairs played and the average policy pair generated by running* `Averaging-OGDA`$([T_1:T_2], \tilde{\boldsymbol{z}}, \eta')$ *with* $\eta' \leq 1$, *then there is a problem-dependent constant* $D_0 = O(\frac{S(A+B)^2}{(1-\gamma)^4})$ *(possibly $D_0 > 1$) such that for any $t \in [T_1:T_2]$,*

$$\text{dist}^2(\boldsymbol{z}^t, \mathcal{Z}^*) \leq D_0^{t-T_1} \cdot \text{dist}^2(\tilde{\boldsymbol{z}}, \mathcal{Z}^*). \tag{11}$$

$$\text{dist}^2(\widehat{\boldsymbol{z}}^{[T_1:T_2]}, \mathcal{Z}^*) \leq D_0^{T_2-T_1} \cdot \text{dist}^2(\tilde{\boldsymbol{z}}, \mathcal{Z}^*). \tag{12}$$

This property is important in our proof for the main theorem (Theorem 2). It means that when running `Global-Slow` in Hidden Phase II, though $\text{dist}(\boldsymbol{z}^t, \mathcal{Z}^*)$ can possibly increase due to $D_0 > 1$, $\text{dist}(\boldsymbol{z}^t, \mathcal{Z}^*)$ can only increase geometrically (see segments $\overline{BC}$, $\overline{DE}$ in Figure 1).

## 5 LOCAL LINEAR CONVERGENCE OF OGDA

We show that OGDA (8) has local linear convergence. Thus, OGDA can be used as the base algorithm `Local-Fast` in `Homotopy-PO`. To prove the local linear convergence, we provide a novel analysis for OGDA which starts from the following two observations.

**Observation I** (Lemma 7) saddle-point metric subregularity (SP-MS) can be generalized to Markov games, i.e., for any policy pair $\boldsymbol{z} \in \mathcal{Z}$ and $s \in \mathcal{S}$,

$$V^{\boldsymbol{x},\dagger}(s) - V^{\dagger,\boldsymbol{y}}(s) \geq c_+ \cdot \text{dist}(\boldsymbol{z}_s, \mathcal{Z}^*). \tag{13}$$

Observation I guarantees the progress of projected gradient descent/ascent (PGDA) is substantial.

**Observation II** (Appendix B, Lemma 10) when running OGDA (8), the change in policy pair becomes smaller when $\boldsymbol{z}^t, \widetilde{\boldsymbol{z}}^t$ are approaching the NE set, i.e.,

$$\|\boldsymbol{z}^{t+1} - \boldsymbol{z}^t\|^2 + \|\widetilde{\boldsymbol{z}}^t - \widetilde{\boldsymbol{z}}^{(t-1)}\|^2 \leq O(\text{dist}^2(\widetilde{\boldsymbol{z}}^{t-1}, \mathcal{Z}^*) + \|\widetilde{\boldsymbol{z}}^{t-1} - \boldsymbol{z}^{t-1}\|^2). \tag{14}$$

Observation II implies the stability of state visitation distribution. Intuitively, it can help us relate Markov games to matrix games in a neighborhood of the NE set.

**Theorem 5** *(Local Linear Convergence) Let $\{\boldsymbol{z}^t\}_{t \in [T_1:T_2]}$ be the policy pairs played when running* `OGDA`$([T_1:T_2], \widehat{\boldsymbol{z}}, \eta)$ *with stepsize* $\eta \leq \frac{(1-\gamma)^{\frac{5}{2}}}{32\sqrt{S}(A+B)}$. *Then, there are problem-dependent constants $c \in (0,1)$, $\delta_0 > 0$ such that if $\text{dist}^2(\widehat{\boldsymbol{z}}, \mathcal{Z}^*) \leq \delta_0 \eta^4$, then for any $t \geq T_1$,*

$$\text{dist}^2(\boldsymbol{z}^t, \mathcal{Z}^*) \leq \frac{8S}{1-\gamma}\left(1 - \frac{c_0 \eta^2}{48}\right)^{t-T_1} \text{dist}^2(\widehat{\boldsymbol{z}}, \mathcal{Z}^*), \tag{15}$$

*where $c_0 = \Omega(c_+^2/\text{poly}(S, A, B, 1/(1-\gamma)))$ and $\delta_0 = \Omega(c_+^4/\text{poly}(S, A, B, 1/(1-\gamma)))$.*

We provide a proof sketch below. The formal proof is in Appendix C.

**Proof sketch of Theorem 5.** We denote the projections by $\widetilde{\boldsymbol{x}}^{t*} = \mathcal{P}_{\mathcal{X}^*}(\widetilde{\boldsymbol{x}}^t)$, $\widetilde{\boldsymbol{y}}^{t*} = \mathcal{P}_{\mathcal{Y}^*}(\widetilde{\boldsymbol{y}}^t)$, $\widetilde{\boldsymbol{z}}^{t*} = \mathcal{P}_{\mathcal{Z}^*}(\widetilde{\boldsymbol{z}}^t)$. Let $\boldsymbol{\rho}_0$ be the uniform distribution on $\mathcal{S}$. Our proof for the local linear convergence of OGDA has the following steps.

**Step I: One-step analysis (Appendix C.1).** The main obstacle in adopting standard analysis for normal form games to Markov games lies in the fact that as Markov games are nonconvex-nonconcave, it may happen that $\left\langle \boldsymbol{x}_s^{t+1} - \widetilde{\boldsymbol{x}}_s^{t*}, \boldsymbol{Q}_s^{t+1}\boldsymbol{y}_s^{t+1}\right\rangle + \left\langle \widetilde{\boldsymbol{y}}_s^{t*} - \boldsymbol{y}_s^{t+1}, \left(\boldsymbol{Q}_s^{t+1}\right)^\top \boldsymbol{x}_s^{t+1}\right\rangle < 0$.

To overcome this problem, our strategy is to consider a weighted sum of $\left\langle \boldsymbol{x}_s^{t+1} - \widetilde{\boldsymbol{x}}_s^{t*}, \boldsymbol{Q}_s^{t+1}\boldsymbol{y}_s^{t+1}\right\rangle$ and $\left\langle \widetilde{\boldsymbol{y}}_s^{t*} - \boldsymbol{y}_s^{t+1}, \left(\boldsymbol{Q}_s^{t+1}\right)^\top \boldsymbol{x}_s^{t+1}\right\rangle$. As $(\widetilde{\boldsymbol{x}}^{t*}, \widetilde{\boldsymbol{y}}^{t*})$ attains a Nash equilibrium, $V^{\boldsymbol{x}^{t+1}, \widetilde{\boldsymbol{y}}^{t*}}(\boldsymbol{\rho}_0) - V^{\widetilde{\boldsymbol{x}}^{t*}, y^{t+1}}(\boldsymbol{\rho}_0) \geq 0$. Then, by applying performance difference lemma (Lemma 20), we have

$$\sum_{s\in\mathcal{S}} \boldsymbol{d}_x^{t+1}(s)\left\langle \boldsymbol{x}_s^{t+1} - \widetilde{\boldsymbol{x}}_s^{t*}, \boldsymbol{Q}_s^{t+1}\boldsymbol{y}_s^{t+1}\right\rangle + \boldsymbol{d}_y^{t+1}(s)\left\langle \widetilde{\boldsymbol{y}}_s^{t*} - \boldsymbol{y}_s^{t+1}, \left(\boldsymbol{Q}_s^{t+1}\right)^\top \boldsymbol{x}_s^{t+1}\right\rangle \geq 0, \quad (16)$$

where $\boldsymbol{d}_x^t(s) = \boldsymbol{d}_{\boldsymbol{\rho}_0}^{\widetilde{\boldsymbol{x}}^{(t-1)*}, y^t}(s)$ and $\boldsymbol{d}_y^t(s) = \boldsymbol{d}_{\boldsymbol{\rho}_0}^{\boldsymbol{x}^t, \widetilde{\boldsymbol{y}}^{(t-1)*}}(s)$ for $t > T_1$.

Since $\boldsymbol{Q}_s^t$ is smooth in $\boldsymbol{z}^t$, we can adopt standard regret analysis for optimistic gradient descent in normal form games to bound $\left\langle \boldsymbol{x}_s^{t+1} - \widetilde{\boldsymbol{x}}_s^{t*}, \boldsymbol{Q}_s^{t+1}\boldsymbol{y}_s^{t+1}\right\rangle$ and $\left\langle \widetilde{\boldsymbol{y}}_s^{t*} - \boldsymbol{y}_s^{t+1}, \left(\boldsymbol{Q}_s^{t+1}\right)^\top \boldsymbol{x}_s^{t+1}\right\rangle$. This yields the following inequality (which is equivalent to Lemma 4)

$$\Lambda^{t+1} \leq \Lambda^t + \underbrace{\widetilde{\Theta}^t - \Theta^t}_{\text{Step III: stability of } \boldsymbol{d}_x^t,\, \boldsymbol{d}_y^t} - \frac{C_\Lambda}{2}\|\widetilde{\boldsymbol{z}}^t - \boldsymbol{z}^t\|^2 - \underbrace{C_\Lambda(\|\widetilde{\boldsymbol{z}}^{t+1} - \boldsymbol{z}^{t+1}\|^2 + \|\widetilde{\boldsymbol{z}}^t - \boldsymbol{z}^t\|^2)}_{\text{Step II: progress of PGDA}}, \quad (17)$$

where $C_\Lambda = \frac{1-\gamma}{4S}$, $\Theta^t = \sum_{s\in\mathcal{S}} \boldsymbol{d}_x^t(s)\text{dist}^2(\widetilde{\boldsymbol{x}}_s^t, \mathcal{X}_s^*) + \boldsymbol{d}_y^t(s)\text{dist}^2(\widetilde{\boldsymbol{y}}_s^t, \mathcal{Y}_s^*)$, $\widetilde{\Theta}^t = \sum_{s\in\mathcal{S}} \boldsymbol{d}_x^{t+1}(s)\text{dist}^2(\widetilde{\boldsymbol{x}}_s^t, \mathcal{X}_s^*) + \boldsymbol{d}_y^{t+1}(s)\text{dist}^2(\widetilde{\boldsymbol{y}}_s^t, \mathcal{Y}_s^*)$, and $\Lambda^t = \Theta^t + C_\Lambda\left\|\widetilde{\boldsymbol{z}}^t - \boldsymbol{z}^t\right\|^2$. As $\boldsymbol{d}_x^t(s) \geq \boldsymbol{d}_s^{\widetilde{\boldsymbol{x}}^{t*}, y^{t+1}}(s) \geq \frac{1-\gamma}{S}$, to show the local linear convergence of OGDA, it suffices to show that for $\Lambda^t$.

**Step II: Progress of projected gradient descent (Appendix C.2).** We combine (13) from Observation I and standard analysis of projected gradient descent (Lemma 5) to essentially show that there exists a problem-dependent constant $c_+' = O(c_+^2\eta^2/\text{poly}(S, A, B, 1/(1-\gamma)))$ such that

$$C_\Lambda(\|\boldsymbol{z}^{t+1} - \widetilde{\boldsymbol{z}}^t\|^2 + \|\widetilde{\boldsymbol{z}}^t - \boldsymbol{z}^t\|^2) \geq c_+' \cdot \Theta^t. \quad (18)$$

**Step III: Stability of visitation distribution near the NE set (Appendix C.3).** Using (14) from Observation II and the non-expansive property of projections onto convex sets, we will show $\|\boldsymbol{z}^{t+1} - \boldsymbol{z}^t\|^2 + \|\widetilde{\boldsymbol{z}}^{t*} - \widetilde{\boldsymbol{z}}^{(t-1)*}\|^2 \leq O\left(\Lambda^{t-1}\right)$. Then, as $\boldsymbol{d}_x^t(s), \boldsymbol{d}_y^t(s)$ in (16) are continuous in $\boldsymbol{z}^t$ and $\widetilde{\boldsymbol{z}}^t$, we can find a problem-dependent constant $\delta = O(c_+^4\eta^4/\text{poly}(S, A, B, 1/(1-\gamma)))$ such that if $\Lambda^{t-1} \leq \delta$, then $\widetilde{\Theta}^t$ can be bounded by

$$\widetilde{\Theta}^t \leq (1 + \frac{c_+'}{2})\Theta^t. \quad (19)$$

**Step IV: Induction (Appendix C.4).** By (17), (18), (19) from Steps I, II, III above, intuitively, we can deduce that when $\Lambda^{t-1} \leq \delta$, the "one-step linear convergence" is achieved

$$\Lambda^{t+1} \leq \Lambda^t + \frac{c_+'}{2}\Theta^t - \frac{C_\Lambda}{2}\|\widetilde{\boldsymbol{z}}^t - \boldsymbol{z}^t\|^2 - c_+'\Theta^t = \Lambda^t - \frac{c_+'}{2}\Theta^t - \frac{C_\Lambda}{2}\|\widetilde{\boldsymbol{z}}^t - \boldsymbol{z}^t\|^2$$

$$\leq \Lambda^t - \min\{\frac{c_+'}{2}, \frac{1}{2}\}(\Theta^t + C_\Lambda\|\widetilde{\boldsymbol{z}}^t - \boldsymbol{z}^t\|^2) = \left(1 - \frac{c_+'}{2}\right)\Lambda^t.$$

By a coupled induction with Step III, given the initial policy $\widehat{\boldsymbol{z}}$ in the neighborhood $\overline{B}(\mathcal{Z}^*, \sqrt{\delta})$ of the NE set, the policy pair $\boldsymbol{z}^t$ will always stay in $\overline{B}(\mathcal{Z}^*, \sqrt{\delta})$. Then, $\Lambda^t$ converges linearly.

This yields the local linear convergence of OGDA as in Theorem 5.

ACKNOWLEDGEMENT

We would like to thank anonymous reviewers for their helpful comments.

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

## A    FURTHER RELATED WORK

**Sampling-based two-player zero-sum Markov games.**  Finding Nash equilibria of zero-sum Markov games in sampling-based/online setting is receiving extensive studies in recent years (Zhang et al., 2020; Liu et al., 2021; Bai et al., 2020; Bai & Jin, 2020; Brafman & Tennenholtz, 2002; Sidford et al., 2020; Tian et al., 2021; Wei et al., 2017; Xie et al., 2020; Chen et al., 2022; Li et al., 2022). In this paper, we are more concerned with known model or perfect recall settings. Specifically, our focus is on how to design efficient policy optimization methods to solve the minimax optimization problem formulated by zero-sum Markov games. Therefore, these works are not directly relevant to us.

**Minimax optimization.**  Zero-sum Markov games are usually studied as minimax optimization problems. Finding Nash equilibria/saddle points in convex-concave and nonconvex-concave problems have been extensively studied (Lin et al., 2020b; Tseng, 1995; Mokhtari et al., 2020a;b; Thekumparampil et al., 2019; Lu et al., 2020; Nouiehed et al., 2019; Kong & Monteiro, 2021; Lin et al., 2020a).

Due to the nonconcexity-nonconcavity of zero-sum Markov games, existing tools in convex-concave and nonconvex-concave optimization are hard to be adapted here. For nonconvex-nonconcave optimization, Nouiehed et al. (2019); Yang et al. (2020) study two-timescale/asymmetric gradient descent/ascent methods under the PŁ condition, where two-time-scale/asymmetric refers to that one-player chooses a much smaller step than its opponent, or one-player waits until its opponent finds the best response. Daskalakis et al. (2020) establish the two-sided gradient dominance condition for zero-sum Markov games, which can be related to the two-sided PŁ condition. And they utilize this gradient dominance property to study the finite-time performance of two-timescale gradient descent/ascent (GDA) algorithm in zero-sum Markov games and prove its sub-linear convergence rate. Zhao et al. (2022) consider function approximation and propose another two-timescale method that finds a NE at $\widetilde{O}(1/t)$ rate.

**Matrix games.**  Matrix games are a special kind of Markov games with single state. Since matrix games are naturally convex-concave, global linear convergence has been achieved in finding Nash equilibria in matrix games (Gilpin et al., 2012; Wei et al., 2020). The linear convergence of their algorithms relies on the following fact: the duality gap of one policy pair can be lower bounded by its distance to the NE set multiplied by a matrix condition measure (see Lemma 22 for more details). This property is called saddle-point metric subregularity (SP-MS) in Wei et al. (2020). Similar techniques have been extended to extensive form games and get linear convergence (Lee et al., 2021; Piliouras et al., 2022).

**Averaging techniques.**  Averaging techniques are usually used to tame nonstationarity in approximate Q functions, where the players utilize information from past iterations to obtain better approximations for value functions and policy gradients. Wei et al. (2021) propose an actor-critic OGDA method which uses averaged value functions to get more accurate policy gradients, and then achieve a $O(1/\sqrt{t})$ last-iterate convergence rate to the NE set. Zhang et al. (2022) propose a modified OFTRL method, where the min-player and the max-players employ a lower and upper bound for value functions separately. The lower and upper bounds are computed from approximate Q-functions in past iterations. Their method has $\widetilde{O}(1/t)$ convergence rate for the average policy. Yang & Ma (2022) show that the average policy of an OFTRL method whose approximate Q-functions are also averaged from past estimates can find Nash equilibria at the rate of $O(1/t)$ with no logarithmic factors.

**Regularized Markov games.**  Adding regularizer can greatly refine the structures of matrix games and Markov games and is considered a powerful tool to tackle nonconvexity-nonconcavity of zero-sum Markov games. Cen et al. (2021) study entropy-regularized matrix games and achieve dimension-free last-iterate linear convergence to the quantal response equilibrium which is an approximation for the Nash equilibrium. They further connect value iteration with matrix games and use the contraction property of the Bellman operator to prove the linear convergence to the quantal response equilibrium of the Markov games. By choosing small regularization weights, their method can find an $\epsilon$-Nash equilibrium in $\widetilde{O}(1/\epsilon)$ iterations. Zeng et al. (2022) also consider adding entropy regularization to help find Nash equilibria in zero-sum Markov games. They prove the $O(t^{-1/3})$ convergence rate of a variant of GDA by driving regularization weights dynamically to zero. How-

ever, to obtain Nash equilibria, the regularization weights have to be reduced to zero in the learning process. The time complexities of existing regularized methods are usually inversely proportional to the regularization weights. Reducing such weights to zero could possibly lead to sub-linear rates.

# B  Stability of projected gradient descent/ascent with respect to the Nash equilibrium set

In this section, we show the stability of the distance to the Nash equilibrium set after one step of projected gradient descent/ascent. The results in this section are important in our proofs for the local linear convergence of OGDA and the geometric boundedness of Averaging OGDA.

The following lemma shows that projected gradient descent/ascent is very "stable" on the NE set. More specifically, if the players have attained a Nash equilibirum, then, their policies will remain invariant by doing projected gradient descent/ascent.

**Lemma 1** *For any Nash equilibrium $z = (x, y) \in \mathcal{Z}^*$, let $x^+, y^+$ be the variables after one step of projected policy gradient descent/ascent with stepsize $\eta > 0$, i.e., for $s \in \mathcal{S}$*

$$x_s^+ = \mathcal{P}_{\Delta_{\mathcal{A}}} \left( x_s - \eta Q_s^* y_s \right), \ y_s^+ = \mathcal{P}_{\Delta_{\mathcal{B}}} \left( y_s + \eta \left( Q_s^* \right)^\top x_s \right).$$

*Let $z^+ = (x^+, y^+)$, then, $z^+ = z$.*

*Proof of Lemma 1.* Let $u_s^* = Q_s^* y_s$. By Lemma 19, $x_s \in \arg\min_{x_s' \in \Delta_{\mathcal{A}}} \langle x_s', Q_s^* y_s \rangle$. Equivalently, $\text{supp}(x_s) \subseteq \arg\min_a u_s^*(a)$, where $\text{supp}(x_s)$ is the index set of the nonzero entries in $x_s$.

Next, we will show $x_s^+ = x_s$. Since $x_s^+$ is the projection onto $\Delta_{\mathcal{A}}$ and Slater's condition is naturally satisfied in the simplex constraint, by the KKT conditions,

$$
\begin{aligned}
& x_s^+(a) - x_s(a) + \eta u_s^*(a) - \lambda_0 + \lambda_a = 0, \\
& \lambda_a x_s^+(a) = 0, \ \forall a \in [A], \\
& \lambda_a \geq 0, \ \forall a \in [A], \\
& x_s^+(a) \geq 0, \ \forall a \in [A], \\
& \sum_{a \in [A]} x_s^+(a) = 1.
\end{aligned}
$$

Then, for $a \in [A]$, $\lambda_a > 0$ only if $x_s^+(a) = 0$; otherwise, $x_s^+(a) = x_s(a) - \eta u_s^*(a) + \lambda_0$. Thus,

$$x_s^+(a) = \max\left\{ x_s(a) - \eta u_s^*(a) + \lambda_0, 0 \right\}.$$

If $\lambda_0 = \eta \cdot \min_{a \in [A]} u_s^*(a)$, then by combining with $\text{supp}(x_s) \subseteq \arg\min_a u_s^*(a)$, we have

$$\max\left\{ x_s(a) - \eta u_s^*(a) + \lambda_0, 0 \right\} = x_s(a),$$

i.e., $\sum_a \max\left\{ x_s(a) - \eta u_s^*(a) + \lambda_0, 0 \right\} = 1$. Thus, for $\lambda_0 > \eta \cdot \min_{a \in [A]} u_s^*(a)$ or $\lambda_0 < \eta \cdot \min_{a \in [A]} u_s^*(a)$, we will have $\sum_a \max\left\{ x_s(a) - \eta u_s^*(a) + \lambda_0, 0 \right\} > 1$ or $\sum_a \max\left\{ x_s(a) - \eta u_s^*(a) + \lambda_0, 0 \right\} < 1$, respectively. To meet the condition $\sum_{a \in [A]} x_s^+(a) = 1$, we have to let $\lambda_0 = \eta \cdot \min_{a \in [A]} u_s^*(a)$. Now,

$$x_s^+(a) = \max\left\{ x_s(a) - \eta u_s^*(a) + \lambda_0, 0 \right\} = x_s(a), \ \forall a \in \mathcal{A}.$$

Analogously, $y_s^+ = y_s$. $\qquad \square$

The following lemma is a perturbed version of Lemma 1.

**Lemma 2** *For any $z = (x, y) \in \mathcal{Z}$, $\widetilde{z} = (\widetilde{x}, \widetilde{y}) \in \mathcal{Z}$ and matrices $\left\{ Q_s, \widehat{Q}_s \right\}_{s \in \mathcal{S}} \subseteq \mathbb{R}^{A \times B}$, let $x^+, y^+$ be the position after one step of projected policy gradient descent/ascent with stepsize $\eta > 0$, i.e., for $s \in \mathcal{S}$*

$$x_s^+ = \mathcal{P}_{\Delta_{\mathcal{A}}} \left( \widetilde{x}_s - \eta Q_s y_s \right), \ y_s^+ = \mathcal{P}_{\Delta_{\mathcal{B}}} \left( \widetilde{y}_s + \eta \left( \widehat{Q}_s \right)^\top x_s \right).$$

*Let $\boldsymbol{z}^+ = (\boldsymbol{x}^+, \boldsymbol{y}^+)$, then,*

$$\left\|\boldsymbol{z}^+ - \widetilde{\boldsymbol{z}}\right\|^2 \leq 8\mathrm{dist}^2\left(\widetilde{\boldsymbol{z}}, \mathcal{Z}^*\right) + 4\eta^2 \sum_{s\in\mathcal{S}} B \max_{(a,b)\in\mathcal{A}\times\mathcal{B}} \left|\boldsymbol{Q}_s(a,b) - \boldsymbol{Q}_s^*(a,b)\right|^2$$

$$+ 4\eta^2 \sum_{s\in\mathcal{S}} A \max_{(a,b)\in\mathcal{A}\times\mathcal{B}} \left|\widehat{\boldsymbol{Q}}_s(a,b) - \boldsymbol{Q}_s^*(a,b)\right|^2$$

$$+ \frac{4\eta^2 \max\{A,B\}^2}{(1-\gamma)^2}\mathrm{dist}^2\left(\boldsymbol{z}, \mathcal{Z}^*\right).$$

*Proof of Lemma 2.* Denote $\boldsymbol{x}^* = \mathcal{P}_{\mathcal{X}^*}(\boldsymbol{x})$, $\boldsymbol{y}^* = \mathcal{P}_{\mathcal{Y}^*}(\boldsymbol{y})$, $\boldsymbol{z}^* = (\boldsymbol{x}^*, \boldsymbol{y}^*)$; $\widetilde{\boldsymbol{x}}^* = \mathcal{P}_{\mathcal{X}^*}(\widetilde{\boldsymbol{x}})$, $\widetilde{\boldsymbol{y}}^* = \mathcal{P}_{\mathcal{Y}^*}(\widetilde{\boldsymbol{y}})$, $\widetilde{\boldsymbol{z}}^* = (\widetilde{\boldsymbol{x}}^*, \widetilde{\boldsymbol{y}}^*)$.

Let $\boldsymbol{u}_s = \boldsymbol{Q}_s \boldsymbol{y}_s$, $\boldsymbol{u}_s^* = \boldsymbol{Q}_s^* \boldsymbol{y}_s^*$, then

$$\|\boldsymbol{u}_s^* - \boldsymbol{u}_s\| \leq \sqrt{B}\|\boldsymbol{u}_s^* - \boldsymbol{u}_s\|_\infty$$

$$\leq \sqrt{B}\left(\max_{(a,b)\in\mathcal{A}\times\mathcal{B}}\left|\boldsymbol{Q}_s(a,b) - \boldsymbol{Q}_s^*(a,b)\right|\|\boldsymbol{y}_s\|_1 + \max_{(a,b)\in\mathcal{A}\times\mathcal{B}}\left|\boldsymbol{Q}_s^*(a,b)\right|\|\boldsymbol{y}_s - \boldsymbol{y}_s^*\|_1\right)$$

$$\leq \sqrt{B}\max_{(a,b)\in\mathcal{A}\times\mathcal{B}}\left|\boldsymbol{Q}_s(a,b) - \boldsymbol{Q}_s^*(a,b)\right| + \frac{B}{1-\gamma}\mathrm{dist}\left(\boldsymbol{y}_s, \mathcal{Y}_s^*\right),$$

i.e.,

$$\|\boldsymbol{u}_s^* - \boldsymbol{u}_s\|^2 \leq 2\left(B \max_{(a,b)\in\mathcal{A}\times\mathcal{B}}\left|\boldsymbol{Q}_s(a,b) - \boldsymbol{Q}_s^*(a,b)\right|^2 + \frac{B^2}{(1-\gamma)^2}\mathrm{dist}^2\left(\boldsymbol{y}_s, \mathcal{Y}_s^*\right)\right). \tag{20}$$

By Lemma 19, $(\widetilde{\boldsymbol{x}}^*, \boldsymbol{y}^*)$ is also a Nash equilibrium. Denote $\widetilde{\boldsymbol{x}}_s^{*+} = \mathcal{P}_{\Delta_{\mathcal{A}}}\left(\widetilde{\boldsymbol{x}}_s^* - \eta\boldsymbol{Q}_s^*\boldsymbol{y}_s^*\right)$. Then, by Lemma 1,

$$\widetilde{\boldsymbol{x}}_s^{*+} = \widetilde{\boldsymbol{x}}_s^*. \tag{21}$$

By triangle inequality, we have

$$\left\|\boldsymbol{x}_s^+ - \widetilde{\boldsymbol{x}}_s\right\| \leq \left\|\boldsymbol{x}_s^+ - \widetilde{\boldsymbol{x}}_s^{*+}\right\| + \left\|\widetilde{\boldsymbol{x}}_s^{*+} - \widetilde{\boldsymbol{x}}_s^*\right\| + \left\|\widetilde{\boldsymbol{x}}_s^* - \widetilde{\boldsymbol{x}}_s\right\|$$

$$= \left\|\mathcal{P}_{\Delta_{\mathcal{A}}}\left(\widetilde{\boldsymbol{x}}_s - \eta\boldsymbol{u}_s\right) - \mathcal{P}_{\Delta_{\mathcal{A}}}\left(\widetilde{\boldsymbol{x}}_s^* - \eta\boldsymbol{u}_s^*\right)\right\| + 0 + \mathrm{dist}\left(\widetilde{\boldsymbol{x}}_s, \mathcal{X}_s^*\right)$$

$$\leq \left\|\widetilde{\boldsymbol{x}}_s - \widetilde{\boldsymbol{x}}_s^*\right\| + \eta\left\|\boldsymbol{u}_s - \boldsymbol{u}_s^*\right\| + \mathrm{dist}\left(\widetilde{\boldsymbol{x}}_s, \mathcal{X}_s^*\right)$$

$$= 2\mathrm{dist}\left(\widetilde{\boldsymbol{x}}_s, \mathcal{X}_s^*\right) + \eta\left\|\boldsymbol{u}_s - \boldsymbol{u}_s^*\right\|,$$

where the first equality is by (21) and the second inequality comes from the fact that for any $\boldsymbol{a}, \boldsymbol{b} \in \mathbb{R}^A$, $\|\mathcal{P}_{\Delta_{\mathcal{A}}}(\boldsymbol{a}) - \mathcal{P}_{\Delta_{\mathcal{A}}}(\boldsymbol{b})\| \leq \|\boldsymbol{a} - \boldsymbol{b}\|$.

Taking square and summing over $s \in \mathcal{S}$ and combining with (20) yield that

$$\left\|\boldsymbol{x}^+ - \widetilde{\boldsymbol{x}}\right\|^2 \leq 8\mathrm{dist}^2\left(\widetilde{\boldsymbol{x}}, \mathcal{X}^*\right)$$

$$+ 4\eta^2\left(B\sum_{s\in\mathcal{S}}\max_{(a,b)\in\mathcal{A}\times\mathcal{B}}\left|\boldsymbol{Q}_s(a,b) - \boldsymbol{Q}_s^*(a,b)\right|^2 + \frac{B^2}{(1-\gamma)^2}\mathrm{dist}^2\left(\boldsymbol{y}, \mathcal{Y}^*\right)\right).$$

Analogously,

$$\left\|\boldsymbol{y}^+ - \widetilde{\boldsymbol{y}}\right\|^2 \leq 8\mathrm{dist}^2\left(\widetilde{\boldsymbol{y}}, \mathcal{Y}^*\right)$$

$$+ 4\eta^2\left(A\sum_{s\in\mathcal{S}}\max_{(a,b)\in\mathcal{A}\times\mathcal{B}}\left|\widehat{\boldsymbol{Q}}_s(a,b) - \boldsymbol{Q}_s^*(a,b)\right|^2 + \frac{A^2}{(1-\gamma)^2}\mathrm{dist}^2\left(\boldsymbol{x}, \mathcal{X}^*\right)\right).$$

Then, the result follows by summing up the bound for $\left\|\boldsymbol{x}^+ - \widetilde{\boldsymbol{x}}\right\|^2$ and $\left\|\boldsymbol{y}^+ - \widetilde{\boldsymbol{y}}\right\|^2$. $\quad\square$

## C PROOF FOR LOCAL LINEAR CONVERGENCE OF OGDA

In this section, we prove the local linear convergence of OGDA (Theorem 5).

For notational simplicity, we assume $T_1 = 0$ in the analysis below. Recall the OGDA algorithm ($T_1 = 0$): the min-player and max-player initialize

$$\widetilde{\boldsymbol{x}}^0 = \boldsymbol{x}^0 = \widehat{\boldsymbol{x}}, \ \widetilde{\boldsymbol{y}}^0 = \boldsymbol{y}^0 = \widehat{\boldsymbol{y}}. \tag{22}$$

and the min-player updates for $t \geq 1$ as follows

$$\boldsymbol{x}_s^t = \mathcal{P}_{\Delta_{\mathcal{A}}} \left( \widetilde{\boldsymbol{x}}_s^{t-1} - \eta \boldsymbol{Q}_s^{t-1} \boldsymbol{y}_s^{t-1} \right), \tag{23a}$$

$$\widetilde{\boldsymbol{x}}_s^t = \mathcal{P}_{\Delta_{\mathcal{A}}} \left( \widetilde{\boldsymbol{x}}_s^{t-1} - \eta \boldsymbol{Q}_s^t \boldsymbol{y}_s^t \right), \tag{23b}$$

while the max-player updates for $t \geq 1$ as follows

$$\boldsymbol{y}_s^t = \mathcal{P}_{\Delta_{\mathcal{A}}} \left( \widetilde{\boldsymbol{y}}_s^{t-1} + \eta \left( \boldsymbol{Q}_s^{t-1} \right)^\top \boldsymbol{x}_s^{t-1} \right), \tag{24a}$$

$$\widetilde{\boldsymbol{y}}_s^t = \mathcal{P}_{\Delta_{\mathcal{A}}} \left( \widetilde{\boldsymbol{y}}_s^{t-1} + \eta \left( \boldsymbol{Q}_s^t \right)^\top \boldsymbol{x}_s^t \right). \tag{24b}$$

Here, we denote

$$\boldsymbol{Q}_s^t = \boldsymbol{Q}_s^{\boldsymbol{x}^t, \boldsymbol{y}^t}, \ \forall t \geq 0.$$

The policy $\boldsymbol{x}^t$ and $\boldsymbol{y}^t$ are played by the min-player and the max-player at iteration $t$. And $\widetilde{\boldsymbol{x}}^t, \widetilde{\boldsymbol{y}}^t$ are local auxiliary variables to help generate the policies $\boldsymbol{x}^t$ and $\boldsymbol{y}^t$.

Since we initialize $\boldsymbol{x}^0 = \widehat{\boldsymbol{x}}, \boldsymbol{y}^0 = \widehat{\boldsymbol{y}}$, we drop the notation of $\widehat{\boldsymbol{x}}, \widehat{\boldsymbol{y}}$ below and directly use $\boldsymbol{x}^0, \boldsymbol{y}^0$ to denote the initial policies.

The OGDA method is a decentralized algorithm, its decentralized implementation is in Algorithm 2 (for the min-player) and Algorithm 3 (for the max-player) of Appendix G.

To prove the local linear convergence of OGDA, we first introduce some notations and auxiliary variables.

**Additional notations and auxiliary variables.** We use $\boldsymbol{1}, \boldsymbol{0}$ to denote the all-ones and all-zeros vectors or matrices, whose dimensions are determined from the context. $\boldsymbol{1}_i$ is the $i$-th standard basis of the Euclidean space, i.e., the $i$-th entry of $\boldsymbol{1}_i$ equals one, and the others entries equal zero. The operators $>, \geq, <, \leq$ are overloaded for vectors and matrices in entry-wise sense.

We denote the policy pairs $\boldsymbol{z}^t = (\boldsymbol{x}^t, \boldsymbol{y}^t), \widetilde{\boldsymbol{z}}^t = \left( \widetilde{\boldsymbol{x}}^t, \widetilde{\boldsymbol{y}}^t \right)$ and denote the projections onto the Nash equilibrium sets as $\widetilde{\boldsymbol{x}}_s^{t*} = \mathcal{P}_{\mathcal{X}_s^*} (\boldsymbol{x}_s^t), \ \widetilde{\boldsymbol{y}}_s^{t*} = \mathcal{P}_{\mathcal{Y}_s^*} (\boldsymbol{y}_s^t), \ \widetilde{\boldsymbol{z}}_s^{t*} = \mathcal{P}_{\mathcal{Z}_s^*} (\boldsymbol{z}_s^t)$. Since $\widetilde{\boldsymbol{x}}^t, \widetilde{\boldsymbol{y}}^t, \widetilde{\boldsymbol{z}}^t$ are treated as concatenated vectors, we have from the elementary property of the $\ell_2$-norm that $\widetilde{\boldsymbol{z}}_s^{t*} = (\widetilde{\boldsymbol{x}}_s^{t*}, \widetilde{\boldsymbol{y}}_s^{t*}), \widetilde{\boldsymbol{x}}^{t*} = \mathcal{P}_{\mathcal{X}^*} \left( \widetilde{\boldsymbol{x}}^t \right) = \left\{ \widetilde{\boldsymbol{x}}_s^{t*} \right\}_{s \in \mathcal{S}}, \widetilde{\boldsymbol{y}}^{t*} = \mathcal{P}_{\mathcal{Y}^*} \left( \widetilde{\boldsymbol{y}}^t \right) = \left\{ \widetilde{\boldsymbol{y}}_s^{t*} \right\}_{s \in \mathcal{S}}, \widetilde{\boldsymbol{z}}^{t*} = \mathcal{P}_{\mathcal{Z}^*} \left( \widetilde{\boldsymbol{z}}^t \right) = \left\{ \widetilde{\boldsymbol{z}}_s^{t*} \right\}_{s \in \mathcal{S}}$, and $\widetilde{\boldsymbol{z}}^{t*} = (\widetilde{\boldsymbol{x}}^{t*}, \widetilde{\boldsymbol{y}}^{t*})$.

Let $\boldsymbol{\rho}_0$ be the uniform distribution on $\mathcal{S}$. Then, we denote the state visitation distribution under the policy pairs $(\widetilde{\boldsymbol{x}}^{(t-1)*}, \boldsymbol{y}^t)$ and $(\boldsymbol{x}^t, \widetilde{\boldsymbol{y}}^{(t-1)*})$ as

$$\boldsymbol{d}_x^t(s) = \boldsymbol{d}_{\boldsymbol{\rho}_0}^{\widetilde{\boldsymbol{x}}^{(t-1)*}, \boldsymbol{y}^t}(s), \ \boldsymbol{d}_y^t(s) = \boldsymbol{d}_{\boldsymbol{\rho}_0}^{\boldsymbol{x}^t, \widetilde{\boldsymbol{y}}^{(t-1)*}}(s). \tag{25}$$

It follows by definition that for any $s \in \mathcal{S}$,

$$\frac{1 - \gamma}{S} \leq \boldsymbol{d}_x^t(s) \leq 1, \ \frac{1 - \gamma}{S} \leq \boldsymbol{d}_y^t(s) \leq 1.$$

Define weighted sums of distances

$$\Theta^t = \sum_{s \in \mathcal{S}} \boldsymbol{d}_x^t(s) \text{dist}^2 \left( \widetilde{\boldsymbol{x}}_s^t, \mathcal{X}_s^* \right) + \boldsymbol{d}_y^t(s) \text{dist}^2 \left( \widetilde{\boldsymbol{y}}_s^t, \mathcal{Y}_s^* \right),$$

$$\widetilde{\Theta}^t = \sum_{s \in \mathcal{S}} \boldsymbol{d}_x^{t+1}(s) \text{dist}^2 \left( \widetilde{\boldsymbol{x}}_s^t, \mathcal{X}_s^* \right) + \boldsymbol{d}_y^{t+1}(s) \text{dist}^2 \left( \widetilde{\boldsymbol{y}}_s^t, \mathcal{Y}_s^* \right), \tag{26}$$

and potential functions

$$
\begin{aligned}
\Lambda^0 &= \operatorname{dist}^2\left(\boldsymbol{z}^0, \mathcal{Z}^*\right) = \operatorname{dist}^2(\widehat{\boldsymbol{z}}, \mathcal{Z}^*), \\
\Lambda^t &= \Theta^t + \frac{1-\gamma}{4S}\left\|\widetilde{\boldsymbol{z}}^t - \boldsymbol{z}^t\right\|^2, \ t \geq 1.
\end{aligned}
\tag{27}
$$

We will show the linear convergence of $\Lambda^t$ given $\operatorname{dist}^2\left(\boldsymbol{z}^0, \mathcal{Z}^*\right) \leq \delta_0 \eta^4$ for some problem-dependent constant $\delta_0 > 0$.

## C.1 ONE-STEP ANALYSIS

Our proof for local linear convergence starts from the following elementary lemma, which is derived by combining a standard analysis of optimistic gradient descent/ascent with the smoothness of $\boldsymbol{Q}_s^{\boldsymbol{x},\boldsymbol{y}}$ with respect to the policy pair $(\boldsymbol{x}, \boldsymbol{y})$.

**Lemma 3** *Let* $\left\{\boldsymbol{x}^t, \widetilde{\boldsymbol{x}}^t, \boldsymbol{y}^t, \widetilde{\boldsymbol{y}}^t\right\}$ *be generated from OGDA* (23), (24). *Then, for any* $t \geq 0$, *we have*

$$
\begin{aligned}
&\eta\left\langle \boldsymbol{x}_s^{t+1} - \widetilde{\boldsymbol{x}}_s^{t*}, \boldsymbol{Q}_s^{t+1} \boldsymbol{y}_s^{t+1}\right\rangle \\
&\leq \frac{1}{2}\left(\left\|\widetilde{\boldsymbol{x}}_s^t - \widetilde{\boldsymbol{x}}_s^{t*}\right\|^2 - \left\|\widetilde{\boldsymbol{x}}_s^{t+1} - \widetilde{\boldsymbol{x}}_s^{t*}\right\|^2\right) - \frac{1}{4}\left\|\widetilde{\boldsymbol{x}}_s^{t+1} - \boldsymbol{x}_s^{t+1}\right\|^2 - \frac{1}{2}\left\|\boldsymbol{x}_s^{t+1} - \widetilde{\boldsymbol{x}}_s^t\right\|^2 \\
&\quad + \frac{16A\left(A+B\right)\eta^2}{\left(1-\gamma\right)^4}\left\|\boldsymbol{z}^{t+1} - \boldsymbol{z}^t\right\|^2
\end{aligned}
\tag{28}
$$

*and*

$$
\begin{aligned}
&\eta\left\langle \widetilde{\boldsymbol{y}}_s^{t*} - \boldsymbol{y}_s^{t+1}, \left(\boldsymbol{Q}_s^{t+1}\right)^\top \boldsymbol{x}_s^{t+1}\right\rangle \\
&\leq \frac{1}{2}\left(\left\|\widetilde{\boldsymbol{y}}_s^t - \widetilde{\boldsymbol{y}}_s^{t*}\right\|^2 - \left\|\widetilde{\boldsymbol{y}}_s^{t+1} - \widetilde{\boldsymbol{y}}_s^{t*}\right\|^2\right) - \frac{1}{4}\left\|\widetilde{\boldsymbol{y}}_s^{t+1} - \boldsymbol{y}_s^{t+1}\right\|^2 - \frac{1}{2}\left\|\boldsymbol{y}_s^{t+1} - \widetilde{\boldsymbol{y}}_s^t\right\|^2 \\
&\quad + \frac{16B\left(A+B\right)\eta^2}{\left(1-\gamma\right)^4}\left\|\boldsymbol{z}^{t+1} - \boldsymbol{z}^t\right\|^2.
\end{aligned}
\tag{29}
$$

*Proof of Lemma 3.* We abbreviate $\widetilde{\boldsymbol{x}}^{t*} = \boldsymbol{x}^*$, $\widetilde{\boldsymbol{x}}_s^{t*} = \boldsymbol{x}_s^*$ in this proof. By (23b), since $\widetilde{\boldsymbol{x}}_s^{t+1}$ is the projection onto $\Delta_{\mathcal{A}}$, we have

$$
\left\langle \boldsymbol{x}_s^* - \widetilde{\boldsymbol{x}}_s^{t+1}, \widetilde{\boldsymbol{x}}_s^{t+1} - \widetilde{\boldsymbol{x}}_s^t + \eta \boldsymbol{Q}_s^{t+1} \boldsymbol{y}_s^{t+1}\right\rangle \geq 0, \ \forall t \geq 0.
$$

Equivalently,

$$
\eta\left\langle \widetilde{\boldsymbol{x}}_s^{t+1} - \boldsymbol{x}_s^*, \boldsymbol{Q}_s^{t+1} \boldsymbol{y}_s^{t+1}\right\rangle \leq \frac{1}{2}\left(\left\|\widetilde{\boldsymbol{x}}_s^t - \boldsymbol{x}_s^*\right\|^2 - \left\|\widetilde{\boldsymbol{x}}_s^{t+1} - \boldsymbol{x}_s^*\right\|^2 - \left\|\widetilde{\boldsymbol{x}}_s^{t+1} - \widetilde{\boldsymbol{x}}_s^t\right\|^2\right).
$$

Similarly, from (23a),

$$
\left\langle \widetilde{\boldsymbol{x}}_s^{t+1} - \boldsymbol{x}_s^{t+1}, \boldsymbol{x}_s^{t+1} - \widetilde{\boldsymbol{x}}_s^t + \eta \boldsymbol{Q}_s^t \boldsymbol{y}_s^t\right\rangle \geq 0, \ \forall t \geq 0
$$

i.e.,

$$
\eta\left\langle \boldsymbol{x}_s^{t+1} - \widetilde{\boldsymbol{x}}_s^{t+1}, \boldsymbol{Q}_s^t \boldsymbol{y}_s^t\right\rangle \leq \frac{1}{2}\left(\left\|\widetilde{\boldsymbol{x}}_s^{t+1} - \widetilde{\boldsymbol{x}}_s^t\right\|^2 - \left\|\widetilde{\boldsymbol{x}}_s^{t+1} - \boldsymbol{x}_s^{t+1}\right\|^2 - \left\|\boldsymbol{x}_s^{t+1} - \widetilde{\boldsymbol{x}}_s^t\right\|^2\right).
$$

Then, we have

$$
\begin{aligned}
&\eta \left\langle \boldsymbol{x}_s^{t+1} - \boldsymbol{x}_s^*, \boldsymbol{Q}_s^{t+1} \boldsymbol{y}_s^{t+1} \right\rangle \\
=&\eta \left\langle \widetilde{\boldsymbol{x}}_s^{t+1} - \boldsymbol{x}_s^*, \boldsymbol{Q}_s^{t+1} \boldsymbol{y}_s^{t+1} \right\rangle + \eta \left\langle \boldsymbol{x}_s^{t+1} - \widetilde{\boldsymbol{x}}_s^{t+1}, \boldsymbol{Q}_s^t \boldsymbol{y}_s^t \right\rangle \\
&+ \eta \left\langle \boldsymbol{x}_s^{t+1} - \widetilde{\boldsymbol{x}}_s^{t+1}, \boldsymbol{Q}_s^{t+1} \boldsymbol{y}_s^{t+1} - \boldsymbol{Q}_s^t \boldsymbol{y}_s^t \right\rangle \\
\leq&\frac{1}{2} \left( \left\| \widetilde{\boldsymbol{x}}_s^t - \boldsymbol{x}_s^* \right\|^2 - \left\| \widetilde{\boldsymbol{x}}_s^{t+1} - \boldsymbol{x}_s^* \right\|^2 - \left\| \widetilde{\boldsymbol{x}}_s^{t+1} - \boldsymbol{x}_s^{t+1} \right\|^2 - \left\| \boldsymbol{x}_s^{t+1} - \widetilde{\boldsymbol{x}}_s^t \right\|^2 \right) \\
&+ \eta \left\langle \boldsymbol{x}_s^{t+1} - \widetilde{\boldsymbol{x}}_s^{t+1}, \boldsymbol{Q}_s^{t+1} \boldsymbol{y}_s^{t+1} - \boldsymbol{Q}_s^t \boldsymbol{y}_s^t \right\rangle \\
\leq&\frac{1}{2} \left( \left\| \widetilde{\boldsymbol{x}}_s^t - \boldsymbol{x}_s^* \right\|^2 - \left\| \widetilde{\boldsymbol{x}}_s^{t+1} - \boldsymbol{x}_s^* \right\|^2 \right) - \frac{1}{4} \left\| \widetilde{\boldsymbol{x}}_s^{t+1} - \boldsymbol{x}_s^{t+1} \right\|^2 - \frac{1}{2} \left\| \boldsymbol{x}_s^{t+1} - \widetilde{\boldsymbol{x}}_s^t \right\|^2 \\
&+ 4\eta^2 A \left\| \boldsymbol{Q}_s^{t+1} \boldsymbol{y}_s^{t+1} - \boldsymbol{Q}_s^t \boldsymbol{y}_s^t \right\|_\infty^2 .
\end{aligned}
\tag{30}
$$

By Lemma 21, we have

$$
\begin{aligned}
&\left\| \boldsymbol{Q}_s^{t+1} \boldsymbol{y}_s^{t+1} - \boldsymbol{Q}_s^t \boldsymbol{y}_s^t \right\|_\infty \\
\leq& \max_{(a,b)\in\mathcal{A}\times\mathcal{B}} \left| \boldsymbol{Q}_s^t(a,b) - \boldsymbol{Q}_s^{t+1}(a,b) \right| \left\| \boldsymbol{y}_s^{t+1} \right\|_1 + \max_{(a,b)\in\mathcal{A}\times\mathcal{B}} \left| \boldsymbol{Q}_s^t(a,b) \right| \left\| \boldsymbol{y}_s^{t+1} - \boldsymbol{y}_s^t \right\|_1 \\
\leq& \frac{\sqrt{A+B} \left\| \boldsymbol{z}^{t+1} - \boldsymbol{z}^t \right\|}{(1-\gamma)^2} + \frac{\sqrt{B} \left\| \boldsymbol{y}_s^{t+1} - \boldsymbol{y}_s^t \right\|}{1-\gamma} \leq \frac{2\sqrt{A+B}}{(1-\gamma)^2} \left\| \boldsymbol{z}^{t+1} - \boldsymbol{z}^t \right\| .
\end{aligned}
\tag{31}
$$

Then, (28) follows by combining (30) with (31). And (29) follows by similar arguments. □

We consider weighted sum of (28) and (29) using the state visitation distribution $\boldsymbol{d}_x^t(s)$, $\boldsymbol{d}_y^t(s)$ defined in (25) as the weighting coefficients.

**Lemma 4** (One-Step Analysis) Let $\left\{ \boldsymbol{x}^t, \widetilde{\boldsymbol{x}}^t, \boldsymbol{y}^t, \widetilde{\boldsymbol{y}}^t \right\}$ be generated from OGDA with $\eta \leq \frac{(1-\gamma)^{\frac{5}{2}}}{32\sqrt{S}(A+B)}$. Then, for any $t \geq 0$,

$$
\begin{aligned}
&\Theta^{t+1} + \frac{1-\gamma}{4S} \left\| \widetilde{\boldsymbol{z}}^{t+1} - \boldsymbol{z}^{t+1} \right\|^2 \\
\leq& \widetilde{\Theta}^t + \frac{1-\gamma}{8S} \left\| \widetilde{\boldsymbol{z}}^t - \boldsymbol{z}^t \right\|^2 - \frac{1-\gamma}{4S} \left( \left\| \widetilde{\boldsymbol{z}}^{t+1} - \boldsymbol{z}^{t+1} \right\|^2 + \left\| \boldsymbol{z}^{t+1} - \widetilde{\boldsymbol{z}}^t \right\|^2 \right) .
\end{aligned}
\tag{32}
$$

*Proof of Lemma 4.* Recall that $\boldsymbol{\rho}_0$ be the uniform distribution on $\mathcal{S}$. By Lemma 20,

$$
\begin{aligned}
&V^{\boldsymbol{x}^{t+1}, \widetilde{\boldsymbol{y}}^{t*}}(\boldsymbol{\rho}_0) - V^{\widetilde{\boldsymbol{x}}^{t*}, \boldsymbol{y}^{t+1}}(\boldsymbol{\rho}_0) \\
=&V^{\boldsymbol{x}^{t+1}, \widetilde{\boldsymbol{y}}^{t*}}(\boldsymbol{\rho}_0) - V^{\boldsymbol{x}^{t+1}, \boldsymbol{y}^{t+1}}(\boldsymbol{\rho}_0) + V^{\boldsymbol{x}^{t+1}, \boldsymbol{y}^{t+1}}(\boldsymbol{\rho}_0) - V^{\widetilde{\boldsymbol{x}}^{t*}, \boldsymbol{y}^{t+1}}(\boldsymbol{\rho}_0) \\
=&\frac{1}{1-\gamma} \sum_{s\in\mathcal{S}} \left( \boldsymbol{d}_x^{t+1}(s) \left\langle \boldsymbol{x}_s^{t+1} - \widetilde{\boldsymbol{x}}_s^{t*}, \boldsymbol{Q}_s^{t+1} \boldsymbol{y}_s^{t+1} \right\rangle - \boldsymbol{d}_y^{t+1}(s) \left\langle \boldsymbol{y}_s^{t+1} - \widetilde{\boldsymbol{y}}_s^{t*}, \left( \boldsymbol{Q}_s^{t+1} \right)^\top \boldsymbol{x}_s^{t+1} \right\rangle \right) .
\end{aligned}
\tag{33}
$$

As $\widetilde{\boldsymbol{x}}^{t*} \in \mathcal{X}^*$, $\widetilde{\boldsymbol{y}}^{t*} \in \mathcal{Y}^*$, by Lemma 19, $\left( \widetilde{\boldsymbol{x}}^{t*}, \widetilde{\boldsymbol{y}}^{t*} \right)$ also attains Nash equilibrium. Thus, we have

$$
\begin{aligned}
&V^{\boldsymbol{x}^{t+1}, \widetilde{\boldsymbol{y}}^{t*}}(\boldsymbol{\rho}_0) - V^{\widetilde{\boldsymbol{x}}^{t*}, \boldsymbol{y}^{t+1}}(\boldsymbol{\rho}_0) \\
=&V^{\boldsymbol{x}^{t+1}, \widetilde{\boldsymbol{y}}^{t*}}(\boldsymbol{\rho}_0) - V^{\widetilde{\boldsymbol{x}}^{t*}, \widetilde{\boldsymbol{y}}^{t*}}(\boldsymbol{\rho}_0) + V^{\widetilde{\boldsymbol{x}}^{t*}, \widetilde{\boldsymbol{y}}^{t*}}(\boldsymbol{\rho}_0) - V^{\widetilde{\boldsymbol{x}}^{t*}, \boldsymbol{y}^{t+1}}(\boldsymbol{\rho}_0) \geq 0.
\end{aligned}
\tag{34}
$$

Substituting (28), (29) into (33) yields that

$$\eta(1 - \gamma)\left(V^{\boldsymbol{x}^{t+1}, \widetilde{\boldsymbol{y}}^{t*}}(\boldsymbol{\rho}_0) - V^{\widetilde{\boldsymbol{x}}^{t*}, \boldsymbol{y}^{t+1}}(\boldsymbol{\rho}_0)\right)$$

$$\leq \frac{1}{2} \sum_{s \in \mathcal{S}} \left(\boldsymbol{d}_x^{t+1}(s)\left\|\widetilde{\boldsymbol{x}}_s^t - \widetilde{\boldsymbol{x}}_s^{t*}\right\|^2 + \boldsymbol{d}_y^{t+1}(s)\left\|\widetilde{\boldsymbol{y}}_s^t - \widetilde{\boldsymbol{y}}_s^{t*}\right\|^2\right)$$

$$- \frac{1}{2} \sum_{s \in \mathcal{S}} \left(\boldsymbol{d}_x^{t+1}(s)\left\|\widetilde{\boldsymbol{x}}_s^{t+1} - \widetilde{\boldsymbol{x}}_s^{t*}\right\|^2 + \boldsymbol{d}_y^{t+1}(s)\left\|\widetilde{\boldsymbol{y}}_s^{t+1} - \widetilde{\boldsymbol{y}}_s^{t*}\right\|^2\right)$$

$$- \frac{1}{4} \sum_{s \in \mathcal{S}} \left(\boldsymbol{d}_x^{t+1}(s)\left\|\widetilde{\boldsymbol{x}}_s^{t+1} - \boldsymbol{x}_s^{t+1}\right\|^2 + \boldsymbol{d}_y^{t+1}(s)\left\|\widetilde{\boldsymbol{y}}_s^{t+1} - \boldsymbol{y}_s^{t+1}\right\|^2\right)$$

$$- \frac{1}{2} \sum_{s \in \mathcal{S}} \left(\boldsymbol{d}_x^{t+1}(s)\left\|\boldsymbol{x}_s^{t+1} - \widetilde{\boldsymbol{x}}_s^t\right\|^2 + \boldsymbol{d}_y^{t+1}(s)\left\|\boldsymbol{y}_s^{t+1} - \widetilde{\boldsymbol{y}}_s^t\right\|^2\right)$$

$$+ \frac{16(A+B)\eta^2}{(1-\gamma)^4} \sum_{s \in \mathcal{S}} \left(\boldsymbol{d}_x^{t+1}(s) + \boldsymbol{d}_y^{t+1}(s)\right)\left\|\boldsymbol{z}^{t+1} - \boldsymbol{z}^t\right\|^2$$

By combining with the facts that $\left\|\widetilde{\boldsymbol{x}}_s^{t+1} - \widetilde{\boldsymbol{x}}_s^{t*}\right\| \geq \operatorname{dist}\left(\widetilde{\boldsymbol{x}}_s^{t+1}, \mathcal{X}_s^*\right)$, $\boldsymbol{d}_x^t(s) \geq \frac{1-\gamma}{S}$, $\sum_{s \in \mathcal{S}} \boldsymbol{d}_x^{t+1}(s) = 1$ and their counterparts for the max-player, we have

$$\eta(1 - \gamma)\left(V^{\boldsymbol{x}^{t+1}, \widetilde{\boldsymbol{y}}^{t*}}(\boldsymbol{\rho}_0) - V^{\widetilde{\boldsymbol{x}}^{t*}, \boldsymbol{y}^{t+1}}(\boldsymbol{\rho}_0)\right)$$

$$\leq \frac{1}{2}\widetilde{\Theta}^t - \frac{1}{2}\Theta^{t+1} - \frac{1-\gamma}{4S}\left\|\widetilde{\boldsymbol{z}}^{t+1} - \boldsymbol{z}^{t+1}\right\|^2 - \frac{1-\gamma}{2S}\left\|\boldsymbol{z}^{t+1} - \widetilde{\boldsymbol{z}}^t\right\|^2$$

$$+ \frac{64(A+B)\eta^2}{(1-\gamma)^4}\left(\left\|\boldsymbol{z}^{t+1} - \widetilde{\boldsymbol{z}}^t\right\|^2 + \left\|\widetilde{\boldsymbol{z}}^t - \boldsymbol{z}^t\right\|^2\right) \tag{35}$$

$$\leq \frac{1}{2}\widetilde{\Theta}^t - \frac{1}{2}\Theta^{t+1} - \frac{1-\gamma}{4S}\left\|\widetilde{\boldsymbol{z}}^{t+1} - \boldsymbol{z}^{t+1}\right\|^2 - \frac{1-\gamma}{8S}\left\|\boldsymbol{z}^{t+1} - \widetilde{\boldsymbol{z}}^t\right\|^2 + \frac{1-\gamma}{16S}\left\|\widetilde{\boldsymbol{z}}^t - \boldsymbol{z}^t\right\|^2,$$

where the last inequality is by our condition on $\eta$.

By combining (34) with (35) and rearranging, we have

$$\Theta^{t+1} + \frac{1-\gamma}{4S}\left\|\widetilde{\boldsymbol{z}}^{t+1} - \boldsymbol{z}^{t+1}\right\|^2$$

$$\leq \widetilde{\Theta}^t + \frac{1-\gamma}{8S}\left\|\widetilde{\boldsymbol{z}}^t - \boldsymbol{z}^t\right\|^2 - \frac{1-\gamma}{4S}\left(\left\|\widetilde{\boldsymbol{z}}^{t+1} - \boldsymbol{z}^{t+1}\right\|^2 + \left\|\boldsymbol{z}^{t+1} - \widetilde{\boldsymbol{z}}^t\right\|^2\right).$$

$\square$

## C.2 PROGRESS OF PROJECTED GRADIENT DESCENT

The following lemma is a standard step in the analysis of projected gradient descent.

**Lemma 5** *If* $\eta \leq \frac{1-\gamma}{\max\{\sqrt{A}, \sqrt{B}\}}$*, for any* $t \geq 0$*, let* $\boldsymbol{\rho}_0$ *be the uniform distribution on* $\mathcal{S}$*, then*

$$\eta^2 \sum_{s \in \mathcal{S}} \left(V^{\widetilde{\boldsymbol{x}}^t, \dagger}(s) - V^{\dagger, \widetilde{\boldsymbol{y}}^t}(s)\right)^2 \leq \frac{36S}{(1-\gamma)^2}\left(\left\|\widetilde{\boldsymbol{z}}^{t+1} - \boldsymbol{z}^{t+1}\right\|^2 + \left\|\boldsymbol{z}^{t+1} - \widetilde{\boldsymbol{z}}^t\right\|^2\right).$$

*Proof of Lemma 5.* Since $\widetilde{\boldsymbol{x}}_s^{t+1}$ is a projection onto $\Delta_\mathcal{A}$, for any $\boldsymbol{x}_s' \in \Delta_\mathcal{A}$,

$$\left\langle \widetilde{\boldsymbol{x}}_s^{t+1} - \widetilde{\boldsymbol{x}}_s^t + \eta \boldsymbol{Q}_s^{t+1}\boldsymbol{y}_s^{t+1}, \boldsymbol{x}_s' - \widetilde{\boldsymbol{x}}_s^{t+1}\right\rangle \geq 0,$$

i.e.,

$$\eta\left\langle \widetilde{\boldsymbol{x}}_s^{t+1} - \boldsymbol{x}_s', \boldsymbol{Q}_s^{t+1}\boldsymbol{y}_s^{t+1}\right\rangle \leq \left\langle \widetilde{\boldsymbol{x}}_s^{t+1} - \widetilde{\boldsymbol{x}}_s^t, \boldsymbol{x}_s' - \widetilde{\boldsymbol{x}}_s^{t+1}\right\rangle.$$

Then, by combining with the condition on $\eta$,

$$\eta \left\langle \boldsymbol{x}_s^{t+1} - \boldsymbol{x}_s', \boldsymbol{Q}_s^{t+1} \boldsymbol{y}_s^{t+1} \right\rangle \leq \eta \left\langle \widetilde{\boldsymbol{x}}_s^{t+1} - \boldsymbol{x}_s', \boldsymbol{Q}_s^{t+1} \boldsymbol{y}_s^{t+1} \right\rangle + \eta \left\| \boldsymbol{x}_s^{t+1} - \widetilde{\boldsymbol{x}}_s^{t+1} \right\| \left\| \boldsymbol{Q}_s^{t+1} \boldsymbol{y}_s^{t+1} \right\|$$

$$\leq \left\langle \widetilde{\boldsymbol{x}}_s^{t+1} - \widetilde{\boldsymbol{x}}_s^t, \boldsymbol{x}_s' - \widetilde{\boldsymbol{x}}_s^{t+1} \right\rangle + \frac{\eta \sqrt{A}}{1-\gamma} \left\| \boldsymbol{x}_s^{t+1} - \widetilde{\boldsymbol{x}}_s^{t+1} \right\|$$

$$\leq 2 \left\| \widetilde{\boldsymbol{x}}_s^{t+1} - \widetilde{\boldsymbol{x}}_s^t \right\| + \left\| \boldsymbol{x}_s^{t+1} - \widetilde{\boldsymbol{x}}_s^{t+1} \right\|$$

$$\leq 2 \left\| \boldsymbol{x}_s^{t+1} - \widetilde{\boldsymbol{x}}_s^t \right\| + 3 \left\| \boldsymbol{x}_s^{t+1} - \widetilde{\boldsymbol{x}}_s^{t+1} \right\|.$$

For any $s_0 \in \mathcal{S}$ and $\boldsymbol{x}' \in \mathcal{X}$, by Lemma 20 and the fact that $\sum_{s \in \mathcal{S}} \boldsymbol{d}_{s_0}^{\boldsymbol{x}', \boldsymbol{y}^{t+1}}(s) = 1$,

$$\eta \left( V^{\boldsymbol{x}^{t+1}, \boldsymbol{y}^{t+1}}(s_0) - V^{\boldsymbol{x}', \boldsymbol{y}^{t+1}}(s_0) \right) = \frac{\eta}{1-\gamma} \sum_{s \in \mathcal{S}} \boldsymbol{d}_{s_0}^{\boldsymbol{x}', \boldsymbol{y}^{t+1}}(s) \left\langle \boldsymbol{x}_s^{t+1} - \boldsymbol{x}_s', \boldsymbol{Q}_s^{t+1} \boldsymbol{y}_s^{t+1} \right\rangle$$

$$\leq \frac{\eta}{1-\gamma} \sum_{s \in \mathcal{S}} \boldsymbol{d}_{s_0}^{\boldsymbol{x}', \boldsymbol{y}^{t+1}}(s) \left( \sup_{\boldsymbol{x}_s'' \in \Delta_{\mathcal{A}}} \left\langle \boldsymbol{x}_s^{t+1} - \boldsymbol{x}_s'', \boldsymbol{Q}_s^{t+1} \boldsymbol{y}_s^{t+1} \right\rangle \right)$$

$$\leq \frac{1}{1-\gamma} \max_{s \in \mathcal{S}} \left( 2 \left\| \boldsymbol{x}_s^{t+1} - \widetilde{\boldsymbol{x}}_s^t \right\| + 3 \left\| \boldsymbol{x}_s^{t+1} - \widetilde{\boldsymbol{x}}_s^{t+1} \right\| \right)$$

$$\leq \frac{1}{1-\gamma} \left( 2 \left\| \boldsymbol{x}^{t+1} - \widetilde{\boldsymbol{x}}^t \right\| + 3 \left\| \boldsymbol{x}^{t+1} - \widetilde{\boldsymbol{x}}^{t+1} \right\| \right),$$

i.e.,

$$\eta \left( V^{\boldsymbol{x}^{t+1}, \boldsymbol{y}^{t+1}}(s_0) - V^{\boldsymbol{x}', \boldsymbol{y}^{t+1}}(s_0) \right) \leq \frac{1}{1-\gamma} \left( 2 \left\| \boldsymbol{x}^{t+1} - \widetilde{\boldsymbol{x}}^t \right\| + 3 \left\| \boldsymbol{x}^{t+1} - \widetilde{\boldsymbol{x}}^{t+1} \right\| \right). \qquad (36)$$

Similarly,

$$\eta \left( V^{\boldsymbol{x}^{t+1}, \dagger}(s_0) - V^{\boldsymbol{x}^{t+1}, \boldsymbol{y}^{t+1}}(s_0) \right) \leq \frac{1}{1-\gamma} \left( 2 \left\| \boldsymbol{y}^{t+1} - \widetilde{\boldsymbol{y}}^t \right\| + 3 \left\| \boldsymbol{y}^{t+1} - \widetilde{\boldsymbol{y}}^{t+1} \right\| \right). \qquad (37)$$

By (93) and (94), we have

$$\begin{aligned} \left| V^{\boldsymbol{x}^{t+1}, \dagger}(s_0) - V^{\widetilde{\boldsymbol{x}}^t, \dagger}(s_0) \right| &\leq \frac{\sqrt{A}}{(1-\gamma)^2} \left\| \boldsymbol{x}^{t+1} - \widetilde{\boldsymbol{x}}^t \right\|, \\ \left| V^{\dagger, \boldsymbol{y}^{t+1}}(s_0) - V^{\dagger, \widetilde{\boldsymbol{y}}^t}(s_0) \right| &\leq \frac{\sqrt{B}}{(1-\gamma)^2} \left\| \boldsymbol{y}^{t+1} - \widetilde{\boldsymbol{y}}^t \right\|. \end{aligned} \qquad (38)$$

Then, by combining (36), (37), (38) and the condition on $\eta$, we have

$$\eta^2 \left( V^{\widetilde{\boldsymbol{x}}^t, \dagger}(s_0) - V^{\dagger, \widetilde{\boldsymbol{y}}^t}(s_0) \right)^2 \leq \frac{36}{(1-\gamma)^2} \left( \left\| \widetilde{\boldsymbol{z}}^{t+1} - \boldsymbol{z}^{t+1} \right\|^2 + \left\| \boldsymbol{z}^{t+1} - \widetilde{\boldsymbol{z}}^t \right\|^2 \right).$$

The result follows by taking sum over $s_0 \in \mathcal{S}$. $\qquad \square$

Next, we extend Lemma 4 of Gilpin et al. (2012) and Theorem 5 of Wei et al. (2020) from matrix games to Markov games. Firstly, we prove the following auxiliary lemma, which is used in the proof of Lemma 7. This lemma is straightforward from the contraction and monotonicity of the Bellman operator, we attach its proof for completeness.

**Lemma 6** *For policies $\boldsymbol{x} \in \mathcal{X}$ and $\boldsymbol{y} \in \mathcal{Y}$, if there is a vector $\boldsymbol{v} \in \mathbb{R}^S$ such that for any $s \in \mathcal{S}$ $\langle \boldsymbol{x}_s, \boldsymbol{Q}_s[\boldsymbol{v}] \boldsymbol{y}_s \rangle \geq \boldsymbol{v}(s)$, then, we have that for any $s \in \mathcal{S}$,*

$$V^{\boldsymbol{x}, \boldsymbol{y}}(s) \geq \boldsymbol{v}(s).$$

*Proof of Lemma 6.* For any vector $\boldsymbol{u} \in \mathbb{R}^S$, define the mapping $\Phi : \mathbb{R}^S \to \mathbb{R}^S$ with

$$\Phi[\boldsymbol{u}](s) = \langle \boldsymbol{x}_s, \boldsymbol{Q}_s[\boldsymbol{u}] \boldsymbol{y}_s \rangle.$$

Then, for any $\boldsymbol{u}_1, \boldsymbol{u}_2 \in \mathbb{R}^S$, by definition,

$$|\Phi[\boldsymbol{u}_1](s) - \Phi[\boldsymbol{u}_2](s)| \leq \gamma \sum_{s' \in \mathcal{S}} \sum_{(a,b) \in \mathcal{A} \times \mathcal{B}} \mathbb{P}(s'|s, a, b) \boldsymbol{x}_s(a) \boldsymbol{y}_s(b) |\boldsymbol{u}_1(s') - \boldsymbol{u}_2(s')|$$
$$\leq \gamma \|\boldsymbol{u}_1 - \boldsymbol{u}_2\|_\infty .$$

Thus, we have

$$\|\Phi[\boldsymbol{u}_1] - \Phi[\boldsymbol{u}_2]\|_\infty \leq \gamma \|\boldsymbol{u}_1 - \boldsymbol{u}_2\|_\infty , \tag{39}$$

i.e., $\Phi$ is a contraction mapping.

Define $\boldsymbol{v}_1 = \Phi[\boldsymbol{v}]$ and $\boldsymbol{v}_{k+1} = \Phi[\boldsymbol{v}_k], \ldots$ Then, by (39), we have

$$\|\boldsymbol{v}_{k+1} - \boldsymbol{v}_k\|_\infty \leq \gamma \|\boldsymbol{v}_k - \boldsymbol{v}_{k-1}\|_\infty \leq \gamma^k \|\boldsymbol{v}_1 - \boldsymbol{v}\|_\infty .$$

Then, the limit of $\boldsymbol{v}_k$ exists and we denote the limit $\boldsymbol{v}_* = \lim_{k \to \infty} \boldsymbol{v}_k$. Obviously, $\boldsymbol{v}_*$ is a fixed point of $\Phi$ because

$$\boldsymbol{v}_* = \lim_{k \to \infty} \boldsymbol{v}_k = \lim_{k \to \infty} \Phi[\boldsymbol{v}_{k-1}] = \Phi[\lim_{k \to \infty} \boldsymbol{v}_{k-1}] = \Phi[\boldsymbol{v}_*].$$

As $V^{\boldsymbol{x},\boldsymbol{y}}(s) = \langle \boldsymbol{x}_s, \boldsymbol{Q}_s[V^{\boldsymbol{x},\boldsymbol{y}}] \boldsymbol{y}_s \rangle$, we have $\Phi[V^{\boldsymbol{x},\boldsymbol{y}}] = V^{\boldsymbol{x},\boldsymbol{y}}$, i.e., $V^{\boldsymbol{x},\boldsymbol{y}}$ is a fixed point of $\Phi$. By the contraction property of $\Phi$ as in (39), its fixed point is unique. Thus,

$$V^{\boldsymbol{x},\boldsymbol{y}} = \boldsymbol{v}_*.$$

By definition, for any $\boldsymbol{u}_1, \boldsymbol{u}_2 \in \mathbb{R}^S$, if $\boldsymbol{u}_1 \geq \boldsymbol{u}_2$ in entry-wise sense, then $\Phi[\boldsymbol{u}_1] \geq \Phi[\boldsymbol{u}_2]$ in entry-wise sense. Since the condition $\langle \boldsymbol{x}_s, \boldsymbol{Q}_s[\boldsymbol{v}] \boldsymbol{y}_s \rangle \geq \boldsymbol{v}(s)$ for any $s \in \mathcal{S}$ is equivalent to $\boldsymbol{v}_1 \geq \boldsymbol{v}$ in entry-wise sense. By induction, we have $\boldsymbol{v}_k(s)$ is non-decreasing in $k$. Combining with the fact that $\boldsymbol{v}_* = \lim_{k \to \infty} \boldsymbol{v}_k$, we have that for any $s \in \mathcal{S}$,

$$V^{\boldsymbol{x},\boldsymbol{y}}(s) = \boldsymbol{v}_*(s) \geq \boldsymbol{v}(s).$$

$\square$

The following lemma is an extension of Lemma 4 of Gilpin et al. (2012) and Theorem 5 of Wei et al. (2020) for matrix games to Markov games, it plays an important role in lower bounding the progress of gradient descent/ascent.

**Lemma 7** *There exists a problem-dependent constant $c_+ > 0$ such that for any $\boldsymbol{z} = (\boldsymbol{x}, \boldsymbol{y}) \in \mathcal{Z}$ and $s \in \mathcal{S}$,*

$$V^{\boldsymbol{x},\dagger}(s) - V^{\dagger,\boldsymbol{y}}(s) \geq c_+ \cdot \text{dist}(\boldsymbol{z}_s, \mathcal{Z}_s^*).$$

*Proof of Lemma 7.* Recall that $v^*(s)$ is the minimax game value at state $s$ and $\boldsymbol{Q}_s^* = \boldsymbol{Q}_s[v^*]$. For any $s \in \mathcal{S}$, choose

$$\widehat{\boldsymbol{y}}_s \in \arg\max_{\boldsymbol{y}_s' \in \Delta_\mathcal{B}} \langle \boldsymbol{x}_s, \boldsymbol{Q}_s^* \boldsymbol{y}_s' \rangle, \quad \widehat{\boldsymbol{x}}_s \in \arg\min_{\boldsymbol{x}_s' \in \Delta_\mathcal{A}} \langle \boldsymbol{x}_s', \boldsymbol{Q}_s^* \boldsymbol{y}_s \rangle.$$

Then, by Shapley's theorem (Lemma 19), $\mathcal{X}_s^* \times \mathcal{Y}_s^*$ is the NE set for the matrix game $\min_{\boldsymbol{x}'} \max_{\boldsymbol{y}'} {\boldsymbol{x}'}^\top \boldsymbol{Q}_s^* \boldsymbol{y}'$.

$$\langle \boldsymbol{x}_s, \boldsymbol{Q}_s^* \widehat{\boldsymbol{y}}_s \rangle \geq v^*(s), \quad \langle \widehat{\boldsymbol{x}}_s, \boldsymbol{Q}_s^* \boldsymbol{y}_s \rangle \leq v^*(s). \tag{40}$$

Then, by Lemma 22, for any $s \in \mathcal{S}$, there exists a constant $\varphi(\boldsymbol{Q}_s^*) > 0$ depending only on the matrix $\boldsymbol{Q}_s^*$ such that

$$\langle \boldsymbol{x}_s, \boldsymbol{Q}_s^* \widehat{\boldsymbol{y}}_s \rangle - \langle \widehat{\boldsymbol{x}}_s, \boldsymbol{Q}_s^* \boldsymbol{y}_s \rangle \geq \varphi(\boldsymbol{Q}_s^*) \cdot \text{dist}(\boldsymbol{z}_s, \mathcal{Z}_s^*). \tag{41}$$

Define the policies $\widehat{\boldsymbol{x}} = \{\widehat{\boldsymbol{x}}_s\}_{s \in \mathcal{S}}$ and $\widehat{\boldsymbol{y}} = \{\widehat{\boldsymbol{y}}_s\}_{s \in \mathcal{S}}$. Combining (40) with Lemma 6 yields that for any $s \in \mathcal{S}$,

$$V^{\boldsymbol{x},\widehat{\boldsymbol{y}}}(s) \geq v^*(s), \quad V^{\widehat{\boldsymbol{x}},\boldsymbol{y}}(s) \leq v^*(s).$$

Then, by definition, in entry-wise sense,

$$\boldsymbol{Q}_s^{\boldsymbol{x},\widehat{\boldsymbol{y}}} = \boldsymbol{Q}_s[V^{\boldsymbol{x},\widehat{\boldsymbol{y}}}] \geq \boldsymbol{Q}_s[v^*].$$

By combining the above equations, we have for any $s \in \mathcal{S}$,

$$
\begin{aligned}
V^{\boldsymbol{x},\dagger}(s) - V^{\dagger,\boldsymbol{y}}(s) &\geq V^{\boldsymbol{x},\widehat{\boldsymbol{y}}}(s) - V^{\widehat{\boldsymbol{x}},\boldsymbol{y}}(s) \\
&= \left\langle \boldsymbol{x}_s, \boldsymbol{Q}_s^{\boldsymbol{x},\widehat{\boldsymbol{y}}} \widehat{\boldsymbol{y}}_s \right\rangle - \left\langle \widehat{\boldsymbol{x}}_s, \boldsymbol{Q}_s^{\widehat{\boldsymbol{x}},\boldsymbol{y}} \boldsymbol{y}_s \right\rangle = \left\langle \boldsymbol{x}_s, \boldsymbol{Q}_s[V^{\boldsymbol{x},\widehat{\boldsymbol{y}}}]\widehat{\boldsymbol{y}}_s \right\rangle - \left\langle \widehat{\boldsymbol{x}}_s, \boldsymbol{Q}_s[V^{\widehat{\boldsymbol{x}},\boldsymbol{y}}]\boldsymbol{y}_s \right\rangle \\
&\geq \langle \boldsymbol{x}_s, \boldsymbol{Q}_s[v^*]\widehat{\boldsymbol{y}}_s \rangle - \langle \widehat{\boldsymbol{x}}_s, \boldsymbol{Q}_s[v^*]\boldsymbol{y}_s \rangle = \langle \boldsymbol{x}_s, \boldsymbol{Q}_s^* \widehat{\boldsymbol{y}}_s \rangle - \langle \widehat{\boldsymbol{x}}_s, \boldsymbol{Q}_s^* \boldsymbol{y}_s \rangle \\
&\geq \varphi(\boldsymbol{Q}_s^*) \cdot \mathrm{dist}(\boldsymbol{z}_s, \mathcal{Z}_s^*),
\end{aligned}
$$

where the second last inequality is by (40), the last inequality is by (41).

Then, the proof is completed by (1). $\qquad\square$

By combining Lemma 5 and Lemma 7, we provide lower bound for the progress of projected gradient descent (PGD).

**Lemma 8** *(Progress of PGD) Let $\left\{ \boldsymbol{z}^t, \widetilde{\boldsymbol{z}}^t \right\}_{t \geq 0}$ be generated from OGDA with $\eta \leq 1 - \gamma$, then for any $t \geq 0$, we have*

$$\left\| \widetilde{\boldsymbol{z}}^{t+1} - \boldsymbol{z}^{t+1} \right\|^2 + \left\| \boldsymbol{z}^{t+1} - \widetilde{\boldsymbol{z}}^t \right\|^2 \geq \frac{(1-\gamma)^2 \eta^2 c_+^2}{36S} \Theta^t.$$

*Proof of Lemma 8.* By Lemma 5 and Lemma 7, we have

$$
\begin{aligned}
\left\| \widetilde{\boldsymbol{z}}^{t+1} - \boldsymbol{z}^{t+1} \right\|^2 + \left\| \boldsymbol{z}^{t+1} - \widetilde{\boldsymbol{z}}^t \right\|^2 &\geq \eta^2 \sum_{s \in \mathcal{S}} \left( V^{\widetilde{\boldsymbol{x}}^t,\dagger}(s) - V^{\dagger,\widetilde{\boldsymbol{y}}^t}(s) \right)^2 \\
&\geq \frac{(1-\gamma)^2 \eta^2 c_+^2}{36S} \mathrm{dist}^2(\widetilde{\boldsymbol{z}}^t, \mathcal{Z}^*) \geq \frac{(1-\gamma)^2 \eta^2 c_+^2}{36S} \Theta^t,
\end{aligned}
$$

where the last inequality above comes from the fact that $\boldsymbol{d}_x^t(s) \leq 1$, $\boldsymbol{d}_y^t(s) \leq 1$ for any $s \in \mathcal{S}$. $\quad\square$

## C.3 STABILITY OF STATE VISITATION DISTRIBUTION NEAR THE NASH EQUILIBRIUM SET

The main motivation behind the proofs in this section is Lemma 1, which shows that projected gradient descent is very "stable" on the NE set.

The following lemma is a perturbed version of Lemma 1. It is extensively used in the proof of Lemma 10. Its proof follows by Lemma 2 and Lemma 21 with a simplification of coefficients.

**Lemma 9** *For any $\boldsymbol{z} = (\boldsymbol{x}, \boldsymbol{y}) \in \mathcal{Z}$ and $\widetilde{\boldsymbol{z}} = (\widetilde{\boldsymbol{x}}, \widetilde{\boldsymbol{y}}) \in \mathcal{Z}$, let $\boldsymbol{x}^+, \boldsymbol{y}^+$ be the policy after one step of projected policy gradient descent/ascent with sepsize $\eta > 0$, i.e., for $s \in \mathcal{S}$*

$$\boldsymbol{x}_s^+ = \mathcal{P}_{\Delta_{\mathcal{A}}} \left( \widetilde{\boldsymbol{x}}_s - \eta \boldsymbol{Q}_s^{\boldsymbol{x},\boldsymbol{y}} \boldsymbol{y}_s \right), \ \boldsymbol{y}_s^+ = \mathcal{P}_{\Delta_{\mathcal{B}}} \left( \widetilde{\boldsymbol{y}}_s + \eta \left( \boldsymbol{Q}_s^{\boldsymbol{x},\boldsymbol{y}} \right)^\top \boldsymbol{x}_s \right).$$

*Let $\boldsymbol{z}^+ = (\boldsymbol{x}^+, \boldsymbol{y}^+)$, then,*

$$\left\| \boldsymbol{z}^+ - \widetilde{\boldsymbol{z}} \right\|^2 \leq 8\mathrm{dist}^2 (\widetilde{\boldsymbol{z}}, \mathcal{Z}^*) + \frac{8S (A + B)^2 \eta^2}{(1-\gamma)^4} \mathrm{dist}^2 (\boldsymbol{z}, \mathcal{Z}^*).$$

*Proof of Lemma 9.* Denote $\boldsymbol{x}^* = \mathcal{P}_{\mathcal{X}^*}(\boldsymbol{x})$, $\boldsymbol{y}^* = \mathcal{P}_{\mathcal{Y}^*}(\boldsymbol{y})$ and $\boldsymbol{z}^* = (\boldsymbol{x}^*, \boldsymbol{y}^*)$. By Lemma 19, $(\boldsymbol{x}^*, \boldsymbol{y}^*)$ attains Nash equilibrium and $\boldsymbol{Q}_s^{\boldsymbol{x}^*,\boldsymbol{y}^*} = \boldsymbol{Q}_s^*$. By (91), we have

$$\max_{(a,b) \in \mathcal{A} \times \mathcal{B}} |\boldsymbol{Q}_s^{\boldsymbol{x},\boldsymbol{y}}(a,b) - \boldsymbol{Q}_s^*(a,b)| \leq \frac{\sqrt{A+B} \, \|\boldsymbol{z} - \boldsymbol{z}^*\|}{(1-\gamma)^2}.$$

Then, by combining with Lemma 2, we have

$$\left\| \boldsymbol{z}^+ - \widetilde{\boldsymbol{z}} \right\|^2 \leq 8\mathrm{dist}^2 (\widetilde{\boldsymbol{z}}, \mathcal{Z}^*) + \frac{8S (A + B)^2 \eta^2}{(1-\gamma)^4} \mathrm{dist}^2 (\boldsymbol{z}, \mathcal{Z}^*).$$

$\square$

The following lemma uses Lemma 9 to show that when $\Lambda^t$ is close to 0, $\left\| z^{t+1} - z^t \right\|$, $\left\| \widetilde{z}^{t+1} - \widetilde{z}^t \right\|$ will be small, which implies the difference between $\widetilde{\Theta}^t$ and $\Theta^t$ will also be small.

**Lemma 10** *Consider the sequence $\left\{ z^t, \widetilde{z}^t \right\}$ generated from OGDA with stepsize $\eta \leq \frac{(1-\gamma)^2}{2\sqrt{2S}(A+B)}$. There is a problem-dependent constant $\delta_1 = O\left( \frac{(1-\gamma)^5}{S^3(A+B)} \right) > 0$ such that for any $\tau > 0$ and $t \geq 1$, if $\Lambda^{t-1} \leq \tau^2 \delta_1$, we have*

$$\left| \widetilde{\Theta}^t - \Theta^t \right| \leq \tau \Theta^t.$$

*Proof of Lemma 10.* By the condition on $\eta$, we have $\frac{8S(A+B)^2\eta^2}{(1-\gamma)^4} \leq 1$. Denote $c' = \frac{S}{1-\gamma}$, $c'' = \frac{8S}{1-\gamma}$ and define the problem-dependent constant

$$\delta_1 = \frac{(1-\gamma)^4}{S^2(A+B)(1704c' + 226c'')} = O\left( \frac{(1-\gamma)^5}{S^3(A+B)} \right). \tag{42}$$

We also denote $\delta = \tau^2 \delta_1$ below.

The positive constants $c_1, c_2, \cdots, c_7$ below are all polynomials in $S, A, B, 1/(1-\gamma)$, the definition for each of them follows from the line it first occurs.

Since $d^t_x(s), d^t_y(s) \geq \frac{1-\gamma}{S} = c'$, the condition $\Lambda^{t-1} \leq \tau^2 \delta_1 = \delta$ implies that

$$\mathrm{dist}^2\left( \widetilde{z}^{t-1}, \mathcal{Z}^* \right) \leq c'\delta, \left\| \widetilde{z}^{t-1} - z^{t-1} \right\|^2 \leq c''\delta.$$

Then,

$$\mathrm{dist}^2\left( z^{t-1}, \mathcal{Z}^* \right) \leq 2\mathrm{dist}^2\left( \widetilde{z}^{t-1}, \mathcal{Z}^* \right) + 2\left\| \widetilde{z}^{t-1} - z^{t-1} \right\|^2 \leq 2(c' + c'')\delta.$$

By applying Lemma 9 with $\widetilde{z} := \widetilde{z}^{t-1}$, $z := z^{t-1}$, we have

$$\left\| z^t - \widetilde{z}^{t-1} \right\|^2 \leq (8c' + 2(c' + c''))\,\delta \overset{\mathrm{def}}{=} c_1\delta. \tag{43}$$

Thus,

$$\mathrm{dist}^2\left( z^t, \mathcal{Z}^* \right) \leq 2\mathrm{dist}^2\left( \widetilde{z}^{t-1}, \mathcal{Z}^* \right) + 2\left\| z^t - \widetilde{z}^{t-1} \right\|^2 \leq (2c' + 2c_1)\,\delta \overset{\mathrm{def}}{=} c_2\delta. \tag{44}$$

By setting $\widetilde{z} := \widetilde{z}^{t-1}$, $z := z^t$ in Lemma 9, we have

$$\left\| \widetilde{z}^t - \widetilde{z}^{t-1} \right\|^2 \leq (8c' + c_2)\,\delta \overset{\mathrm{def}}{=} c_3\delta. \tag{45}$$

Therefore,

$$\mathrm{dist}^2\left( \widetilde{z}^t, \mathcal{Z}^* \right) \leq 2\mathrm{dist}^2\left( \widetilde{z}^{t-1}, \mathcal{Z}^* \right) + 2\left\| \widetilde{z}^t - \widetilde{z}^{t-1} \right\|^2 \leq (2c' + 2c_3)\,\delta \overset{\mathrm{def}}{=} c_4\delta. \tag{46}$$

Again, utilize Lemma 9 with $\widetilde{z} := \widetilde{z}^t$, $z := z^t$, we have

$$\left\| z^{t+1} - \widetilde{z}^t \right\|^2 \leq (8c_4 + c_2)\,\delta \overset{\mathrm{def}}{=} c_5\delta. \tag{47}$$

Thus,

$$\left\| z^{t+1} - z^t \right\|^2 \leq 3\left( \left\| z^{t+1} - \widetilde{z}^t \right\|^2 + \left\| \widetilde{z}^t - \widetilde{z}^{t-1} \right\|^2 + \left\| \widetilde{z}^{t-1} - z^t \right\|^2 \right)$$
$$\leq 3\left( c_5 + c_3 + c_1 \right)\delta \overset{\mathrm{def}}{=} c_6\delta. \tag{48}$$

Now we can bound

$$\left\| \widetilde{z}^t - \widetilde{z}^{t-1} \right\|^2 + \left\| z^{t+1} - z^t \right\|^2 \leq (c_3 + c_6)\,\delta \overset{\mathrm{def}}{=} c_7\delta. \tag{49}$$

Since $\mathcal{X}_s^*$ is a convex set, the projection onto it is non-expansive, i.e., $\left\|\widetilde{\boldsymbol{x}}_s^{t*} - \widetilde{\boldsymbol{x}}_s^{(t-1)*}\right\| = \left\|\mathcal{P}_{\mathcal{X}_s^*}\left(\widetilde{\boldsymbol{x}}_s^t\right) - \mathcal{P}_{\mathcal{X}_s^*}\left(\widetilde{\boldsymbol{x}}_s^{t-1}\right)\right\| \le \left\|\widetilde{\boldsymbol{x}}_s^t - \widetilde{\boldsymbol{x}}_s^{t-1}\right\|$. Then,

$$\left\|\left(\widetilde{\boldsymbol{x}}^{t*}, \boldsymbol{y}^{t+1}\right) - \left(\widetilde{\boldsymbol{x}}^{(t-1)*}, \boldsymbol{y}^t\right)\right\|^2 \le \left\|\widetilde{\boldsymbol{x}}^{t*} - \widetilde{\boldsymbol{x}}^{(t-1)*}\right\|^2 + \left\|\boldsymbol{y}^{t+1} - \boldsymbol{y}^t\right\|^2$$

$$\le \left\|\widetilde{\boldsymbol{x}}^t - \widetilde{\boldsymbol{x}}^{t-1}\right\|^2 + \left\|\boldsymbol{y}^{t+1} - \boldsymbol{y}^t\right\|^2 \le \left\|\widetilde{\boldsymbol{z}}^t - \widetilde{\boldsymbol{z}}^{t-1}\right\|^2 + \left\|\boldsymbol{z}^{t+1} - \boldsymbol{z}^t\right\|^2 \le c_7\delta.$$

Analogously,

$$\left\|\left(\boldsymbol{x}^{t+1}, \boldsymbol{y}^{t*}\right) - \left(\boldsymbol{x}^t, \widetilde{\boldsymbol{y}}^{(t-1)*}\right)\right\|^2 \le \left\|\widetilde{\boldsymbol{z}}^t - \widetilde{\boldsymbol{z}}^{t-1}\right\|^2 + \left\|\boldsymbol{z}^{t+1} - \boldsymbol{z}^t\right\|^2 \le c_7\delta.$$

By Lemma 21 and (25), for any $s \in \mathcal{S}$,

$$\left|\boldsymbol{d}_x^{t+1}(s) - \boldsymbol{d}_x^t(s)\right| = \left|\boldsymbol{d}_{\boldsymbol{\rho}_0}^{\widetilde{\boldsymbol{x}}^{t*}, \boldsymbol{y}^{t+1}}(s) - \boldsymbol{d}_{\boldsymbol{\rho}_0}^{\widetilde{\boldsymbol{x}}^{(t-1)*}, \boldsymbol{y}^t}(s)\right|$$

$$\le \frac{\sqrt{A+B}\left\|\left(\widetilde{\boldsymbol{x}}^{t*}, \boldsymbol{y}^{t+1}\right) - \left(\widetilde{\boldsymbol{x}}^{(t-1)*}, \boldsymbol{y}^t\right)\right\|}{1-\gamma} \le \frac{\sqrt{(A+B)\,c_7\delta}}{1-\gamma}. \tag{50}$$

Similarly, we also have for any $s \in \mathcal{S}$,

$$\left|\boldsymbol{d}_y^{t+1}(s) - \boldsymbol{d}_y^t(s)\right| \le \frac{\sqrt{(A+B)\,c_7\delta}}{1-\gamma}. \tag{51}$$

What remains is to bound the term $\frac{\sqrt{(A+B)c_7\delta}}{1-\gamma}$ on the RHS of (50) and (51). Using (43)-(49), we have

- (by (43)) $c_1 = 10c' + 2c''$
- (by (44)) $c_2 = 22c' + 4c''$
- (by (45)) $c_3 = 30c' + 4c''$
- (by (46)) $c_4 = 62c' + 8c''$
- (by (47)) $c_5 = 518c' + 68c''$
- (by (48)) $c_6 = 1674c' + 222c''$
- (by (49)) $c_7 = 1704c' + 226c''$

By the definition of $\delta_1$ in (42) and our notation $\delta = \tau^2\delta_1$, we have

$$\frac{\sqrt{(A+B)\,c_7\delta}}{1-\gamma} = \frac{\tau(1-\gamma)}{S}.$$

Then, by combining with (50), we have $\left|\boldsymbol{d}_x^t(s) - \boldsymbol{d}_x^{t+1}(s)\right| \le \frac{\tau(1-\gamma)}{S}$. By combining with the fact that $\boldsymbol{d}_x^t(s) \ge \frac{1-\gamma}{S}$, we have

$$\left|\boldsymbol{d}_x^t(s) - \boldsymbol{d}_x^{t+1}(s)\right| \le \tau\boldsymbol{d}_x^t(s).$$

Analogously, for any $s \in \mathcal{S}$,

$$\left|\boldsymbol{d}_y^t(s) - \boldsymbol{d}_y^{t+1}(s)\right| \le \tau\boldsymbol{d}_y^t(s).$$

Then, the result follows by the definition of $\Theta^t$ and $\widetilde{\Theta}^t$ in (26). $\qquad\square$

### C.4 Proof of Theorem 5

By Lemma 8, we have

$$\frac{1-\gamma}{4S}\left(\left\|\widetilde{\boldsymbol{z}}^{t+1} - \boldsymbol{z}^{t+1}\right\|^2 + \left\|\boldsymbol{z}^{t+1} - \widetilde{\boldsymbol{z}}^t\right\|^2\right) \ge c_8\eta^2\Theta^t, \tag{52}$$

where

$$c_8 = \frac{c_+^2 (1-\gamma)^3}{144 S^2}.$$ (53)

Then, combining (52) with Lemma 4 and the definitions of $\Lambda^t, \Theta^t, \widetilde{\Theta}^t$ in (27), (26) yield that for any $t \geq 1$,

$$
\begin{aligned}
\Lambda^{t+1} \leq & \Lambda^t + \widetilde{\Theta}^t - \Theta^t - \frac{1-\gamma}{8S} \left\| \widetilde{z}^t - z^t \right\|^2 - \frac{1-\gamma}{4S} \left( \left\| z^{t+1} - \widetilde{z}^t \right\|^2 + \left\| \widetilde{z}^t - z^t \right\|^2 \right) \\
\leq & \Lambda^t + \widetilde{\Theta}^t - \Theta^t - \frac{1-\gamma}{8S} \left\| \widetilde{z}^t - z^t \right\|^2 - c_8 \eta^2 \Theta^t.
\end{aligned}
$$ (54)

Define

$$c_0 = \min \left\{ \frac{(1-\gamma)c_8}{S}, \frac{c_8}{2}, \frac{1}{2} \right\} > 0.$$ (55)

Recall the problem-dependent constant $\delta_1 > 0$ defined in Lemma 10, we define

$$\delta_0 = c_0^2 \delta_1 > 0.$$ (56)

Now, we prove $\Lambda^{t+1} \leq (1-c_0)^t \Lambda^t$ by induction. For the case $t = 0$, firstly, by the definitions of $\Lambda^0, \widetilde{\Theta}^0, \Theta^0$ in (27), (26) and the fact that $\frac{1-\gamma}{S} \leq d_x^t(s) \leq 1, \frac{1-\gamma}{S} \leq d_y^t(s) \leq 1$, we have

$$\Lambda^0 \geq \widetilde{\Theta}^0, \quad \Theta^0 \geq \frac{1-\gamma}{S} \Lambda^0.$$

Then, by combining with Lemma 4, (52) and the fact that $\widetilde{z}^0 = z^0$, we have

$$\Lambda^1 \leq \Lambda^0 - c_8 \eta^2 \Theta^0 \leq \left( 1 - \frac{(1-\gamma)c_8 \eta^2}{S} \right) \Lambda^0 \leq \left( 1 - c_0 \eta^2 \right) \Lambda^0.$$

If we have shown $\Lambda^{j+1} \leq (1-c_0)^j \Lambda^j$ for $j = 0, \cdots, t-1$, we next prove it for $t$. By induction hypothesis,

$$\Lambda^{t-1} \leq \Lambda^0 \leq \delta_0 \eta^4 = (c_0 \eta^2)^2 \delta_1.$$

By Lemma 10,

$$\widetilde{\Theta}^t \leq \left( 1 + c_0 \eta^2 \right) \Theta^t.$$

Then, by combining with (54) and the fact that $c_0 \leq c_8/2, c_0 \leq 1/2$ from the definition of $c_0$ in (55), we have

$$
\begin{aligned}
\Lambda^{t+1} \leq & \Lambda^t + c_0 \eta^2 \Theta^t - \frac{1-\gamma}{16S} \left\| \widetilde{z}^t - z^t \right\|^2 - 2c_0 \eta^2 \Theta^t \\
\leq & \Lambda^t - \min \left\{ c_0 \eta^2, \frac{1}{2} \right\} \left( \Theta^t + \frac{1-\gamma}{8S} \left\| \widetilde{z}^t - z^t \right\|^2 \right) \\
= & \left( 1 - c_0 \eta^2 \right) \Lambda^t.
\end{aligned}
$$

By induction, we have for any $t \geq 0$,

$$\Lambda^t \leq \left( 1 - c_0 \eta^2 \right)^t \Lambda^0.$$

Using the fact that $\frac{1-\gamma}{S} \leq d_x^t(s) \leq 1, \frac{1-\gamma}{S} \leq d_y^t(s) \leq 1$ and the definition of $\Lambda^t$ in (27), we have

$$
\begin{aligned}
\mathrm{dist}^2 \left( z^t, \mathcal{Z}^* \right) \leq & 2 \left( \mathrm{dist}^2 \left( \widetilde{z}^t, \mathcal{Z}^* \right) + \left\| \widetilde{z}^t - z^t \right\|^2 \right) \leq \frac{8S}{1-\gamma} \Lambda^t \\
\leq & \frac{8S}{1-\gamma} \left( 1 - c_0 \eta^2 \right)^t \Lambda^0 \leq \frac{8S}{1-\gamma} \left( 1 - c_0 \eta^2 \right)^t \mathrm{dist}^2 \left( z^0, \mathcal{Z}^* \right) \\
= & \frac{8S}{1-\gamma} \left( 1 - c_0 \eta^2 \right)^t \mathrm{dist}^2 \left( \widehat{z}, \mathcal{Z}^* \right),
\end{aligned}
$$

where $\widehat{z} = (\widehat{x}, \widehat{y})$ is the initial policy pair (22).

This completes the proof for local linear convergence of OGDA.

As for the order of $c_0$ and $\delta_0$, by (53) and (55),

$$c_0 = O\left(\frac{(1-\gamma)^4 c_+^2}{S^3}\right). \tag{57}$$

By Lemma 10, $\delta_1 = O\left(\frac{(1-\gamma)^5}{S^3(A+B)}\right)$. Then, by (57) and (56),

$$\delta_0 = O\left(\frac{(1-\gamma)^{13} c_+^4}{S^9(A+B)}\right). \tag{58}$$

Since we need $\eta \le O\left(\frac{(1-\gamma)^{\frac{5}{2}}}{\sqrt{S}(A+B)}\right)$ in Theorem 5, by setting $\eta = O\left(\frac{(1-\gamma)^{\frac{5}{2}}}{\sqrt{S}(A+B)}\right)$, we have the linear convergence rate

$$1 - c_0\eta^2 = 1 - O\left(\frac{(1-\gamma)^9 c_+^2}{S^4(A+B)^2}\right)$$

and to have linear convergence, $\mathrm{dist}(z^{T_1}, \mathcal{Z}^*)$ needs to satisfy

$$\mathrm{dist}(z^{T_1}, \mathcal{Z}^*) \le \sqrt{\delta_0 \eta^4} = O\left(\frac{(1-\gamma)^{\frac{23}{2}} c_+^2}{S^{\frac{11}{2}}(A+B)^{\frac{5}{2}}}\right).$$

# D  PROOFS FOR GLOBAL CONVERGENCE AND GEOMETRIC BOUNDEDNESS OF AVERAGING OGDA

In this section, we prove that the Averaging OGDA method introduced in (6) of Section 3.2 can serve as `Global-Slow` in the meta algorithm `Homotopy-PO`. The proof of global convergence (Theorem 3) is in Appendix D.1. The proof of geometric boundedness (Theorem 4) is in Appendix D.2.

To begin with, let us recall the Averaging OGDA method: The min-player initializes

$$\widetilde{x}^{T_1} = x^{T_1} = \tilde{x}, \; \underline{V}^{T_1}(s) = V^{\dagger, \tilde{y}}(s) = V^{\dagger, y^{T_1}}(s) \tag{59}$$

while the max-player initializes

$$\widetilde{y}^{T_1} = y^{T_1} = \tilde{y}, \; \overline{V}^{T_1}(s) = V^{\tilde{x},\dagger}(s) = V^{x^{T_1},\dagger}(s). \tag{60}$$

The min-player updates for $t > T_1$ as follows:

$$\underline{V}^t(s) = \min_{a \in \mathcal{A}} \sum_{j=T_1}^{t-1} \alpha_{t-T_1}^{j-T_1+1} \underline{q}_s^j(a), \tag{61a}$$

$$x_s^t = \mathcal{P}_{\Delta_{\mathcal{A}}}\left(\widetilde{x}_s^{t-1} - \eta \underline{q}_s^{t-1}\right), \tag{61b}$$

$$\widetilde{x}_s^t = \mathcal{P}_{\Delta_{\mathcal{A}}}\left(\widetilde{x}_s^{t-1} - \eta \underline{q}_s^t\right), \tag{61c}$$

where

$$\underline{q}_s^t = Q_s[\underline{V}^t] y_s^t, \tag{62}$$

and $Q_s[\cdot]$ is the Bellman target operator defined in the introduction. Meanwhile, the max-player updates for $t > T_1$ as follows:

$$\overline{V}^t(s) = \max_{b \in \mathcal{B}} \sum_{j=T_1}^{t-1} \alpha_{t-T_1}^{j-T_1+1} \overline{q}_s^j(b), \tag{63a}$$

$$y_s^t = \mathcal{P}_{\Delta_{\mathcal{B}}}\left(\widetilde{y}_s^{t-1} + \eta \overline{q}_s^{t-1}\right), \tag{63b}$$

$$\widetilde{y}_s^t = \mathcal{P}_{\Delta_{\mathcal{B}}}\left(\widetilde{y}_s^{t-1} + \eta \overline{q}_s^t\right), \tag{63c}$$

where

$$\overline{\boldsymbol{q}}_s^t = \left( \boldsymbol{Q}_s[\overline{V}^t] \right)^\top \boldsymbol{x}_s^t.$$

At the end iteration $T_2$, the min-player and the max-player compute the following average policies respectively

$$\widehat{\boldsymbol{x}}^{[T_1:T_2]} = \sum_{t=T_1}^{T_2} \alpha_{T_2-T_1+1}^{t-T_1+1} \boldsymbol{x}^t, \ \widehat{\boldsymbol{y}}^{[T_1:T_2]} = \sum_{t=T_1}^{T_2} \alpha_{T_2-T_1+1}^{t-T_1+1} \boldsymbol{y}^t.$$

The min-player plays policy $\boldsymbol{x}^t$ and the max-player plays policy $\boldsymbol{y}^t$ at iteration $t$. The variables $\widetilde{\boldsymbol{x}}^t$, $\underline{V}^t$ and $\widetilde{\boldsymbol{y}}^t$, $\overline{V}^t$ are all local auxiliary variables to help generate the policies $\boldsymbol{x}^t$ and $\boldsymbol{y}^t$.

Averaging OGDA is a decentralized algorithm whose decentralized implementation is in Algorithm 4 (for the min-player) and Algorithm 5 (for the max-player) of Appendix G.

### D.1 GLOBAL CONVERGENCE RATE OF AVERAGING OGDA

Our task in this section is to prove the global convergence of Averaging OGDA (Theorem 3). To this end, we need to bound $\mathrm{dist}^2 \left( \widehat{\boldsymbol{z}}^{[T_1:T_2]}, \mathcal{Z}^* \right)$ by $O(\log(T_2 - T_1)/(T_2 - T_1))$. Our roadmap can be depicted as follows:

$$\mathrm{dist}^2 \left( \widehat{\boldsymbol{z}}^{[T_1:T_2]}, \mathcal{Z}^* \right) \overset{\text{Lemma 11}}{\Longleftarrow} \left\| \overline{V}^t - \underline{V}^t \right\|_\infty \overset{\text{Lemma 12}}{\Longleftarrow} \mathrm{Reg}^{T_1:t} \overset{\text{Lemma 14}}{\leq} O(1/(T_2 - T_1))$$

The regrets above are defined as

$$\mathrm{Reg}_x^{T_1:t}(s) = \min_{\boldsymbol{x}_s' \in \Delta_{\mathcal{A}}} \sum_{j=T_1}^{t} \alpha_{t-T_1+1}^{j-T_1+1} \left\langle \boldsymbol{x}_s' - \boldsymbol{x}_s^j, \boldsymbol{Q}_s[\underline{V}^j] \boldsymbol{y}_s^j \right\rangle,$$

$$\mathrm{Reg}_y^{T_1:t}(s) = \max_{\boldsymbol{y}_s' \in \Delta_{\mathcal{B}}} \sum_{j=T_1}^{t} \alpha_{t-T_1+1}^{j-T_1+1} \left\langle \boldsymbol{x}_s^j, \boldsymbol{Q}_s[\underline{V}^j] \left( \boldsymbol{y}_s' - \boldsymbol{y}_s^j \right) \right\rangle,$$

$$\mathrm{Reg}^{T_1:t} = \max_{s \in \mathcal{S}} \left( \mathrm{Reg}_y^{T_1:t}(s) - \mathrm{Reg}_x^{T_1:t}(s) \right).$$

More specifically, we bound the distance $\mathrm{dist}(\widehat{\boldsymbol{z}}^{[T_1:T_2]}, \mathcal{Z}^*)$ in the following steps:

1. (Lemma 11) bounding $\mathrm{dist}(\widehat{\boldsymbol{z}}^{[T_1:T_2]}, \mathcal{Z}^*)$ by $O(\left\| \overline{V}^{T_2+1} - \underline{V}^{T_2+1} \right\|_\infty)$:

$$\mathrm{dist} \left( \widehat{\boldsymbol{z}}^{[T_1:T_2]}, \mathcal{Z}^* \right) \leq O \left( \left\| \overline{V}^{T_2+1} - \underline{V}^{T_2+1} \right\|_\infty \right)$$

2. (Lemma 12) bounding $O(\left\| \overline{V}^{T_2+1} - \underline{V}^{T_2+1} \right\|_\infty)$ by regrets:

$$\left\| \overline{V}^{T_2+1} - \underline{V}^{T_2+1} \right\|_\infty \leq \mathrm{Reg}^{T_1:T_2} + O(\frac{1}{T_2 - T_1}) \cdot (\sum_{t=T_1}^{T_2} \mathrm{Reg}^{T_1:t} + \|\overline{V}^{T_1} - \underline{V}^{T_1}\|_\infty)$$

3. (Lemma 14) bounding the regrets:

$$\mathrm{Reg}^{T_1:t} \leq O \left( \frac{1}{\eta(t - T_1)} \right)$$

The following fact about $\alpha_t^j$ can be found in Section 4 of Jin et al. (2018). It will be used extensively in our proofs below.

**Fact 1** *The stepsize $\alpha_t^j$ satisfy:*

*(i)* $\sum_{t=j}^{\infty} \alpha_t^j = 1 + \frac{1}{H}, \forall\, t \geq 1.$

*(ii)* $\sum_{j=1}^{t} \alpha_t^j = 1, \forall\, t \geq 1.$

*(iii)* $\alpha_t^j \leq \alpha_t$ and $\alpha_{t+1}^j \leq \alpha_t^j, \forall\, t \geq 1, 1 \leq j \leq t.$

Firstly, we show that the local auxiliary variables $\underline{V}^t(s)$, $\overline{V}^t(s)$ are lower and upper bounds for $v^*(s)$. Then, to bound $\left\|\overline{V}^t - v^*\right\|_\infty$ and $\left\|\underline{V}^t - v^*\right\|_\infty$, it suffices to bound $\left\|\overline{V}^t - \underline{V}^t\right\|_\infty$.

**Fact 2** *For any $t \in [T_1 : T_2]$ and $s \in \mathcal{S}$,*

$$0 \leq \underline{V}^t(s) \leq v^*(s) \leq \overline{V}^t(s) \leq \frac{1}{1-\gamma},$$

$$0 \leq \min_{a \in \mathcal{A}} \underline{\boldsymbol{q}}_s^t(a) \leq v^*(s) \leq \max_{b \in \mathcal{B}} \overline{\boldsymbol{q}}_s^t(b) \leq \frac{1}{1-\gamma}.$$

*Proof of Fact 2.* By (61a), we have

$$\underline{V}^{T_1}(s) = V^{\dagger, \boldsymbol{y}^{T_1}}(s) \leq v^*(s).$$

By the definition of $\underline{\boldsymbol{q}}_s^t$ in (62),

$$\min_{a \in \mathcal{A}} \underline{\boldsymbol{q}}_s^t(a) = \min_{\boldsymbol{x}_s' \in \Delta_\mathcal{A}} \left\langle \boldsymbol{x}_s', \boldsymbol{Q}_s[\underline{V}^t]\boldsymbol{y}_s^t \right\rangle.$$

Recall that by Lemma 19, $v^*(s) = \min_{\boldsymbol{x}_s} \max_{\boldsymbol{y}_s} \left\langle \boldsymbol{x}_s, \boldsymbol{Q}_s^* \boldsymbol{y}_s \right\rangle$ and $\boldsymbol{Q}_s^* = \boldsymbol{Q}_s[v^*]$.

Suppose $\underline{V}^j(s) \leq v^*(s)$ for any $s \in \mathcal{S}$ and $j \in [T_1 : t]$, then we have

$$\min_{a \in \mathcal{A}} \underline{\boldsymbol{q}}_s^j(a) = \min_{\boldsymbol{x}_s' \in \Delta_\mathcal{A}} \left\langle \boldsymbol{x}_s', \boldsymbol{Q}_s[\underline{V}^j]\boldsymbol{y}_s^j \right\rangle \leq \min_{\boldsymbol{x}_s' \in \Delta_\mathcal{A}} \left\langle \boldsymbol{x}_s', \boldsymbol{Q}_s[v^*]\boldsymbol{y}_s^t \right\rangle$$

$$\leq \min_{\boldsymbol{x}_s'} \max_{\boldsymbol{y}_s'} \left\langle \boldsymbol{x}_s', \boldsymbol{Q}_s^* \boldsymbol{y}_s' \right\rangle = v^*(s),$$

which leads to $\underline{V}^{t+1}(s) \leq v^*(s)$ for any $s \in \mathcal{S}$.

Then, it follows by induction that $\underline{V}^t(s) \leq v^*(s)$, $\min_{a \in \mathcal{A}} \underline{\boldsymbol{q}}_s^t(a) \leq v^*(s)$ for any $t \in [T_1 : T_2]$ and $s \in \mathcal{S}$. Analogously, $\overline{V}^t(s) \geq v^*(s)$, $\max_{b \in \mathcal{B}} \overline{\boldsymbol{q}}_s^t(a) \geq v^*(s)$ for any $t \in [T_1 : T_2]$ and $s \in \mathcal{S}$.

It also follows by induction directly that the value of $\underline{V}^t(s), \overline{V}^t(s), \min_{a \in \mathcal{A}} \underline{\boldsymbol{q}}_s^t(a), \max_{b \in \mathcal{B}} \overline{\boldsymbol{q}}_s^t(a)$ stays in $[0, \frac{1}{1-\gamma}]$.

$\square$

The following lemma shows that to bound $\mathrm{dist}^2\left(\widehat{\boldsymbol{z}}^{[T_1:T_2]}, \mathcal{Z}^*\right)$, it suffices to bound $\left\|\overline{V}^t - \underline{V}^t\right\|_\infty$.

**Lemma 11** *There is a problem-dependent constant $\widehat{C} = \frac{\sqrt{S}}{c_+} > 0$ such that the average policy $\widehat{\boldsymbol{z}}^{[T_1:T_2]} = \left(\widehat{\boldsymbol{x}}^{[T_1:T_2]}, \widehat{\boldsymbol{y}}^{[T_1:T_2]}\right)$ satisfies*

$$\mathrm{dist}\left(\widehat{\boldsymbol{z}}^{[T_1:T_2]}, \mathcal{Z}^*\right) \leq \widehat{C} \cdot \left\|\overline{V}^{T_2+1} - \underline{V}^{T_2+1}\right\|_\infty.$$

*Proof of Lemma 11.* Recall that $\boldsymbol{Q}_s^* = \boldsymbol{Q}_s[v^*]$. By (62) and Fact 2,

$$\min_{a \in \mathcal{A}} \sum_{t=T_1}^{T_2} \alpha_{T_2-T_1}^{t-T_1+1} \underline{\boldsymbol{q}}_s^t(a) = \min_{\boldsymbol{x}_s' \in \Delta_\mathcal{A}} \sum_{t=T_1}^{T_2} \alpha_{T_2-T_1}^{t-T_1+1} \left\langle \boldsymbol{x}_s', \boldsymbol{Q}_s[\underline{V}^t]\boldsymbol{y}_s^t \right\rangle$$

$$\leq \min_{\boldsymbol{x}_s' \in \Delta_\mathcal{A}} \sum_{t=T_1}^{T_2} \alpha_{T_2-T_1}^{t-T_1+1} \left\langle \boldsymbol{x}_s', \boldsymbol{Q}_s^* \boldsymbol{y}_s^t \right\rangle.$$

Analogously,

$$\max_{b \in \mathcal{B}} \sum_{t=T_1}^{T_2} \alpha_{T_2-T_1}^{t-T_1+1} \overline{q}_s^t(b) = \max_{y_s' \in \Delta_{\mathcal{B}}} \sum_{t=T_1}^{T_2} \alpha_{T_2-T_1}^{t-T_1+1} \left\langle x_s^t, Q_s[\overline{V}^t] y_s' \right\rangle$$

$$\geq \max_{y_s' \in \Delta_{\mathcal{B}}} \sum_{t=T_1}^{T_2} \alpha_{T_2-T_1}^{t-T_1+1} \left\langle x_s^t, Q_s^* y_s' \right\rangle.$$

Thus,

$$\overline{V}^{T_2+1}(s) - \underline{V}^{T_2+1}(s)$$

$$= \max_{b \in \mathcal{B}} \sum_{t=T_1}^{T_2} \alpha_{T_2-T_1}^{t-T_1+1} \overline{q}_s^t(b) - \min_{a \in \mathcal{A}} \sum_{t=T_1}^{T_2} \alpha_{T_2-T_1}^{t-T_1+1} \underline{q}_s^t(a)$$

$$\geq \max_{y_s' \in \Delta_{\mathcal{B}}} \sum_{t=T_1}^{T_2} \alpha_{T_2-T_1}^{t-T_1+1} \left\langle x_s^t, Q_s^* y_s' \right\rangle - \min_{x_s' \in \Delta_{\mathcal{A}}} \sum_{t=T_1}^{T_2} \alpha_{T_2-T_1}^{t-T_1+1} \left\langle x_s', Q_s^* y_s^t \right\rangle$$

$$= \max_{y_s' \in \Delta_{\mathcal{B}}} \left\langle \widehat{x}_s^{[T_1:T_2]}, Q_s^* y_s' \right\rangle - \min_{x_s' \in \Delta_{\mathcal{A}}} \left\langle x_s', Q_s^* \widehat{y}_s^{[T_1:T_2]} \right\rangle.$$

By (1),

$$\max_{y_s' \in \Delta_{\mathcal{B}}} \left\langle \widehat{x}_s^{[T_1:T_2]}, Q_s^* y_s' \right\rangle - \min_{x_s' \in \Delta_{\mathcal{A}}} \left\langle x_s', Q_s^* \widehat{y}_s^{[T_1:T_2]} \right\rangle \geq c_+ \cdot \mathrm{dist}\left( \widehat{z}_s^{[T_1:T_2]}, \mathcal{Z}_s^* \right).$$

The positiveness of $c_+$ is guaranteed by Lemma 22.

Let $\widehat{C} = \frac{\sqrt{S}}{c_+}$, then,

$$\left\| \overline{V}^{T_2+1} - \underline{V}^{T_2+1} \right\|_\infty \geq \max_{s \in \mathcal{S}} \left( \max_{y_s' \in \Delta_{\mathcal{B}}} \left\langle \widehat{x}_s^{[T_1:T_2]}, Q_s^* y_s' \right\rangle - \min_{x_s' \in \Delta_{\mathcal{A}}} \left\langle x_s', Q_s^* \widehat{y}^{[T_1:T_2]} \right\rangle \right)$$

$$\geq \max_{s \in \mathcal{S}} c_+ \cdot \mathrm{dist}\left( \widehat{z}_s^{[T_1:T_2]}, \mathcal{Z}_s^* \right) \geq \frac{1}{\widehat{C}} \cdot \mathrm{dist}\left( \widehat{z}^{[T_1:T_2]}, \mathcal{Z}^* \right).$$

$$\square$$

The following lemma mainly uses Fact 1 (i) and induction to show that $\left\| \overline{V}^{T_2+1} - \underline{V}^{T_2+1} \right\|_\infty$ can be bounded by weighted sum of the regrets.

**Lemma 12** *The value functions $\overline{V}^{T_2+1}$, $\underline{V}^{T_2+1}$ satisfies*

$$\left\| \overline{V}^{T_2+1} - \underline{V}^{T_2+1} \right\|_\infty \leq \mathrm{Reg}^{T_1:T_2} + \frac{2\gamma(H+1)}{(1-\gamma)(T_2-T_1+1)} \left( \sum_{t=T_1}^{T_2} \mathrm{Reg}^{T_1:t} + \| \overline{V}^{T_1} - \underline{V}^{T_1} \|_\infty \right).$$

*Proof of Lemma 12.* By Fact 2 and the definition of the operator $Q_s[\cdot]$, we have

$$\max_{(a,b) \in \mathcal{A} \times \mathcal{B}} \left( Q_s[\overline{V}^t](a,b) - Q_s[\underline{V}^t](a,b) \right) \leq \gamma \left\| \overline{V}^t - \underline{V}^t \right\|_\infty. \tag{64}$$

The following relation follows by definitions of $\underline{V}^t$ in (61a) and $\underline{q}_s^j$ in (62),

$$\underline{V}^t(s) = \min_{a \in \mathcal{A}} \sum_{j=T_1}^{t} \alpha_{t-T_1+1}^{j-T_1+1} \underline{q}_s^j(a) = \min_{x_s' \in \Delta_{\mathcal{A}}} \sum_{j=T_1}^{t} \alpha_{t-T_1+1}^{j-T_1+1} \left\langle x_s', Q_s[\underline{V}^j] y_s^j \right\rangle.$$

Analogously,

$$\overline{V}^t(s) = \max_{b \in \mathcal{B}} \sum_{j=T_1}^{t} \alpha_{t-T_1+1}^{j-T_1+1} \overline{q}_s^j(b) = \max_{y_s' \in \Delta_{\mathcal{B}}} \sum_{j=T_1}^{t} \alpha_{t-T_1+1}^{j-T_1+1} \left\langle x_s^j, Q_s[\overline{V}^j] y_s' \right\rangle.$$

Summing up the above two equations yields that

$$
\begin{aligned}
&\overline{V}^{t+1}(s) - \underline{V}^{t+1}(s) \\
&= \max_{\boldsymbol{y}'_s \in \Delta_{\mathcal{B}}} \sum_{j=T_1}^{t} \alpha_{t-T_1+1}^{j-T_1+1} \left\langle \boldsymbol{x}_s^j, \boldsymbol{Q}_s[\overline{V}^j] \boldsymbol{y}_s^j \right\rangle - \min_{\boldsymbol{x}'_s \in \Delta_{\mathcal{A}}} \sum_{j=T_1}^{t} \alpha_{t-T_1+1}^{j-T_1+1} \left\langle \boldsymbol{x}'_s, \boldsymbol{Q}_s[\underline{V}^j] \boldsymbol{y}_s^j \right\rangle \\
&\leq \max_{\boldsymbol{y}'_s \in \Delta_{\mathcal{B}}} \sum_{j=T_1}^{t} \alpha_{t-T_1+1}^{j-T_1+1} \left\langle \boldsymbol{x}_s^j, \boldsymbol{Q}_s[\overline{V}^j] \left( \boldsymbol{y}'_s - \boldsymbol{y}_s^j \right) \right\rangle \\
&\quad - \min_{\boldsymbol{x}'_s \in \Delta_{\mathcal{A}}} \sum_{j=T_1}^{t} \alpha_{t-T_1+1}^{j-T_1+1} \left\langle \boldsymbol{x}'_s - \boldsymbol{x}_s^j, \boldsymbol{Q}_s[\underline{V}^j] \boldsymbol{y}_s^j \right\rangle \\
&\quad + \sum_{j=T_1}^{t} \alpha_{t-T_1+1}^{j-T_1+1} \left\langle \boldsymbol{x}_s^j, \left( \boldsymbol{Q}_s[\overline{V}^j] - \boldsymbol{Q}_s[\underline{V}^j] \right) \boldsymbol{y}_s^j \right\rangle \\
&\leq \mathrm{Reg}_y^{T_1:t}(s) - \mathrm{Reg}_x^{T_1:t}(s) + \gamma \sum_{j=T_1}^{t} \alpha_{t-T_1+1}^{j-T_1+1} \left\| \overline{V}^j - \underline{V}^j \right\|_\infty,
\end{aligned}
$$

where the last inequality is by (64). Thus,

$$
\left\| \overline{V}^{t+1} - \underline{V}^{t+1} \right\|_\infty \leq \mathrm{Reg}^{T_1:t} + \gamma \sum_{j=T_1}^{t} \alpha_{t-T_1+1}^{j-T_1+1} \left\| \overline{V}^j - \underline{V}^j \right\|_\infty. \tag{65}
$$

Taking sum on both sides of the above equation and combining with Fact 1 (i) yield that

$$
\begin{aligned}
\sum_{t=T_1}^{T_2} \left\| \overline{V}^{t+1} - \underline{V}^{t+1} \right\|_\infty &\leq \sum_{t=T_1}^{T_2} \mathrm{Reg}^{T_1:t} + \gamma \sum_{t=T_1}^{T_2} \sum_{j=T_1}^{t} \alpha_{t-T_1+1}^{j-T_1+1} \left\| \overline{V}^j - \underline{V}^j \right\|_\infty \\
&\leq \sum_{t=T_1}^{T_2} \mathrm{Reg}^{T_1:t} + \gamma \sum_{j=T_1}^{T_2} \sum_{t=j}^{T_2} \alpha_{t-T_1+1}^{j-T_1+1} \left\| \overline{V}^j - \underline{V}^j \right\|_\infty \\
&\leq \sum_{t=T_1}^{T_2} \mathrm{Reg}^{T_1:t} + \gamma \sum_{j=T_1}^{T_2} \left( 1 + \frac{1}{H} \right) \left\| \overline{V}^j - \underline{V}^j \right\|_\infty \\
&\leq \sum_{t=T_1}^{T_2} \mathrm{Reg}^{T_1:t} + \gamma \left( 1 + \frac{1}{H} \right) \sum_{j=T_1}^{T_2} \left\| \overline{V}^j - \underline{V}^j \right\|_\infty \\
&\leq \sum_{t=T_1}^{T_2} \mathrm{Reg}^{T_1:t} + \frac{2\gamma}{1+\gamma} \sum_{j=T_1}^{T_2} \left\| \overline{V}^j - \underline{V}^j \right\|_\infty,
\end{aligned}
$$

where the last inequality is from the fact that $H = \frac{1+\gamma}{1-\gamma}$.

After rearranging, we have

$$
\sum_{t=T_1}^{T_2} \left\| \overline{V}^{t+1} - \underline{V}^{t+1} \right\|_\infty \leq \frac{1+\gamma}{1-\gamma} \left( \sum_{t=T_1}^{T_2} \mathrm{Reg}^{T_1:t} + \frac{2\gamma}{1+\gamma} \left\| \overline{V}^{T_1} - \underline{V}^{T_1} \right\|_\infty \right). \tag{66}
$$

Since $\alpha_{T_2-T_1+1}^{j-T_1+1} \leq \alpha_{T_2-T_1+1} \leq \frac{H+1}{T_2-T_1+1}$ for any $j \in [T_1:t]$, by setting $t := T_2$ in (65) and substituting (66), we have

$$\left\|\overline{V}^{T_2+1} - \underline{V}^{T_2+1}\right\|_\infty \leq \mathrm{Reg}^{T_1:T_2} + \gamma \frac{H+1}{T_2-T_1+1} \sum_{j=T_1}^{T_2} \left\|\overline{V}^j - \underline{V}^j\right\|_\infty$$

$$\leq \mathrm{Reg}^{T_1:T_2} + \gamma \frac{H+1}{T_2-T_1+1}$$

$$\bullet \left(\frac{1+\gamma}{1-\gamma}\left(\sum_{t=T_1}^{T_2} \mathrm{Reg}^{T_1:t} + \frac{2\gamma}{1+\gamma}\left\|\overline{V}^{T_1} - \underline{V}^{T_1}\right\|_\infty\right) + \left\|\overline{V}^{T_1} - \underline{V}^{T_1}\right\|_\infty\right)$$

$$\leq \mathrm{Reg}^{T_1:T_2} + \frac{2\gamma(H+1)}{(1-\gamma)(T_2-T_1+1)}\left(\sum_{t=T_1}^{T_2} \mathrm{Reg}^{T_1:t} + \left\|\overline{V}^{T_1} - \underline{V}^{T_1}\right\|_\infty\right).$$

$\square$

The next lemma is used to derive Lemma 14.

**Lemma 13** *For any $t \in [T_1:T_2-1]$ and $s \in \mathcal{S}$,*

$$\left\|\underline{q}_s^t - \underline{q}_s^{t+1}\right\|^2 \leq \frac{8B\gamma^2(\alpha_{t-T_1+1})^2}{(1-\gamma)^2} + \frac{2B^2}{(1-\gamma)^2}\left\|y_s^t - y_s^{t+1}\right\|^2$$

$$\left\|\overline{q}_s^t - \overline{q}_s^{t+1}\right\|^2 \leq \frac{8A\gamma^2(\alpha_{t-T_1+1})^2}{(1-\gamma)^2} + \frac{2A^2}{(1-\gamma)^2}\left\|x_s^t - x_s^{t+1}\right\|^2.$$

*Proof of Lemma 13.* By (62) and Fact 2, we have

$$\left\|\underline{q}_s^t - \underline{q}_s^{t+1}\right\|^2 \leq 2B \max_{(a,b)\in\mathcal{A}\times\mathcal{B}}\left|Q_s[\underline{V}^t](a,b) - Q_s[\underline{V}^{t+1}](a,b)\right|^2\left\|y_s^t\right\|_1^2$$

$$+ 2B^2 \max_{(a,b)\in\mathcal{A}\times\mathcal{B}}\left|Q_s[\underline{V}^{t+1}]\right|^2\left\|y_s^t - y_s^{t+1}\right\|^2 \qquad (67)$$

$$\leq 2B\gamma^2\left\|\underline{V}^t - \underline{V}^{t+1}\right\|_\infty^2 + \frac{2B^2}{(1-\gamma)^2}\left\|y_s^t - y_s^{t+1}\right\|^2.$$

By Fact 2, $\left\|\underline{q}_s^t\right\|_\infty \leq \frac{1}{1-\gamma}$. Then, by the definition of $\underline{V}^t$ in (61a), for any $s \in \mathcal{S}$,

$$\left|\underline{V}^{t+1}(s) - \underline{V}^t(s)\right| \leq \left\|\sum_{j=T_1}^{t} \alpha_{t-T_1+1}^{j-T_1+1}\underline{q}_s^j - \sum_{j=T_1}^{t-1}\alpha_{t-T_1}^{j-T_1+1}\underline{q}_s^j\right\|_\infty$$

$$\leq \alpha_{t-T_1+1}^{t-T_1+1}\left\|\underline{q}_s^{t+1}\right\|_\infty + \sum_{j=T_1}^{t-1}\left|\alpha_{t-T_1}^{j-T_1+1} - \alpha_{t-T_1+1}^{j-T_1+1}\right|\left\|\underline{q}_s^j\right\|_\infty$$

$$\leq \frac{1}{1-\gamma}\left(\alpha_{t-T_1+1} + 1 - (1-\alpha_{t-T_1+1})\right)$$

$$\leq \frac{2\alpha_{t-T_1+1}}{1-\gamma},$$

where the third inequality uses the facts that $\sum_{j'=1}^t \alpha_t^{j'} = 1$ and $\alpha_{t+1}^j \leq \alpha_t^j$, $\alpha_t^j \leq \alpha_t$ for any $1 \leq j \leq t$.

Thus,

$$\left\|\underline{V}^{t+1} - \underline{V}^t\right\|_\infty \leq \frac{2\alpha_{t-T_1+1}}{1-\gamma}. \qquad (68)$$

By substituting (68) into (67), we have

$$\left\|\underline{q}_s^t - \underline{q}_s^{t+1}\right\|^2 \leq \frac{8B\gamma^2(\alpha_{t-T_1+1})^2}{(1-\gamma)^2} + \frac{2B^2}{(1-\gamma)^2}\left\|y_s^t - y_s^{t+1}\right\|^2.$$

The bound for $\left\| \overline{\boldsymbol{q}}_s^t - \overline{\boldsymbol{q}}_s^{t+1} \right\|^2$ follows analogously. $\qquad\square$

We bound the regrets in the following lemma. Its proof is mainly from combining standard analysis in RVU property (see for instance Rakhlin & Sridharan (2013); Syrgkanis et al. (2015)) and with Lemma 13.

**Lemma 14** *For any* $t \in [T_1 : T_2]$, *if* $\eta \leq \frac{1-\gamma}{8\sqrt{2}\max\{A,B\}}$, *we have*

$$\operatorname{Reg}^{T_1:t} \leq \frac{136\,(A+B)\,H}{\eta\,(1-\gamma)^2}\alpha_{t-T_1+1}.$$

*Proof of Lemma 14.* Choose an arbitrary point $\boldsymbol{x}_s^*$ from $\Delta_{\mathcal{A}}$. Since $\widetilde{\boldsymbol{x}}_s^{t+1}$ is the projection onto $\Delta_{\mathcal{A}}$, we have

$$\left\langle \boldsymbol{x}_s^* - \widetilde{\boldsymbol{x}}_s^{t+1}, \widetilde{\boldsymbol{x}}_s^{t+1} - \widetilde{\boldsymbol{x}}_s^t + \eta\underline{\boldsymbol{q}}_s^{t+1} \right\rangle \geq 0,\ \forall t \in [T_1 : T_2 - 1].$$

Then, we have

$$\eta\left\langle \widetilde{\boldsymbol{x}}_s^{t+1} - \boldsymbol{x}_s^*, \underline{\boldsymbol{q}}_s^{t+1} \right\rangle \leq \frac{1}{2}\left( \left\| \widetilde{\boldsymbol{x}}_s^t - \boldsymbol{x}_s^* \right\|^2 - \left\| \widetilde{\boldsymbol{x}}_s^{t+1} - \boldsymbol{x}_s^* \right\|^2 - \left\| \widetilde{\boldsymbol{x}}_s^{t+1} - \widetilde{\boldsymbol{x}}_s^t \right\|^2 \right).$$

Analogously,

$$\eta\left\langle \boldsymbol{x}_s^{t+1} - \widetilde{\boldsymbol{x}}_s^{t+1}, \underline{\boldsymbol{q}}_s^t \right\rangle \leq \frac{1}{2}\left( \left\| \widetilde{\boldsymbol{x}}_s^{t+1} - \widetilde{\boldsymbol{x}}_s^t \right\| - \left\| \widetilde{\boldsymbol{x}}_s^{t+1} - \boldsymbol{x}_s^{t+1} \right\|^2 - \left\| \boldsymbol{x}_s^{t+1} - \widetilde{\boldsymbol{x}}_s^t \right\|^2 \right).$$

Then, by combining the above two equations, we have

$$\begin{aligned}
&\eta\left\langle \boldsymbol{x}_s^{t+1} - \boldsymbol{x}_s^*, \underline{\boldsymbol{q}}_s^{t+1} \right\rangle \\
={}&\eta\left\langle \widetilde{\boldsymbol{x}}_s^{t+1} - \boldsymbol{x}_s^*, \underline{\boldsymbol{q}}_s^{t+1} \right\rangle + \eta\left\langle \boldsymbol{x}_s^{t+1} - \widetilde{\boldsymbol{x}}_s^{t+1}, \underline{\boldsymbol{q}}_s^t \right\rangle + \eta\left\langle \boldsymbol{x}_s^{t+1} - \widetilde{\boldsymbol{x}}_s^{t+1}, \underline{\boldsymbol{q}}_s^{t+1} - \underline{\boldsymbol{q}}_s^t \right\rangle \\
\leq{}&\frac{1}{2}\left( \left\| \widetilde{\boldsymbol{x}}_s^t - \boldsymbol{x}_s^* \right\|^2 - \left\| \widetilde{\boldsymbol{x}}_s^{t+1} - \boldsymbol{x}_s^* \right\|^2 - \left\| \widetilde{\boldsymbol{x}}_s^{t+1} - \boldsymbol{x}_s^{t+1} \right\|^2 - \left\| \boldsymbol{x}_s^{t+1} - \widetilde{\boldsymbol{x}}_s^t \right\|^2 \right) \\
&+ \eta\left\langle \boldsymbol{x}_s^{t+1} - \widetilde{\boldsymbol{x}}_s^{t+1}, \underline{\boldsymbol{q}}_s^{t+1} - \underline{\boldsymbol{q}}_s^t \right\rangle \\
\leq{}&\frac{1}{2}\left( \left\| \widetilde{\boldsymbol{x}}_s^t - \boldsymbol{x}_s^* \right\|^2 - \left\| \widetilde{\boldsymbol{x}}_s^{t+1} - \boldsymbol{x}_s^* \right\|^2 \right) + \Delta_x^{t+1},
\end{aligned}$$

where

$$\Delta_x^{t+1} = -\frac{1}{4}\left\| \widetilde{\boldsymbol{x}}_s^{t+1} - \boldsymbol{x}_s^{t+1} \right\|^2 - \frac{1}{2}\left\| \boldsymbol{x}_s^{t+1} - \widetilde{\boldsymbol{x}}_s^t \right\|^2 + 4\eta^2\left\| \underline{\boldsymbol{q}}_s^{t+1} - \underline{\boldsymbol{q}}_s^t \right\|^2.$$

By taking sum on both sides of the above equation, we have

$$\begin{aligned}
&\eta\sum_{t=T_1}^{T_2}\alpha_{T_2-T_1+1}^{t-T_1+1}\left\langle \boldsymbol{x}_s^t - \boldsymbol{x}_s^*, \underline{\boldsymbol{q}}_s^t \right\rangle \\
\leq{}&\frac{\alpha_{T_2-T_1+1}^1}{2}\left\| \boldsymbol{x}_s^{T_1} - \boldsymbol{x}_s^* \right\|_1\left\| \underline{\boldsymbol{q}}_s^{T_1} \right\|_\infty + \frac{\alpha_{T_2-T_1+1}^2}{2}\left\| \widetilde{\boldsymbol{x}}^{T_1} - \boldsymbol{x}_s^* \right\|^2 \\
&+ \sum_{t=T_1+1}^{T_2-1}\frac{\alpha_{T_2-T_1+1}^{t-T_1+2} - \alpha_{T_2-T_1+1}^{t-T_1+1}}{2}\left\| \widetilde{\boldsymbol{x}}^t - \boldsymbol{x}_s^* \right\|^2 + \sum_{t=T_1}^{T_2-1}\alpha_{T_2-T_1+1}^{t-T_1+2}\Delta_x^{t+1} \\
\leq{}&\frac{\alpha_{T_2-T_1+1}^1}{1-\gamma} + \alpha_{T_2-T_1+1}^2 + \sum_{t=T_1+1}^{T_2-1}\left( \alpha_{T_2-T_1+1}^{t-T_1+2} - \alpha_{T_2-T_1+1}^{t-T_1+1} \right) + \sum_{t=T_1}^{T_2-1}\alpha_{T_2-T_1+1}^{t-T_1+2}\Delta_x^{t+1} \\
\leq{}&\frac{\alpha_{T_2-T_1+1}}{1-\gamma} + 2\alpha_{T_2-T_1+1} + \sum_{t=T_1}^{T_2-1}\alpha_{T_2-T_1+1}^{t-T_1+2}\Delta_x^{t+1}.
\end{aligned}$$
(69)

Analogously, for any $\boldsymbol{y}_s^* \in \Delta_{\mathcal{B}}$,

$$\eta \sum_{t=T_1}^{T_2} \alpha_{T_2-T_1+1}^{t-T_1+1} \left\langle \boldsymbol{y}_s^t - \boldsymbol{y}_s^*, \overline{\boldsymbol{q}}_s^t \right\rangle \le \frac{\alpha_{T_2-T_1+1}}{1-\gamma} + 2\alpha_{T_2-T_1+1} + \sum_{t=T_1}^{T_2-1} \alpha_{T_2-T_1+1}^{t-T_1+2} \Delta_y^{t+1}, \qquad (70)$$

where

$$\Delta_y^{t+1} = -\frac{1}{4} \left\| \widetilde{\boldsymbol{y}}_s^{t+1} - \boldsymbol{y}_s^{t+1} \right\|^2 - \frac{1}{2} \left\| \boldsymbol{y}_s^{t+1} - \widetilde{\boldsymbol{y}}_s^t \right\|^2 + 4\eta^2 \left\| \overline{\boldsymbol{q}}_s^{t+1} - \overline{\boldsymbol{q}}_s^t \right\|^2.$$

Since $H \ge 1$, we have $\alpha_{T_2-T_1+1}^{t-T_1+2} / \alpha_{T_2-T_1+1}^{t-T_1+1} \le 2$. Then, by combining with the condition on $\eta$ and the fact that $\left\| \boldsymbol{x}^{t+1} - \boldsymbol{x}^t \right\|^2 \le 2 \left\| \boldsymbol{x}^{t+1} - \widetilde{\boldsymbol{x}}^t \right\|^2 + 2 \left\| \widetilde{\boldsymbol{x}}^t - \boldsymbol{x}^t \right\|^2$, we have

$$-\frac{\alpha_{T_2-T_1+1}^{t-T_1+2}}{2} \left\| \boldsymbol{x}^{t+1} - \widetilde{\boldsymbol{x}}^t \right\|^2 - \frac{\alpha_{T_2-T_1+1}^{t-T_1+1}}{4} \left\| \widetilde{\boldsymbol{x}}^t - \boldsymbol{x}^t \right\|^2$$
$$+ \frac{8\alpha_{T_2-T_1+1}^{t-T_1+2} \max \left\{ A^2, B^2 \right\} \eta^2}{(1-\gamma)^2} \left\| \boldsymbol{x}^{t+1} - \boldsymbol{x}^t \right\|^2 \qquad (71)$$
$$\le -\frac{\alpha_{T_2-T_1+1}^{t-T_1+2}}{16} \left( -2 \left\| \boldsymbol{x}^{t+1} - \widetilde{\boldsymbol{x}}^t \right\|^2 - 2 \left\| \widetilde{\boldsymbol{x}}^t - \boldsymbol{x}^t \right\|^2 + \left\| \boldsymbol{x}^{t+1} - \boldsymbol{x}^t \right\|^2 \right) \le 0.$$

Then, by combining the definitions of $\Delta_x^{t+1}$ and $\Delta_y^{t+1}$ with Lemma 13, we have

$$\sum_{t=T_1}^{T_2-1} \alpha_{T_2-T_1+1}^{t-T_1+2} \left( \Delta_x^{t+1} + \Delta_y^{t+1} \right)$$
$$\le \frac{8\alpha_{T_2-T_1+1} \max \left\{ A^2, B^2 \right\} \eta^2}{(1-\gamma)^2} \left( \left\| \boldsymbol{x}^{T_1+1} - \boldsymbol{x}^{T_1} \right\|^2 + \left\| \boldsymbol{y}^{T_1+1} - \boldsymbol{y}^{T_1} \right\|^2 \right)$$
$$+ \sum_{t=T_1}^{T_2-1} \alpha_{T_2-T_1+1}^{t-T_1+2} \frac{32 \left( A + B \right) \gamma^2 \left( \alpha_{t-T_1+1} \right)^2}{(1-\gamma)^2} \qquad (72)$$
$$\le 2\alpha_{T_2-T_1+1} + \frac{32 \left( A + B \right) \gamma^2}{(1-\gamma)^2} \sum_{t=T_1}^{T_2-1} \alpha_{T_2-T_1+1} \left( \frac{H+1}{H+t-T_1+1} \right)^2$$
$$\le \left( 2 + \frac{32 \left( A + B \right) \gamma^2}{(1-\gamma)^2} \cdot \frac{(H+1)^2}{H} \right) \alpha_{T_2-T_1+1},$$

where the first inequality also uses (71) and the max-player's counterpart of (71), the second inequality is by the condition on $\eta$ and Fact 1.

By combining (69), (70), (72),

$$\text{Reg}^{T_1:T_2} \le \frac{1}{\eta} \left( \frac{2}{1-\gamma} + 6 + \frac{32 \left( A + B \right) \gamma^2}{(1-\gamma)^2} \cdot \frac{(H+1)^2}{H} \right) \alpha_{T_2-T_1+1}$$
$$\le \frac{136 \left( A + B \right) H}{\eta (1-\gamma)^2} \alpha_{T_2-T_1+1}.$$

The bound of $\text{Reg}^{T_1:t}$ for $t \in [T_1 : T_2]$ follows by similar arguments. $\qquad \square$

Now, we can prove the global convergence of Averaging OGDA (Theorem 3) by combining Lemma 11, Lemma 12 and Lemma 14.

**Proof of Theorem 3.** By Lemma 12, Lemma 14, we have

$$\left\| \overline{V}^{T_2+1} - \underline{V}^{T_2+1} \right\|_\infty$$
$$\le \frac{136 \left( A + B \right) H}{\eta (1-\gamma)^2} \alpha_{T_2-T_1+1}$$
$$+ \frac{2\gamma (H+1)}{(1-\gamma)(T_2-T_1+1)} \left( \sum_{t=T_1}^{T_2} \frac{136 \left( A + B \right) H}{\eta (1-\gamma)^2} \alpha_{t-T_1+1} + \left\| \overline{V}^{T_1} - \underline{V}^{T_1} \right\|_\infty \right).$$

Since $\sum_{t=T_1}^{T_2} \alpha_{t-T_1+1} \le \frac{(H+1)\log(T_2-T_1+1)}{T_2-T_1+1}$, we have

$$\left\| \overline{V}^{T_2+1} - \underline{V}^{T_2+1} \right\|_\infty \le \frac{408(H+1)^3 (A+B) \log(T_2 - T_1 + 1)}{\eta (1-\gamma)^3 (T_2 - T_1 + 1)} + \frac{2\gamma(H+1)}{(1-\gamma)^2 (T_2 - T_1 + 1)}.$$

By Lemma 11, we have

$$\mathrm{dist}\left( \widehat{z}^{[T_1:T_2]}, \mathcal{Z}^* \right)$$

$$\le \widehat{C} \cdot \left( \frac{408(H+1)^3 (A+B) \log(T_2 - T_1 + 1)}{\eta(1-\gamma)^3 (T_2 - T_1 + 1)} + \frac{2\gamma(H+1)}{(1-\gamma)^2 (T_2 - T_1 + 1)} \right)$$

$$\le \frac{C' \log(T_2 - T_1 + 1)}{\eta(T_2 - T_1 + 1)},$$

where

$$C' = \frac{3280 \widehat{C}(A+B)}{(1-\gamma)^6} = \frac{3280 \sqrt{S}(A+B)}{c_+ (1-\gamma)^6}.$$

$\square$

**Remark 1** *The initialization $\underline{V}^{T_1} = V^{\dagger, y^{T_1}}$ and $\overline{V}^{T_1} = V^{x^{T_1}, \dagger}$ is only used to show the geometric boundedness in Theorem 4. When Averaging OGDA is used independently rather than called in* `Homotopy-PO` *(Algorithm 1), we can simply choose $\underline{V}^{T_1}(s) = 0$ and $\overline{V}^{T_1} = \frac{1}{1-\gamma}$ for any $s \in \mathcal{S}$. The global convergence rate in Theorem 3 still holds.*

### D.2 GEOMETRIC BOUNDEDNESS OF AVERAGING OGDA

In this section, we prove the geometric boundedness of Averaging OGDA (Theorem 4).

The geometric boundedness of averaging OGDA essentially relies on the stability of projected gradient descent/ascent characterized in Lemma 2. Intuitively, when $\{z^j\}_{j \in [T_1:t]}$ are close to the Nash equilibrium set, $\left\{ \underline{V}^j(s), \overline{V}^j(s) \right\}_{j \in [T_1:t]}$ will be close to $v^*(s)$. Thus, $\min_a \underline{q}_s^t(a)$, $\max_b \overline{q}_s^t(b)$ will also be close to $v^*(s)$. Then, by Lemma 2, $z^{t+1}$ will not be far away from the Nash equilibrium set.

Our proofs in this section can be summarized as: providing mutual bounds among $\{\mathrm{dist}(z^t, \mathcal{Z}^*)\}$, $\left\{ \mathrm{dist}\left( \widetilde{z}^t, \mathcal{Z}^* \right) \right\}$, $\left\{ \left\| \overline{V}^t - \underline{V}^t \right\|_\infty \right\}$, $\left\{ \max_b \overline{q}_s^t(b) - \min_a \underline{q}_s^t(a) \right\}$ by induction.

The following fact shows that $\left\| \overline{V}^{T_1} - \underline{V}^{T_1} \right\|_\infty$ can be bounded by $\mathrm{dist}\left( z^{T_1}, \mathcal{Z}^* \right)$.

**Lemma 15** *The approximate value functions $\underline{V}^{T_1}, \overline{V}^{T_1}$ satisfy*

$$\left\| \overline{V}^{T_1} - \underline{V}^{T_1} \right\|_\infty \le \frac{\max\left\{ \sqrt{2A}, \sqrt{2B} \right\}}{(1-\gamma)^2} \mathrm{dist}\left( z^{T_1}, \mathcal{Z}^* \right).$$

*Proof of Lemma 15.* By Fact 2, $\underline{V}^{T_1}(s) \le v^*(s) \le \overline{V}^{T_1}(s)$. By Lemma 19, $V^{\dagger, y^{t^*}}(s) = v^*(s)$. Since the min-player initializes $\underline{V}^{T_1}(s) = V^{\dagger, y^{T_1}}(s)$, by combining with Lemma 21, we have

$$v^*(s) - \underline{V}^{T_1}(s) = V^{\dagger, y^{t^*}}(s) - V^{\dagger, y^{T_1}}(s) \le \frac{\sqrt{B} \left\| y^{T_1} - y^{T_1 *} \right\|}{(1-\gamma)^2} \le \frac{\sqrt{B} \mathrm{dist}\left( y^{T_1}, \mathcal{Z}^* \right)}{(1-\gamma)^2}.$$

Analogously,

$$\overline{V}^{T_1}(s) - v^*(s) \le \frac{\sqrt{A} \mathrm{dist}\left( x^{T_1}, \mathcal{X}^* \right)}{(1-\gamma)^2}.$$

The result follows by summing the above two equations and combining with the fact that $\text{dist}(\boldsymbol{z}^{T_1}, \mathcal{Z}^*) \leq \sqrt{2}\text{dist}(\boldsymbol{x}^{T_1}, \mathcal{X}^*) + \sqrt{2}\text{dist}(\boldsymbol{y}^{T_1}, \mathcal{Y}^*)$. $\qquad\square$

The following lemma follows directly by the definition of $\underline{V}^t$, $\overline{V}^t$ in (61a), (63a) and the fact that $\sum_{j=1}^t \alpha_t^j = 1$.

**Lemma 16** *For any $t \in [T_1 : T_2 - 1]$ and $s \in \mathcal{S}$*

$$\overline{V}^{t+1}(s) - \underline{V}^{t+1}(s) \leq \max_{j \in [T_1:t]} \left( \max_{b \in \mathcal{B}} \overline{\boldsymbol{q}}_s^j(b) - \min_{a \in \mathcal{A}} \underline{\boldsymbol{q}}_s^j(a) \right).$$

The following lemma bound the expansion of $\text{dist}(\boldsymbol{z}^t, \mathcal{Z}^*)$. Its proof mainly uses Lemma 2.

**Lemma 17** *For any $t \in [T_1 + 1 : T_2 - 1]$, we have*

$$\text{dist}^2\left(\widetilde{\boldsymbol{z}}^t, \mathcal{Z}^*\right) \leq 18\text{dist}^2\left(\widetilde{\boldsymbol{z}}^{t-1}, \mathcal{Z}^*\right) + 8\eta^2 S \max\{A, B\} \left\|\overline{V}^t - \underline{V}^t\right\|_\infty^2$$
$$+ 8\eta^2 \frac{\max\{A, B\}^2}{(1-\gamma)^2} \text{dist}^2\left(\boldsymbol{z}^t, \mathcal{Z}^*\right),$$

$$\text{dist}^2\left(\boldsymbol{z}^{t+1}, \mathcal{Z}^*\right) \leq 324\text{dist}^2\left(\widetilde{\boldsymbol{z}}^{t-1}, \mathcal{Z}^*\right) + 152\eta^2 S \max\{A, B\} \left\|\overline{V}^t - \underline{V}^t\right\|_\infty^2$$
$$+ 152\eta^2 \frac{\max\{A, B\}^2}{(1-\gamma)^2} \text{dist}^2\left(\boldsymbol{z}^t, \mathcal{Z}^*\right).$$

*In addition,*

$$\text{dist}^2\left(\boldsymbol{z}^{T_1+1}, \mathcal{Z}^*\right) \leq \left(8 + \frac{8\eta^2 S \max\{A^2, B^2\}}{(1-\gamma)^4} + \frac{4\eta^2 \max\{A, B\}^2}{(1-\gamma)^2}\right)\text{dist}^2\left(\boldsymbol{z}^{T_1}, \mathcal{Z}^*\right).$$

*Proof of Lemma 17.* By Fact 2, we have $\left\|\underline{V}^t - v^*\right\|_\infty^2 + \left\|\overline{V}^t - v^*\right\|_\infty^2 \leq \left\|\overline{V}^t - \underline{V}^t\right\|_\infty^2$. Then,

$$B \max_{(a,b) \in \mathcal{A} \times \mathcal{B}} \left|\boldsymbol{Q}_s[\underline{V}^t](a, b) - \boldsymbol{Q}_s^*(a, b)\right|^2 + A \max_{(a,b) \in \mathcal{A} \times \mathcal{B}} \left|\boldsymbol{Q}_s[\overline{V}^t](a, b) - \boldsymbol{Q}_s^*(a, b)\right|^2$$
$$\leq \gamma^2 \max\{A, B\} \left(\left\|\underline{V}^t - v^*\right\|_\infty^2 + \left\|\overline{V}^t - v^*\right\|_\infty^2\right) \leq \max\{A, B\} \left\|\overline{V}^t - \underline{V}^t\right\|_\infty^2.$$

Then, by Lemma 2, we have

$$\left\|\widetilde{\boldsymbol{z}}^t - \widetilde{\boldsymbol{z}}^{t-1}\right\|^2 \leq 8\text{dist}^2\left(\widetilde{\boldsymbol{z}}^{t-1}, \mathcal{Z}^*\right) + 4\eta^2 S \max\{A, B\} \left\|\overline{V}^t - \underline{V}^t\right\|_\infty^2$$
$$+ 4\eta^2 \frac{\max\{A, B\}^2}{(1-\gamma)^2} \text{dist}^2\left(\boldsymbol{z}^t, \mathcal{Z}^*\right).$$

$$\left\|\boldsymbol{z}^{t+1} - \widetilde{\boldsymbol{z}}^t\right\|^2 \leq 8\text{dist}^2\left(\widetilde{\boldsymbol{z}}^t, \mathcal{Z}^*\right) + 4\eta^2 S \max\{A, B\} \left\|\overline{V}^t - \underline{V}^t\right\|_\infty^2$$
$$+ 4\eta^2 \frac{\max\{A, B\}^2}{(1-\gamma)^2} \text{dist}^2\left(\boldsymbol{z}^t, \mathcal{Z}^*\right).$$

$$\left\|\boldsymbol{z}^{T_1+1} - \boldsymbol{z}^{T_1}\right\|^2 \leq 8\text{dist}^2\left(\boldsymbol{z}^{T_1}, \mathcal{Z}^*\right) + 4\eta^2 S \max\{A, B\} \left\|\overline{V}^{T_1} - \underline{V}^{T_1}\right\|_\infty^2$$
$$+ 4\eta^2 \frac{\max\{A, B\}^2}{(1-\gamma)^2} \text{dist}^2\left(\boldsymbol{z}^{T_1}, \mathcal{Z}^*\right).$$

The bound of $\text{dist}^2(\widetilde{\boldsymbol{z}}^t, \mathcal{Z}^*)$ follows by the fact that $\text{dist}^2\left(\widetilde{\boldsymbol{z}}^t, \mathcal{Z}^*\right) \leq 2\text{dist}^2\left(\widetilde{\boldsymbol{z}}^{t-1}, \mathcal{Z}^*\right) + 2\left\|\widetilde{\boldsymbol{z}}^t - \widetilde{\boldsymbol{z}}^{t-1}\right\|^2$. The bound of $\text{dist}^2(\boldsymbol{z}^{t+1}, \mathcal{Z}^*)$ follows by the fact that $\text{dist}^2\left(\boldsymbol{z}^{t+1}, \mathcal{Z}^*\right) \leq$

$2\mathrm{dist}^2\left(\widetilde{\boldsymbol{z}}^t, \mathcal{Z}^*\right) + 2\left\|\boldsymbol{z}^{t+1} - \widetilde{\boldsymbol{z}}^t\right\|^2$. The bound of $\mathrm{dist}^2(\boldsymbol{z}^{T_1+1}, \mathcal{Z}^*)$ follows by combining with Lemma 15. $\qquad\square$

The following lemma is straightforward from the definitions of $\underline{\boldsymbol{q}}_s^t$ and $\overline{\boldsymbol{q}}_s^t$.

**Lemma 18** *For any $t \in [T_1 : T_2]$ and $s \in \mathcal{S}$,*

$$\max_{b \in \mathcal{B}} \overline{\boldsymbol{q}}_s^t(b) - \min_{a \in \mathcal{A}} \underline{\boldsymbol{q}}_s^t(a) \le \left\|\overline{V}^t - \underline{V}^t\right\|_\infty + \frac{\max\left\{\sqrt{2A}, \sqrt{2B}\right\}}{1 - \gamma} \mathrm{dist}\left(\boldsymbol{z}_s^t, \mathcal{Z}_s^*\right).$$

*Proof of Lemma 18.* For any $s \in \mathcal{S}$, we have

$$v^*(s) - \min_{a \in \mathcal{A}} \underline{\boldsymbol{q}}_s^t(a) = \min_{a \in \mathcal{A}}\left(\boldsymbol{Q}_s[v^*]\boldsymbol{y}_s^{t*}\right)(a) - \min_{a \in \mathcal{A}}(\boldsymbol{Q}_s[\underline{V}^t]\boldsymbol{y}_s^t)(a)$$

$$\le \left\|\boldsymbol{Q}_s[v^*]\boldsymbol{y}_s^{t*} - \boldsymbol{Q}_s[\underline{V}^t]\boldsymbol{y}_s^t\right\|_\infty$$

$$\le \max_{(a,b) \in \mathcal{A} \times \mathcal{B}}\left|\boldsymbol{Q}_s[v^*](a,b) - \boldsymbol{Q}_s[\underline{V}^t](a,b)\right| \left\|\boldsymbol{y}_s^{t*}\right\|_1 + \max_{(a,b) \in \mathcal{A} \times \mathcal{B}}\left|\boldsymbol{Q}_s[\underline{V}^t]\right| \left\|\boldsymbol{y}_s^t - \boldsymbol{y}_s^{t*}\right\|_1$$

$$\le \left\|v^* - \underline{V}^t\right\|_\infty + \frac{\sqrt{B}}{1 - \gamma} \mathrm{dist}\left(\boldsymbol{y}_s^t, \mathcal{Y}_s^*\right).$$

Analogously,

$$\max_{b \in \mathcal{B}} \overline{\boldsymbol{q}}_s^t(b) - v^*(s) \le \left\|\overline{V}^t - v^*\right\|_\infty + \frac{\sqrt{A}}{1 - \gamma} \mathrm{dist}\left(\boldsymbol{x}_s^t, \mathcal{X}_s^*\right).$$

Then, the proof is completed by combining the above two equations with the facts that $\mathrm{dist}(\boldsymbol{z}_s, \mathcal{Z}_s^*) \le \sqrt{2}\mathrm{dist}(\boldsymbol{x}_s, \mathcal{X}^*) + \sqrt{2}\mathrm{dist}(\boldsymbol{y}_s, \mathcal{Y}^*)$. $\qquad\square$

Now, we can prove the geometric boundedness of Averaging OGDA (Theorem 4) by combining Lemma 16, Lemma 17, Lemma 18 inductively.

**Proof of Theorem 4.** By Lemma 15,

$$\left\|\overline{V}^{T_1} - \underline{V}^{T_1}\right\|_\infty \le C_1 \mathrm{dist}\left(\boldsymbol{z}^{T_1}, \mathcal{Z}^*\right), \tag{73}$$

where

$$C_1 = \frac{\max\left\{\sqrt{2A}, \sqrt{2B}\right\}}{(1 - \gamma)^2}.$$

By Lemma 18,

$$\max_{s \in \mathcal{S}}\left(\max_{b \in \mathcal{B}} \overline{\boldsymbol{q}}_s^t(b) - \min_{a \in \mathcal{A}} \underline{\boldsymbol{q}}_s^t(a)\right) \le \left\|\overline{V}^t - \underline{V}^t\right\|_\infty + C_2 \mathrm{dist}\left(\boldsymbol{z}_s^{T_1}, \mathcal{Z}_s^*\right), \tag{74}$$

where

$$C_2 = \frac{\max\left\{\sqrt{2A}, \sqrt{2B}\right\}}{1 - \gamma}.$$

By Lemma 17 and the fact that $\sqrt{A_1 + A_2 + A_3} \le \sqrt{A_1} + \sqrt{A_2} + \sqrt{A_3}$, we have

$$\mathrm{dist}\left(\boldsymbol{z}^{T_1+1}, \mathcal{Z}^*\right) \le D_1 \mathrm{dist}\left(\boldsymbol{z}^{T_1}, \mathcal{Z}^*\right), \tag{75}$$

$$\mathrm{dist}\left(\widetilde{\boldsymbol{z}}^{t+1}, \mathcal{Z}^*\right) \le D_2 \mathrm{dist}\left(\widetilde{\boldsymbol{z}}^t, \mathcal{Z}^*\right) + C_3 \left\|\overline{V}^{t+1} - \underline{V}^{t+1}\right\|_\infty + C_4 \mathrm{dist}\left(\boldsymbol{z}^{t+1}, \mathcal{Z}^*\right), \tag{76}$$

$$\mathrm{dist}\left(\boldsymbol{z}^{t+2}, \mathcal{Z}^*\right) \le D_3 \mathrm{dist}\left(\widetilde{\boldsymbol{z}}^t, \mathcal{Z}^*\right) + C_5 \left\|\overline{V}^{t+1} - \underline{V}^{t+1}\right\|_\infty + C_6 \mathrm{dist}\left(\boldsymbol{z}^{t+1}, \mathcal{Z}^*\right)^2, \tag{77}$$

where

$$D_1 = \sqrt{8 + \frac{8\eta^2 S \max\{A^2, B^2\}}{(1-\gamma)^4} + \frac{4\eta^2 \max\{A, B\}^2}{(1-\gamma)^2}},$$

$$D_2 = \sqrt{18}, \ C_3 = \eta\sqrt{8S \max\{A, B\}}, \ C_4 = \frac{\sqrt{8}\eta \max\{A, B\}}{1-\gamma},$$

$$D_3 = \sqrt{324}, \ C_5 = \eta\sqrt{152S \max\{A, B\}}, \ C_6 = \frac{\sqrt{152}\eta \max\{A, B\}}{1-\gamma}.$$

Define

$$\widetilde{D} = \max\{D_1, C_1 + C_2, 1 + C_2, D_2 + C_3 + C_4, D_3 + C_5 + C_6\}.$$

Next, we prove (78) by induction

$$\max\left\{\text{dist}\left(\boldsymbol{z}_s^{j+1}, \mathcal{Z}_s^*\right), \text{dist}\left(\widetilde{\boldsymbol{z}}_s^j, \mathcal{Z}_s^*\right), \left\|\overline{V}^j - \underline{V}^j\right\|_\infty, \max_{s \in \mathcal{S}}\left(\max_{b \in \mathcal{B}} \overline{\boldsymbol{q}}_s^j(b) - \min_{a \in \mathcal{A}} \underline{\boldsymbol{q}}_s^j(a)\right)\right\} \quad (78)$$

$$\leq \widetilde{D}^{j-T_1+1} \cdot \text{dist}\left(\boldsymbol{z}^{T_1}, \mathcal{Z}^*\right).$$

The case of $j = T_1$ follows by (73), (74), (75).

Now, suppose that we have shown (78) for $j \in [T_1 : t]$. Then, by Lemma 16 and the induction hypothesis (78),

$$\left\|\overline{V}^{t+1} - \underline{V}^{t+1}\right\|_\infty \leq \widetilde{D}^{t-T_1+1} \cdot \text{dist}\left(\boldsymbol{z}^{T_1}, \mathcal{Z}^*\right).$$

By combining the above equation with (74) and the induction hypothesis (78),

$$\max_{s \in \mathcal{S}}\left(\max_{b \in \mathcal{B}} \overline{\boldsymbol{q}}_s^{t+1}(b) - \min_{a \in \mathcal{A}} \underline{\boldsymbol{q}}_s^{t+1}(a)\right) \leq \left\|\overline{V}^t - \underline{V}^t + C_2\text{dist}\left(\boldsymbol{z}_s^t, \mathcal{Z}_s^*\right)\right\|_\infty$$

$$\leq (1 + C_2)\widetilde{D}^{t-T_1+1}\text{dist}\left(\boldsymbol{z}^{T_1}, \mathcal{Z}^*\right).$$

By combining the above two equations with (76), (77) and the induction hypothesis (78),

$$\text{dist}\left(\widetilde{\boldsymbol{z}}^{t+1}, \mathcal{Z}^*\right) \leq (D_2 + C_3 + C_4)\widetilde{D}^{t-T_1+1}\text{dist}\left(\boldsymbol{z}^{T_1}, \mathcal{Z}^*\right),$$

$$\text{dist}\left(\boldsymbol{z}^{t+2}, \mathcal{Z}^*\right) \leq (D_3 + C_5 + C_6)\widetilde{D}^{t-T_1+1}\text{dist}\left(\boldsymbol{z}^{T_1}, \mathcal{Z}^*\right).$$

By the definition of $\widetilde{D}$, we have proved (78) for $t+1$. By induction, (78) holds for any $t \in [T_1 : T_2]$. The following relation is implied by (78) directly

$$\text{dist}\left(\boldsymbol{z}^t, \mathcal{Z}^*\right) \leq \widetilde{D}^{t-T_1} \cdot \text{dist}\left(\boldsymbol{z}^{T_1}, \mathcal{Z}^*\right) = \widetilde{D}^{t-T_1} \cdot \text{dist}\left(\tilde{\boldsymbol{z}}, \mathcal{Z}^*\right), \quad (79)$$

where $\tilde{\boldsymbol{z}} = (\tilde{\boldsymbol{x}}, \tilde{\boldsymbol{y}})$ is the initial policy pair (59), (60).

Then, (11) follows by setting $D_0 = \widetilde{D}^2$.

By definition, we have $D_0 = O(S(A + B)^2/(1 - \gamma)^4)$ under the condition $\eta \leq 1$.

By Shapley's theorem (Lemma 19), $\mathcal{Z}_s^* = \mathcal{X}_s^* \times \mathcal{Y}_s^*$ is the set of Nash equilibria of a matrix game. Thus, $\mathcal{Z}_s^*$ is convex, then, $\mathcal{Z}^*$ is also convex. Thus, we have

$$\text{dist}\left(\sum_{t=T_1}^{T_2} \alpha_{T_2-T_1+1}^{t-T_1+1}\boldsymbol{z}^t, \mathcal{Z}^*\right) \leq \sum_{t=T_1}^{T_2} \alpha_{T_2-T_1+1}^{t-T_1+1}\text{dist}\left(\boldsymbol{z}^t, \mathcal{Z}^*\right).$$

As $D_0 \geq 1$ in our definition, we have

$$\text{dist}\left(\widehat{\boldsymbol{z}}^{[T_1:T_2]}, \mathcal{Z}^*\right) \leq \text{dist}\left(\sum_{t=T_1}^{T_2} \alpha_{T_2-T_1+1}^{t-T_1+1}\boldsymbol{z}^t, \mathcal{Z}^*\right) \leq \sum_{t=T_1}^{T_2} \alpha_{T_2-T_1+1}^{t-T_1+1}\text{dist}\left(\boldsymbol{z}^t, \mathcal{Z}^*\right)$$

$$\leq \left(\sqrt{D_0}\right)^{T_2-T_1}\text{dist}\left(\boldsymbol{z}^{T_1}, \mathcal{Z}^*\right) = \left(\sqrt{D_0}\right)^{T_2-T_1}\text{dist}\left(\tilde{\boldsymbol{z}}, \mathcal{Z}^*\right).$$

This gives (12). $\qquad\square$

## E  PROOFS FOR GLOBAL LINEAR CONVERGENCE

**Proof of Theorem 1.**  Recall the constants $c_0, \delta_0$ defined in the local linear convergence of `Local-Fast`, $D_0$ defined in the geometric boundedness of `Global-Slow`, $C'$ defined in the global convergence of `Global-Slow` in Section 3.1.

Define

$$M_1^* = \min \left\{ t \geq 1 : \frac{C' \log(t)}{\eta' t} \leq \sqrt{\delta_0 \eta^4} \right\},$$

$$M_2^* = \max \left\{ \frac{3}{c_0 \eta^2} \lceil \log \Gamma_0 \rceil, 0 \right\} + 1,$$

$$M_3^* = \frac{6}{c_0 \eta^2} \left( \lceil \log \max \{D_0, 1\} \rceil + 1 \right).$$

Let $M^* = \max \left\{ (M_1^*)^2, M_2^*, (M_3^*)^2 \right\}$. Then, the order of $M^*$

$$M^* \leq O \left( \frac{C'^2 \log^2(C'/(\delta_0 \eta \eta'))}{\delta_0 \eta^4 \eta'^2} + \frac{\log^2(D_0 + 1)}{c_0^2 \eta^4} \right). \tag{80}$$

For simplicity we denote

$$\widehat{z}^k = \widehat{z}^{[\mathcal{I}_{gs}^k : \widetilde{\mathcal{I}}_{gs}^k]}.$$

Note that $\widehat{z}^k = \widehat{z}^{[\mathcal{I}_{gs}^k : \widetilde{\mathcal{I}}_{gs}^k]}$ is the initial policy pair of the $k$-th call to `Local-Fast`.

Define $k^*$ as

$$k^* = \min \left\{ k \in \mathbb{Z}_+ : 2^k \geq M_1^*, 4^k \geq M_2^*, 2^k \geq M_3^* \right\}.$$

Then, $2^{k^*-1} \leq M_1^*, 4^{k^*-1} \leq M_2^*, 2^{k^*-1} \leq M_3^*$, i.e.,

$$4^{k^*} \leq 4 \max \left\{ (M_1^*)^2, M_2^*, (M_3^*)^2 \right\} = 4M^*. \tag{81}$$

Firstly, we provide bounds for $\widehat{z}^k$ after $k \geq k^*$.

For any $k \geq k^*$, since $\widetilde{\mathcal{I}}_{gs}^k - \mathcal{I}_{gs}^k + 1 = 2^k \geq 2^{k^*} \geq M_1^*$, by (2) and the definition of $M_1^*$, the policy pair $\widehat{z}^k$ satisfies

$$\text{dist}^2 \left( \widehat{z}^k, \mathcal{Z}^* \right) \leq \left( \frac{C' \log(2^{k^*})}{\eta' \cdot 2^{k^*}} \right)^2 \leq \delta_0 \eta^4.$$

Since $\widehat{z}^k$ is the initial policy pair of `Local-Fast` in time interval $[\mathcal{I}_{lf}^k : \widetilde{\mathcal{I}}_{lf}^k]$, by (5), for $t \in [\mathcal{I}_{lf}^k : \widetilde{\mathcal{I}}_{lf}^k]$,

$$\text{dist}^2 \left( z^t, \mathcal{Z}^* \right) \leq \Gamma_0 \cdot \left( 1 - c_0 \eta^2 \right)^{t - \mathcal{I}_{lf}^k} \text{dist}^2 \left( \widehat{z}^k, \mathcal{Z}^* \right).$$

Since $4^k \geq 4^{k^*} \geq M_2^*$,

$$\Gamma_0 \cdot \left( 1 - \frac{c_0 \eta^2}{3} \right)^{4^k - 1} \leq 1.$$

Since $2^k \geq 2^{k^*} \geq M_3^*$, we have

$$\left( 1 - \frac{c_0 \eta^2}{3} \right)^{4^k - 1} \leq \left( 1 - \frac{c_0 \eta^2}{3} \right)^{2^{k+1} \cdot \left( 2^{k-1} - 1 \right)} \leq \frac{1}{\max \{D_0, 1\}^{2^{k+1}}}.$$

Then, by combining the above three equations,

$$\operatorname{dist}^2\left(z^{\widetilde{\mathcal{I}}_{\mathrm{lf}}^k}, \mathcal{Z}^*\right) \leq \Gamma_0 \cdot \left(1 - c_0\eta^2\right)^{\widetilde{\mathcal{I}}_{\mathrm{lf}}^k - \mathcal{I}_{\mathrm{lf}}^k} \operatorname{dist}^2\left(\widehat{z}^k, \mathcal{Z}^*\right)$$

$$= \Gamma_0 \cdot \left(1 - c_0\eta^2\right)^{4^k-1} \operatorname{dist}^2\left(\widehat{z}^k, \mathcal{Z}^*\right) \leq \Gamma_0 \cdot \left(1 - \frac{c_0\eta^2}{3}\right)^{3 \cdot \left(4^k-1\right)} \operatorname{dist}^2\left(\widehat{z}^k, \mathcal{Z}^*\right) \tag{82}$$

$$\leq \frac{1}{\max\{D_0, 1\}^{2^{k+1}}} \left(1 - \frac{c_0\eta^2}{3}\right)^{4^k-1} \operatorname{dist}^2\left(\widehat{z}^k, \mathcal{Z}^*\right).$$

By (4) and the fact that $z^{\widetilde{\mathcal{I}}_{\mathrm{lf}}^k}$ is the initial policy pair of the $(k+1)$-th call to `Global-Slow`,

$$\operatorname{dist}^2\left(\widehat{z}^{k+1}, \mathcal{Z}^*\right) \leq D_0^{\widetilde{\mathcal{I}}_{\mathrm{gs}}^{k+1} - \mathcal{I}_{\mathrm{gs}}^{k+1}} \operatorname{dist}^2\left(z^{\mathcal{I}_{\mathrm{gs}}^{k+1}}, \mathcal{Z}^*\right) = D_0^{2^{k+1}-1} \operatorname{dist}^2\left(z^{\widetilde{\mathcal{I}}_{\mathrm{lf}}^k}, \mathcal{Z}^*\right). \tag{83}$$

Then, by combining (82) and (83), we have

$$\operatorname{dist}^2\left(\widehat{z}^{k+1}, \mathcal{Z}^*\right) \leq D_0^{2^{k+1}-1} \cdot \frac{1}{\max\{D_0, 1\}^{2^{k+1}}} \left(1 - \frac{c_0\eta^2}{3}\right)^{4^k-1} \operatorname{dist}^2\left(\widehat{z}^k, \mathcal{Z}^*\right)$$

$$\leq \left(1 - \frac{c_0\eta^2}{3}\right)^{4^k-1} \operatorname{dist}^2\left(\widehat{z}^k, \mathcal{Z}^*\right). \tag{84}$$

Next, we give a rough bound of $\operatorname{dist}^2\left(z^t, \mathcal{Z}^*\right)$ for $t \in [\mathcal{I}_{\mathrm{lf}}^k : \widetilde{\mathcal{I}}_{\mathrm{gs}}^{k+1}]$.

For $t \in [\mathcal{I}_{\mathrm{lf}}^k : \widetilde{\mathcal{I}}_{\mathrm{lf}}^k]$, by (5),

$$\operatorname{dist}^2\left(z^t, \mathcal{Z}^*\right) \leq \Gamma_0 \cdot \left(1 - c_0\eta^2\right)^{t - \mathcal{I}_{\mathrm{lf}}^k} \operatorname{dist}^2\left(\widehat{z}^k, \mathcal{Z}^*\right) \leq \Gamma_0 \operatorname{dist}^2\left(\widehat{z}^k, \mathcal{Z}^*\right).$$

For $t \in [\mathcal{I}_{\mathrm{gs}}^{k+1} : \widetilde{\mathcal{I}}_{\mathrm{gs}}^{k+1}]$, since $z^{\widetilde{\mathcal{I}}_{\mathrm{lf}}^k}$ is the initial policy pair of the $(k+1)$-th call to `Global-Slow`, it follows by (3) that

$$\operatorname{dist}^2\left(z^t, \mathcal{Z}^*\right) \leq D_0^{t - \mathcal{I}_{\mathrm{gs}}^{k+1}} \operatorname{dist}^2\left(z^{\widetilde{\mathcal{I}}_{\mathrm{lf}}^k}, \mathcal{Z}^*\right) \leq \max\{D_0, 1\}^{2^{k+1}} \operatorname{dist}^2\left(z^{\widetilde{\mathcal{I}}_{\mathrm{lf}}^k}, \mathcal{Z}^*\right)$$

$$\leq \left(1 - \frac{c_0\eta^2}{3}\right)^{4^k-1} \operatorname{dist}^2\left(\widehat{z}^k, \mathcal{Z}^*\right) \leq \operatorname{dist}^2\left(\widehat{z}^k, \mathcal{Z}^*\right),$$

where the first inequality is from (3); the second inequality is from the fact that $\left|[\mathcal{I}_{\mathrm{gs}}^{k+1} : \widetilde{\mathcal{I}}_{\mathrm{gs}}^{k+1}]\right| = 2^{k+1}$; the third inequality is by (82).

Thus, for any $t \in [\mathcal{I}_{\mathrm{lf}}^k : \widetilde{\mathcal{I}}_{\mathrm{gs}}^{k+1}]$,

$$\operatorname{dist}^2\left(z^t, \mathcal{Z}^*\right) \leq \Gamma_0 \operatorname{dist}^2\left(\widehat{z}^k, \mathcal{Z}^*\right). \tag{85}$$

Now, we are ready to bound $\operatorname{dist}(z^t, \mathcal{Z}^*)$ for each $t \in [0 : T]$.

Firstly, we fix a $k' \geq k^* + 1$ and a $t' \in [\mathcal{I}_{\mathrm{lf}}^{k'} : \widetilde{\mathcal{I}}_{\mathrm{gs}}^{k'+1}]$. Then, the time interval $[0 : t']$ can be divided into:

$$[0 : t'] = [0 : \widetilde{\mathcal{I}}_{\mathrm{gs}}^{k^*}] \cup [\mathcal{I}_{\mathrm{lf}}^{k^*} : \widetilde{\mathcal{I}}_{\mathrm{gs}}^{k^*+1}] \cup \cdots \cup [\mathcal{I}_{\mathrm{lf}}^{k'-1} : \widetilde{\mathcal{I}}_{\mathrm{gs}}^{k'}] \cup [\mathcal{I}_{\mathrm{lf}}^{k'} : t'].$$

By (84), we have

$$\operatorname{dist}^2\left(\widehat{z}^{k'}, \mathcal{Z}^*\right) \leq \left(1 - \frac{c_0\eta^2}{3}\right)^{\sum_{k=k^*}^{k'-1}\left(4^k-1\right)} \operatorname{dist}^2\left(\widehat{z}^{k^*}, \mathcal{Z}^*\right) \leq 2S \left(1 - \frac{c_0\eta^2}{3}\right)^{\sum_{k=k^*}^{k'-1}\left(4^k-1\right)}$$

By combining with (85), we have

$$\operatorname{dist}^2\left(z^{t'}, \mathcal{Z}^*\right) \leq (2S\Gamma_0) \cdot \left(1 - \frac{c_0\eta^2}{3}\right)^{\sum_{k=k^*}^{k'-1}\left(4^k-1\right)}.$$

By (81),

$$\widetilde{\mathcal{I}}_{\mathrm{gs}}^{k^*} \le 2^{k^*} + \sum_{k=1}^{k^*-1} \left(2^k + 4^k\right) \le 2 \sum_{k=0}^{k^*} 4^k \le \frac{8}{3} \cdot 4^{k^*} \le \frac{32M^*}{3}. \tag{86}$$

Thus,

$$\sum_{k=k^*}^{k'-1} \left(4^k - 1\right) \ge \frac{1}{2} \sum_{k=k^*}^{k'-1} 4^k \ge \frac{1}{4} \sum_{k=k^*}^{k'-1} \left(4^k + 2^{k+1}\right) \ge \frac{1}{16} \sum_{k=k^*}^{k'} \left(4^k + 2^{k+1}\right)$$
$$= \frac{1}{16} \sum_{k=k^*}^{k'} \left(\widetilde{\mathcal{I}}_{\mathrm{gs}}^{k+1} - \widetilde{\mathcal{I}}_{\mathrm{gs}}^{k}\right) = \frac{\widetilde{\mathcal{I}}_{\mathrm{gs}}^{k'+1} - \widetilde{\mathcal{I}}_{\mathrm{gs}}^{k^*}}{16} \ge \frac{t' - \widetilde{\mathcal{I}}_{\mathrm{gs}}^{k^*}}{16} \ge \frac{t' - 32M^*/3}{16}.$$

Then, for the time $t'$ we have fixed,

$$\mathrm{dist}^2\left(\boldsymbol{z}^{t'}, \mathcal{Z}^*\right) \le (2S\Gamma_0) \cdot \left(1 - \frac{c_0\eta^2}{3}\right)^{\sum_{k=k^*}^{k'-1}\left(4^k - 1\right)} \le (2S\Gamma_0) \cdot \left(1 - \frac{c_0\eta^2}{3}\right)^{\frac{t' - 32M^*/3}{16}}. \tag{87}$$

Since the above arguments can be applied to any $k' \ge k^* + 1$ and $t \in [\mathcal{I}_{\mathrm{lf}}^{k'} : \widetilde{\mathcal{I}}_{\mathrm{gs}}^{k'+1}]$, we have that (87) holds for any $t \ge \mathcal{I}_{\mathrm{lf}}^{k^*+1}$.

By similar arguments to (86), we have $\widetilde{\mathcal{I}}_{\mathrm{gs}}^{k^*+1} \le 128M^*/3$. Then, for any $t \in [0 : \widetilde{\mathcal{I}}_{\mathrm{gs}}^{k^*+1}]$,

$$\mathrm{dist}(\boldsymbol{z}^t, \mathcal{Z}^*) \le 2S \le 2S \max\{\Gamma_0, 1\} \cdot \left(1 - \frac{c_0\eta^2}{3}\right)^{\frac{t - \widetilde{\mathcal{I}}_{\mathrm{gs}}^{k^*+1}}{16}}$$
$$\le 2S \max\{\Gamma_0, 1\} \cdot \left(1 - \frac{c_0\eta^2}{3}\right)^{\frac{t - 128M^*/3}{16}}.$$

Then, by combining with (87), for any $t \in [0 : T]$,

$$\mathrm{dist}^2\left(\boldsymbol{z}^t, \mathcal{Z}^*\right) \le 2S \max\{\Gamma_0, 1\} \cdot \left(1 - \frac{c_0\eta^2}{3}\right)^{\frac{t - 128M^*/3}{16}}$$
$$\le 2S \max\{\Gamma_0, 1\} \cdot \left(1 - \frac{c_0\eta^2}{48}\right)^{t - 128M^*/3}. \tag{88}$$

$\square$

**Proof of Theorem 2.** By Theorem 3 and Theorem 4, Averaging OGDA can serve as the base algorithm `Global-Slow` in the meta algorithm `Homotopy-PO`. By Theorem 5, OGDA can serve as the base algorithm `Local-Fast` in the meta algorithm `Homotopy-PO`.

Then, by Theorem 1, we have the global linear convergence of the instantiation of `Homotopy-PO` with OGDA and Averaging OGDA.

More specifically, by Theorem 1 and (57), the constant $c$ in (9) satisfies $c > 0$ and it is of order

$$c = \frac{c_0}{48} = O\left(\frac{(1-\gamma)^4 c_+^2}{S^3}\right).$$

By combining (80) with Theorem 3, Theorem 4, (57), (58), the constant $M$ in (9) is of order

$$M = O\left(\frac{S^{10}(A+B)^3 \log^2(SAB/(c_+(1-\gamma)))}{(1-\gamma)^{25} c_+^6}\right)$$

This completes the proof for global linear convergence of our instantiation for `Homotopy-PO`. $\square$

**Remark 2** *Theorem 2 requires* $\eta \leq O(\frac{(1-\gamma)^{\frac{5}{2}}}{\sqrt{S}(A+B)})$ *for OGDA and* $\eta' \leq O(\frac{1-\gamma}{A+B})$ *for Averaging OGDA. If we set* $\eta = O(\frac{(1-\gamma)^{\frac{5}{2}}}{\sqrt{S}(A+B)})$, *then the linear convergence rate is*

$$1 - c\eta^2 = 1 - O\left(\frac{(1-\gamma)^9 c_+^2}{S^4(A+B)^2}\right).$$

*If we set* $\eta = O(\frac{(1-\gamma)^{\frac{5}{2}}}{\sqrt{S}(A+B)})$ *for OGDA and* $\eta' = O(\frac{1-\gamma}{A+B})$ *for Averaging OGDA, then the length of Hidden Phase I is of order*

$$\frac{M\log^2(SAB/(c_+\eta\eta'))}{\eta^4\eta'^2} = O\left(\frac{S^{12}(A+B)^9\log^2(SAB/(c_+(1-\gamma)))}{(1-\gamma)^{37}c_+^6}\right).$$

# F   AUXILIARY LEMMAS

The following lemma gives a characterization of Nash equilibrium. Its proof can be found in Section 3.9 of Filar & Vrieze (2012).

**Lemma 19** *Consider Markov game* $\mathcal{G} = (\mathcal{S}, \mathcal{A}, \mathcal{B}, r, \mathbb{P}, \gamma)$. *Given the minimax game value* $v^*(s) = \min_{\boldsymbol{x}\in\mathcal{X}}\max_{\boldsymbol{y}\in\mathcal{Y}} V^{\boldsymbol{x},\boldsymbol{y}}(s)$. *A policy pair* $(\boldsymbol{x}^*, \boldsymbol{y}^*) \in \mathcal{X} \times \mathcal{Y}$ *is a Nash equilibrium if and only if it holds for any* $s \in \mathcal{S}$ *that* $(\boldsymbol{x}_s^*, \boldsymbol{y}_s^*)$ *is a Nash equilibrium of the matrix game*

$$\min_{\boldsymbol{x}_s\in\Delta_{\mathcal{A}}} \max_{\boldsymbol{y}_s\in\Delta_{\mathcal{B}}} \boldsymbol{x}_s^\top \boldsymbol{Q}_s^* \boldsymbol{y}_s, \tag{89}$$

*where* $\boldsymbol{Q}_s^*$ *is an A-by-B matrix with* $\boldsymbol{Q}_s^*(a, b) = \boldsymbol{R}_s(a, b) + \gamma \sum_{s'\in\mathcal{S}} v^*(s')\mathbb{P}(s'|s, a, b)$. *In addition, the minimax game value and the Nash equilibrium set of the matrix game* (89) *are* $v^*(s)$ *and* $\mathcal{Z}_s^* = \mathcal{X}_s^* \times \mathcal{Y}_s^*$, *respectively. Then, the Nash equilibrum set of Markov game* $\mathcal{G}$ *is* $\mathcal{Z}^* = \prod_{s\in\mathcal{S}} \mathcal{Z}_s^*$.

The following lemma is known as "performance difference lemma" (Kakade & Langford, 2002). It is used extensively throughout this paper.

**Lemma 20** *(Performance Difference Lemma) For any policies* $\boldsymbol{x}, \boldsymbol{x}' \in \mathcal{X}$, $\boldsymbol{y} \in \mathcal{Y}$ *and state* $s_0 \in \mathcal{S}$, *we have*

$$V^{\boldsymbol{x}',\boldsymbol{y}}(s_0) - V^{\boldsymbol{x},\boldsymbol{y}}(s_0) = \frac{1}{1-\gamma} \sum_{s\in\mathcal{S}} \boldsymbol{d}_{s_0}^{\boldsymbol{x}',\boldsymbol{y}}(s) \langle \boldsymbol{x}_s' - \boldsymbol{x}_s, Q_s^{\boldsymbol{x},\boldsymbol{y}}\boldsymbol{y}_s \rangle.$$

The following lemma is standard. We provide its proof for completeness.

**Lemma 21** *For any policies* $\boldsymbol{x}, \boldsymbol{x}' \in \mathcal{X}$, $\boldsymbol{y}, \boldsymbol{y}' \in \mathcal{Y}$ *and state* $s \in \mathcal{S}$, *state distribution* $\rho \in \Delta_{\mathcal{S}}$, *action pair* $(a, b) \in \mathcal{A} \times \mathcal{B}$. *Let* $\boldsymbol{z} = (\boldsymbol{x}, \boldsymbol{y})$ *and* $\boldsymbol{z}' = (\boldsymbol{x}', \boldsymbol{y}')$, *then*

$$\left|V^{\boldsymbol{x},\boldsymbol{y}}(s) - V^{\boldsymbol{x}',\boldsymbol{y}'}(s)\right| \leq \frac{\sqrt{A+B}\,\|\boldsymbol{z}-\boldsymbol{z}'\|}{(1-\gamma)^2}, \tag{90}$$

$$\left|\boldsymbol{Q}_s^{\boldsymbol{x},\boldsymbol{y}}(a, b) - \boldsymbol{Q}_s^{\boldsymbol{x}',\boldsymbol{y}'}(a, b)\right| \leq \frac{\gamma\sqrt{A+B}\,\|\boldsymbol{z}-\boldsymbol{z}'\|}{(1-\gamma)^2}, \tag{91}$$

$$\left|\boldsymbol{d}_{\boldsymbol{\rho}}^{\boldsymbol{x},\boldsymbol{y}}(s) - \boldsymbol{d}_{\boldsymbol{\rho}}^{\boldsymbol{x}',\boldsymbol{y}'}(s)\right| \leq \frac{\sqrt{A+B}\,\|\boldsymbol{z}-\boldsymbol{z}'\|}{1-\gamma}, \tag{92}$$

$$\left|V^{\boldsymbol{x},\dagger}(s) - V^{\boldsymbol{x}',\dagger}(s)\right| \leq \frac{\sqrt{A}\,\|\boldsymbol{x}-\boldsymbol{x}'\|}{(1-\gamma)^2}, \tag{93}$$

$$\left|V^{\dagger,\boldsymbol{y}}(s) - V^{\dagger,\boldsymbol{y}'}(s)\right| \leq \frac{\sqrt{B}\,\|\boldsymbol{y}-\boldsymbol{y}'\|}{(1-\gamma)^2}. \tag{94}$$

*Proof of Lemma 21.* By performance difference lemma (Lemma 20),

$$\left|V^{\boldsymbol{x},\boldsymbol{y}}(s) - V^{\boldsymbol{x}',\boldsymbol{y}}(s)\right| \leq \frac{1}{1-\gamma} \sum_{s'\in\mathcal{S}} \boldsymbol{d}_s^{\boldsymbol{x}',\boldsymbol{y}}(s') \,\|\boldsymbol{x}_{s'} - \boldsymbol{x}_{s'}'\|_1 \,\|\boldsymbol{Q}_{s'}^{\boldsymbol{x},\boldsymbol{y}}\boldsymbol{y}_s\|_\infty$$

$$\leq \frac{1}{(1-\gamma)^2} \sum_{s'\in\mathcal{S}} \boldsymbol{d}_s^{\boldsymbol{x}',\boldsymbol{y}}(s') \,\|\boldsymbol{x}_{s'} - \boldsymbol{x}_{s'}'\|_1$$

Similarly,

$$\left| V^{\boldsymbol{x}',\boldsymbol{y}}(s) - V^{\boldsymbol{x}',\boldsymbol{y}'}(s) \right| \le \frac{1}{1-\gamma} \sum_{s' \in \mathcal{S}} \boldsymbol{d}_s^{\boldsymbol{x}',\boldsymbol{y}}(s') \left\| \boldsymbol{y}_{s'} - \boldsymbol{y}'_{s'} \right\|_1 \left\| {\boldsymbol{Q}_{s'}^{\boldsymbol{x}',\boldsymbol{y}'}}^\top \boldsymbol{x}'_s \right\|_\infty$$

$$\le \frac{1}{(1-\gamma)^2} \sum_{s' \in \mathcal{S}} \boldsymbol{d}_s^{\boldsymbol{x}',\boldsymbol{y}}(s') \left\| \boldsymbol{y}_{s'} - \boldsymbol{y}'_{s'} \right\|_1 .$$

Then, by triangle inequality and the fact that $\sum_{s' \in \mathcal{S}} \boldsymbol{d}_s^{\boldsymbol{x}',\boldsymbol{y}}(s') = 1$, we have

$$\left| V^{\boldsymbol{x},\boldsymbol{y}}(s) - V^{\boldsymbol{x}',\boldsymbol{y}'}(s) \right| \le \frac{1}{(1-\gamma)^2} \sum_{s' \in \mathcal{S}} \boldsymbol{d}_s^{\boldsymbol{x}',\boldsymbol{y}}(s') \left\| \boldsymbol{z}_{s'} - \boldsymbol{z}'_{s'} \right\|_1$$

$$\le \frac{\sqrt{A+B} \max_{s' \in \mathcal{S}} \left\| \boldsymbol{z}'_s - \boldsymbol{z}'_{s'} \right\|}{(1-\gamma)^2} \le \frac{\sqrt{A+B} \left\| \boldsymbol{z} - \boldsymbol{z}' \right\|}{(1-\gamma)^2} .$$

Then, (91) follows by combining (90) with the definition $\boldsymbol{Q}_s^{\boldsymbol{x},\boldsymbol{y}} = \boldsymbol{Q}_s[V^{\boldsymbol{x},\boldsymbol{y}}]$.

To bound the difference of state visitation distribution, we fix $s, s' \in \mathcal{S}$. Let $\boldsymbol{P} \in \mathbb{R}^{S \times S}$ be the transition matrix of policy pair $(\boldsymbol{x}, \boldsymbol{y})$, i.e.,

$$\boldsymbol{P}(s, s_1) = \sum_{a \in \mathcal{A}} \sum_{b \in \mathcal{B}} \boldsymbol{x}_s(a) \boldsymbol{y}_s(b) \mathbb{P}(s_1 | s, a, b) .$$

Similarly, define $\boldsymbol{P}'$ as the transition matrix of $(\boldsymbol{x}', \boldsymbol{y}')$. Then, $\boldsymbol{d}_s^{\boldsymbol{x},\boldsymbol{y}}(s_1)$ is the $(s, s_1)$-th entry of $(1-\gamma)(\boldsymbol{I} - \boldsymbol{P})^{-1}$; $\boldsymbol{d}_s^{\boldsymbol{x}',\boldsymbol{y}'}(s_1)$ is the $(s, s_1)$-th entry of $(\boldsymbol{I} - \boldsymbol{P}')^{-1}$. By definition, for any $s, s_1 \in \mathcal{S}$,

$$\sum_{s_1 \in \mathcal{S}} \left| \boldsymbol{P}(s, s_1) - \boldsymbol{P}'(s, s_1) \right|$$

$$\le \sum_{s_1 \in \mathcal{S}} \sum_{a \in \mathcal{A}} \sum_{b \in \mathcal{B}} \left| \boldsymbol{x}_s(a) - \boldsymbol{x}'_s(a) \right| \boldsymbol{y}_s(b) \mathbb{P}(s_1 | s, a, b)$$

$$+ \sum_{s_1 \in \mathcal{S}} \sum_{a \in \mathcal{A}} \sum_{b \in \mathcal{B}} \boldsymbol{x}'_s(a) \left| \boldsymbol{y}_s(b) - \boldsymbol{y}'_s(b) \right| \mathbb{P}(s_1 | s, a, b)$$

$$\le \left\| \boldsymbol{z}_s - \boldsymbol{z}'_s \right\|_1 .$$

Thus, we have $\left\| \boldsymbol{P} - \boldsymbol{P}' \right\|_\infty \le \max_{\tilde{s} \in \mathcal{S}} \left\| \boldsymbol{z}_{\tilde{s}} - \boldsymbol{z}'_{\tilde{s}} \right\|_1$.

By combining with the fact that $\left\| (\boldsymbol{I} - \boldsymbol{P})^{-1} \right\|_\infty \le \sum_{i=0}^\infty \gamma^i \left\| \boldsymbol{P}^i \right\|_\infty \le \frac{1}{1-\gamma}$, we have

$$\left| \boldsymbol{d}_s^{\boldsymbol{x},\boldsymbol{y}}(s_1) - \boldsymbol{d}_s^{\boldsymbol{x}',\boldsymbol{y}'}(s_1) \right| = (1-\gamma) \left| \left\langle \mathbf{1}_s, (\boldsymbol{I} - \boldsymbol{P})^{-1} (\boldsymbol{P} - \boldsymbol{P}') (\boldsymbol{I} - \boldsymbol{P}')^{-1} \mathbf{1}_{s_1} \right\rangle \right|$$

$$\le (1-\gamma) \left\| (\boldsymbol{I} - \boldsymbol{P})^{-1} \right\|_\infty \left\| \boldsymbol{P} - \boldsymbol{P}' \right\|_\infty \left\| (\boldsymbol{I} - \boldsymbol{P}')^{-1} \right\|_\infty$$

$$\le \frac{\sqrt{A+B} \max_{s' \in \mathcal{S}} \left\| \boldsymbol{z}'_s - \boldsymbol{z}'_{s'} \right\|}{1-\gamma} \le \frac{\sqrt{A+B} \left\| \boldsymbol{z} - \boldsymbol{z}' \right\|}{1-\gamma} .$$

Then,

$$\left| \boldsymbol{d}_{\boldsymbol{\rho}}^{\boldsymbol{x},\boldsymbol{y}}(s) - \boldsymbol{d}_{\boldsymbol{\rho}}^{\boldsymbol{x}',\boldsymbol{y}'}(s) \right| \le \sum_{s_0 \in \mathcal{S}} \rho(s_0) \left| \boldsymbol{d}_{s_0}^{\boldsymbol{x},\boldsymbol{y}}(s) - \boldsymbol{d}_{s_0}^{\boldsymbol{x}',\boldsymbol{y}'}(s) \right| \le \frac{\sqrt{A+B} \left\| \boldsymbol{z} - \boldsymbol{z}' \right\|}{1-\gamma} .$$

To show (93), we choose $\widehat{\boldsymbol{y}} \in \arg\max_{\boldsymbol{y}} V^{\boldsymbol{x},\boldsymbol{y}}(s)$, then, by performance difference lemma (Lemma 20),

$$V^{\boldsymbol{x},\widehat{\boldsymbol{y}}}(s) - V^{\boldsymbol{x}',\widehat{\boldsymbol{y}}}(s) \le \frac{1}{1-\gamma} \sum_{s' \in \mathcal{S}} \boldsymbol{d}_s^{\boldsymbol{x},\widehat{\boldsymbol{y}}}(s') \left\| \boldsymbol{x}_{s'} - \boldsymbol{x}'_{s'} \right\|_1 \left\| \boldsymbol{Q}_{s'}^{\boldsymbol{x}',\widehat{\boldsymbol{y}}} \widehat{\boldsymbol{y}}_s \right\|_\infty$$

$$\le \frac{\max_{\tilde{s} \in \mathcal{S}} \left\| \boldsymbol{x}_{\tilde{s}} - \boldsymbol{x}'_{\tilde{s}} \right\|_1}{(1-\gamma)^2} \le \frac{\sqrt{A} \left\| \boldsymbol{x} - \boldsymbol{x}' \right\|}{(1-\gamma)^2} .$$

Analogously, $V^{\boldsymbol{x}',\dagger}(s) - V^{\boldsymbol{x},\dagger}(s) \leq \frac{\sqrt{A}\|\boldsymbol{x}-\boldsymbol{x}'\|}{(1-\gamma)^2}$. Thus, $\left|V^{\boldsymbol{x},\dagger}(s) - V^{\boldsymbol{x}',\dagger}(s)\right| \leq \frac{\sqrt{A}\|\boldsymbol{x}-\boldsymbol{x}'\|}{(1-\gamma)^2}$. The inequality (94) follows similarly. $\qquad\square$

As a direct corollary of (93), (94), we can bound the Nash gap $\max_{s\in\mathcal{S}} V^{\boldsymbol{x},\dagger}(s) - V^{\dagger,\boldsymbol{y}}(s)$ by $\mathrm{dist}(\boldsymbol{z},\mathcal{Z}^*)$.

**Corollary 1** *For any $\boldsymbol{z} = (\boldsymbol{x},\boldsymbol{y}) \in \mathcal{Z}$,*

$$\max_{s\in\mathcal{S}} V^{\boldsymbol{x},\dagger}(s) - V^{\dagger,\boldsymbol{y}}(s) \leq \frac{\max\{\sqrt{2A},\sqrt{2B}\}}{(1-\gamma)^2} \cdot \mathrm{dist}(\boldsymbol{z},\mathcal{Z}^*).$$

*Proof of Corollary 1.* Denote $\mathcal{P}_{\mathcal{X}^*}(\boldsymbol{x}) = \boldsymbol{x}^*$, $\mathcal{P}_{\mathcal{Y}^*}(\boldsymbol{y}) = \boldsymbol{y}^*$, then $\boldsymbol{z}^* = (\boldsymbol{x}^*,\boldsymbol{y}^*) = \mathcal{P}_{\mathcal{Z}^*}(\boldsymbol{z})$. By the definition of Nash equilibria,

$$V^{\boldsymbol{x}^*,\boldsymbol{y}^*}(s) = V^{\boldsymbol{x}^*,\dagger}(s) = V^{\dagger,\boldsymbol{y}^*}(s).$$

Then, by combining with (93), (94), for any $s \in \mathcal{S}$,

$$\max_{s\in\mathcal{S}} V^{\boldsymbol{x},\dagger}(s) - V^{\dagger,\boldsymbol{y}}(s) = \max_{s\in\mathcal{S}} V^{\boldsymbol{x},\dagger}(s) - V^{\boldsymbol{x}^*,\dagger}(s) + V^{\dagger,\boldsymbol{y}^*}(s) - V^{\dagger,\boldsymbol{y}}(s)$$

$$\leq \max_{s\in\mathcal{S}} \frac{\sqrt{A}\|\boldsymbol{x}-\boldsymbol{x}^*\|}{(1-\gamma)^2} + \frac{\sqrt{B}\|\boldsymbol{y}-\boldsymbol{y}^*\|}{(1-\gamma)^2} \leq \frac{\max\{\sqrt{2A},\sqrt{2B}\}\mathrm{dist}(\boldsymbol{z},\mathcal{Z}^*)}{(1-\gamma)^2}.$$

This completes the proof. $\qquad\square$

The following lemma is paraphrased from Lemma 4 of Gilpin et al. (2012) and is also known as saddle-point metric subregularity of matrix games as in Theorem 5 of Wei et al. (2020). It essentially shows that in matrix game $\min_{\boldsymbol{x}} \max_{\boldsymbol{y}} \boldsymbol{x}^\top \boldsymbol{G}\boldsymbol{y}$, the suboptimality of any policy pair can be lower bounded by a certain condition measure $\varphi(\boldsymbol{G})$ of the matrix $\boldsymbol{G}$ multiplied by the policy pair's distance to the Nash equilibrium set of the matrix game.

**Lemma 22** *(Lemma 4 of Gilpin et al. (2012), Theorem 5 of Wei et al. (2020)) For any matrix $\boldsymbol{G} \in \mathbb{R}^{A\times B}$, let $\mathcal{X}^*(\boldsymbol{G}) = \arg\min_{\boldsymbol{x}'\in\Delta_A}(\max_{\boldsymbol{y}'\in\Delta_B} \boldsymbol{x}'^\top \boldsymbol{G}\boldsymbol{y}')$ and $\mathcal{Y}^*(\boldsymbol{G}) = \arg\max_{\boldsymbol{y}'\in\Delta_B}(\min_{\boldsymbol{x}'\in\Delta_A} \boldsymbol{x}'^\top \boldsymbol{G}\boldsymbol{y}')$. Then, it holds that for any $\boldsymbol{x}\in\Delta_A$ and $\boldsymbol{y}\in\Delta_B$,*

$$\max_{\boldsymbol{y}'\in\Delta_B} \boldsymbol{x}^\top \boldsymbol{Q}\boldsymbol{y}' - \min_{\boldsymbol{x}'\in\Delta_A} \boldsymbol{x}'^\top \boldsymbol{Q}\boldsymbol{y} \geq \varphi(\boldsymbol{Q}) \cdot \sqrt{\mathrm{dist}^2(\boldsymbol{x},\mathcal{X}^*(\boldsymbol{G})) + \mathrm{dist}^2(\boldsymbol{y},\mathcal{Y}^*(\boldsymbol{G}))},$$

*where $\varphi(\boldsymbol{Q}) > 0$ is a certain condition measure of the matrix $\boldsymbol{Q}$.*

As a direct corollary of Lemma 22, we can instantiate the value of $c_+$ in (1).

**Corollary 2** *Let $c_+ = \min_{s\in\mathcal{S}} \varphi(\boldsymbol{Q}_s^*)$, then, for any policy pair $\boldsymbol{z} = (\boldsymbol{x},\boldsymbol{y}) \in \mathcal{Z}$ and $s\in\mathcal{S}$,*

$$\max_{\boldsymbol{y}_s'\in\Delta_{\mathcal{B}}} \boldsymbol{x}_s^\top \boldsymbol{Q}_s^*\boldsymbol{y}_s' - \min_{\boldsymbol{x}_s'\in\Delta_{\mathcal{A}}} \boldsymbol{x}_s'^\top \boldsymbol{Q}_s^*\boldsymbol{y}_s \geq c_+ \cdot \mathrm{dist}(\boldsymbol{z}_s,\mathcal{Z}_s^*).$$

## G DECENTRALIZED IMPLEMENTATION OF THE ALGORITHMS

Recall that in our interaction protocol, the min-player only has access to its marginal reward function $\boldsymbol{r}_x^t$ and marginal transition kernel $\mathbb{P}_x^t$, while the max-player only has access to its marginal reward function $\boldsymbol{r}_y^t$ and marginal transition kernel $\mathbb{P}_y^t$.

Equivalently, in each iteration, the min-player receives full information of the Markov Decision Process (MDP) $\mathcal{M}_x^t = (\mathcal{S},\mathcal{A},\mathbb{P}_x^t,\boldsymbol{r}_x^t,\gamma)$, the max-player receives $\mathcal{M}_y^t = (\mathcal{S},\mathcal{B},\mathbb{P}_y^t,\boldsymbol{r}_y^t,\gamma)$.

The marginal rewards and transition kernels are defined as

$$\boldsymbol{r}_x^t(s,a) = \sum_{b\in\mathcal{B}} \boldsymbol{y}_s^t(b)\boldsymbol{R}_s(a,b), \quad \mathbb{P}_x^t(s'|s,a) = \sum_{b\in\mathcal{B}} \boldsymbol{y}_s^t(b)\mathbb{P}(s'|s,a,b),$$

$$\boldsymbol{r}_y^t(s,b) = \sum_{a\in\mathcal{A}} \boldsymbol{x}_s^t(a)\boldsymbol{R}_s(a,b), \quad \mathbb{P}_y^t(s'|s,a) = \sum_{a\in\mathcal{A}} \boldsymbol{x}_s^t(a)\mathbb{P}(s'|s,a,b).$$

(95)

The value function of the policy $\boldsymbol{x}$ in the MDP $\mathcal{M}_x^t$ is defined as an $S$-dimensional vector containing the expected cumulative rewards of each state, i.e.,

$$V^{\boldsymbol{x},\mathcal{M}_x^t}(s) = \mathbb{E}_{\boldsymbol{x},\boldsymbol{y}^t} \left[ \sum_{j=0}^{+\infty} \gamma^j \boldsymbol{r}_x^t \left( s^j, a^j \right) | s^0 = s \right].$$

The q-function $\boldsymbol{q}^{\boldsymbol{x},\mathcal{M}_x^t} = \{ \boldsymbol{q}_s^{\boldsymbol{x},\mathcal{M}_x^t} \}_{s \in \mathcal{S}}$ is defined as a collection of $A$-dimensional vector with

$$\boldsymbol{q}_s^{\boldsymbol{x},\mathcal{M}_x^t}(a) = \boldsymbol{r}_x^t(s,a) + \gamma \sum_{s' \in \mathcal{S}} \mathbb{P}_x^t(s'|s,a) V^{\boldsymbol{x},\mathcal{M}_x^t}(s').$$

The counterparts $V^{\mathcal{M}_y^t,\boldsymbol{y}}(s)$, $\boldsymbol{q}_s^{\mathcal{M}_y^t,\boldsymbol{y}}$ for the max-player are defined similarly.

In the pseudocodes below, for any set $\mathcal{C}$, $\mathbb{I}_{\mathcal{C}}$ denotes its indicator.

The decentralized implementation of OGDA (8) is in Algorithm 2 (min-player's perspective) and Algorithm 3 (max-player's perspective).

The decentralized implementation of Averaging OGDA (6) is in Algorithm 4 (min-player's perspective) and Algorithm 5 (max-player's perspective).

Then, our instantiation of the meta algorithm `Homotopy-PO` which uses Averaging OGDA as `Global-Slow` and OGDA as `Local-Fast` is naturally a decentralized algorithm. The pseudocodes are presented in Algorithm 6 (min-player's perspective) and Algorithm 7 (max-player's perspective).

● **Equivalence between OGDA** (8) **and Algorithm 2, 3**

To prove the equivalence between OGDA (8) and Algorithm 2, 3, it suffices to show that $\boldsymbol{q}_s^{\boldsymbol{x}^t,\mathcal{M}_x^t} = \boldsymbol{Q}_s^t \boldsymbol{y}_s^t$. Actually, both $\boldsymbol{q}_s^{\boldsymbol{x}^t,\mathcal{M}_x^t}$ and $\boldsymbol{Q}_s^t \boldsymbol{y}_s^t$ equals the marginal q-function of the local MDP $\mathcal{M}_x^t = \{\mathcal{S}, \mathcal{A}, \mathbb{P}_x^t, \boldsymbol{r}_x^t, \gamma\}$ observed by the min-player at iteration $t$.

By definition, we have for any $s \in \mathcal{S}$, $V^{\boldsymbol{x}^t,\boldsymbol{y}^t}(s) = V^{\boldsymbol{x}^t,\mathcal{M}_x^t}(s) = V^{\mathcal{M}_y^t,\boldsymbol{y}^t}(s)$. Then, we have

$$\begin{aligned}
\boldsymbol{q}_s^{\boldsymbol{x}^t,\mathcal{M}_x^T}(a) &= \sum_{b \in \mathcal{B}} \boldsymbol{R}_s(a,b) \boldsymbol{y}_s^t(b) + \sum_{b \in \mathcal{B}} \sum_{s' \in \mathcal{S}} \mathbb{P}_x^t(s'|s,a,b) V^{\boldsymbol{x}^t,\mathcal{M}_x^t}(s') \boldsymbol{y}_s^t(b) \\
&= \sum_{b \in \mathcal{B}} \boldsymbol{R}_s(a,b) \boldsymbol{y}_s^t(b) + \sum_{b \in \mathcal{B}} \sum_{s' \in \mathcal{S}} \mathbb{P}_x^t(s'|s,a,b) V^{\boldsymbol{x}^t,\boldsymbol{y}^t}(s') \boldsymbol{y}_s^t(b) \\
&= \left\langle \mathbf{1}_a, \boldsymbol{Q}_s^{\boldsymbol{x}^t,\boldsymbol{y}^t} \boldsymbol{y}_s^t \right\rangle = \left\langle \mathbf{1}_a, \boldsymbol{Q}_s^t \boldsymbol{y}_s^t \right\rangle.
\end{aligned}$$

Thus, $\boldsymbol{q}_s^{\boldsymbol{x}^t,\mathcal{M}_x^T} = \boldsymbol{Q}_s^t \boldsymbol{y}_s^t$. Analogously, $\boldsymbol{q}_s^{\boldsymbol{y}^t,\mathcal{M}_y^T} = \left( \boldsymbol{Q}_s^t \right)^\top \boldsymbol{x}_s^t$. This gives the equivalence between OGDA (8) and Algorithm 2, 3.

---

**Algorithm 2:** `x-OGDA`

---

**Input:** time interval: $[T_1 : T_2]$, initial policy: $\widehat{\boldsymbol{x}} \in \mathcal{X}$, stepsize: $\eta$
Initialize $\boldsymbol{x}^{T_1} = \widehat{\boldsymbol{x}}$
**for** $t = T_1, \cdots, T_2$ **do**
    play policy $\boldsymbol{x}^t$
    receive $\boldsymbol{r}_x^t$ and $\mathbb{P}_x^t$
    compute the q-function $\left\{ \boldsymbol{q}_s^{\boldsymbol{x}^t,\mathcal{M}_x^t} \right\}_{s \in \mathcal{S}}$ in the MDP $\mathcal{M}_x^t = (\mathcal{S}, \mathcal{A}, \mathbb{P}_x^t, \boldsymbol{r}_x^t, \gamma)$
    optimistic gradient descent

$$\widetilde{\boldsymbol{x}}_s^t = \mathbb{I}_{\{t=T_1\}} \cdot \boldsymbol{x}_s^{T_1} + \mathbb{I}_{\{t>T_1\}} \cdot \mathcal{P}_{\Delta_\mathcal{A}} \left( \widetilde{\boldsymbol{x}}_s^{t-1} - \eta \boldsymbol{q}_s^{\boldsymbol{x}^t,\mathcal{M}_x^t} \right)$$

$$\boldsymbol{x}_s^{t+1} = \mathcal{P}_{\Delta_\mathcal{A}} \left( \widetilde{\boldsymbol{x}}_s^t - \eta \boldsymbol{q}_s^{\boldsymbol{x}^t,\mathcal{M}_x^t} \right)$$

**end**

---

---

**Algorithm 3:** `y-OGDA`

---

**Input:** time interval: $[T_1 : T_2]$, initial policy: $\widehat{\boldsymbol{y}} \in \mathcal{Y}$, stepsize: $\eta$,

Initialize $\boldsymbol{y}^{T_1} = \widehat{\boldsymbol{y}}$

**for** $t = T_1, \cdots, T_2$ **do**

> play policy $\boldsymbol{y}^t$
>
> receive $\boldsymbol{r}_y^t$ and $\mathbb{P}_y^t$
>
> compute the q-function $\left\{ \boldsymbol{q}_s^{\mathcal{M}_y^t, \boldsymbol{y}^t} \right\}_{s \in \mathcal{S}}$ in the MDP $\mathcal{M}_y^t = \left( \mathcal{S}, \mathcal{B}, \mathbb{P}_y^t, \boldsymbol{r}_y^t, \gamma \right)$
>
> optimistic gradient ascent
>
> $$\widetilde{\boldsymbol{y}}_s^t = \mathbb{I}_{\{t=T_1\}} \cdot \boldsymbol{y}_s^{T_1} + \mathbb{I}_{\{t>T_1\}} \cdot \mathcal{P}_{\Delta_{\mathcal{B}}} \left( \widetilde{\boldsymbol{y}}_s^{t-1} + \eta \boldsymbol{q}_s^{\mathcal{M}_y^t, \boldsymbol{y}^t} \right)$$
>
> $$\boldsymbol{y}_s^{t+1} = \mathcal{P}_{\Delta_{\mathcal{B}}} \left( \widetilde{\boldsymbol{y}}_s^t + \eta \boldsymbol{q}_s^{\mathcal{M}_y^t, \boldsymbol{y}^t} \right)$$

**end**

---

**Algorithm 4:** `x-Averaging-OGDA`

---

**Input:** time interval: $[T_1 : T_2]$, initial policy $\tilde{\boldsymbol{x}} \in \mathcal{X}$, stepsize: $\eta$

Initialize $\boldsymbol{x}^{T_1} = \tilde{\boldsymbol{x}}$

**for** $t = T_1, \cdots, T_2$ **do**

> play policy $\boldsymbol{x}^t$
>
> receive $\boldsymbol{r}_x^t$ and $\mathbb{P}_x^t$
>
> **if** $t == T_1$ **then**
>
> > solve the MDP $\mathcal{M}_x^{T_1} = \left( \mathcal{S}, \mathcal{A}, \mathbb{P}_x^{T_1}, \boldsymbol{r}_x^{T_1}, \gamma \right)$ to compute
> >
> > $\underline{V}^{T_1}(s) = \min_{\boldsymbol{x}' \in \mathcal{X}} V^{\boldsymbol{x}', \mathcal{M}_x^{T_1}}(s)$ for any $s \in \mathcal{S}$
>
> **end**
>
> compute for $(s, a) \in \mathcal{S} \times \mathcal{A}$, $\underline{\boldsymbol{q}}_s^t(a) = \boldsymbol{r}_x^t(s, a) + \gamma \sum_{s' \in \mathcal{S}} \mathbb{P}_x^t(s'|s, a) \underline{V}^t(s')$
>
> optimistic gradient descent
>
> $$\widetilde{\boldsymbol{x}}_s^t = \mathbb{I}_{\{t=T_1\}} \cdot \boldsymbol{x}_s^{T_1} + \mathbb{I}_{\{t>T_1\}} \cdot \mathcal{P}_{\Delta_{\mathcal{A}}} \left( \widetilde{\boldsymbol{x}}_s^{t-1} - \eta \underline{\boldsymbol{q}}_s^t \right)$$
>
> $$\boldsymbol{x}_s^{t+1} = \mathcal{P}_{\Delta_{\mathcal{A}}} \left( \widetilde{\boldsymbol{x}}_s^t - \eta \underline{\boldsymbol{q}}_s^t \right)$$
>
> update value function $\underline{V}^{t+1}(s) = \min_{a \in \mathcal{A}} \sum_{j=T_1}^t \alpha_{t-T_1+1}^{j-T_1+1} \underline{\boldsymbol{q}}_s^j(a)$

**end**

Compute the average policy $\widehat{\boldsymbol{x}}^{[T_1:T_2]} = \sum_{t=T_1}^{T_2} \alpha_{T_2-T_1+1}^{t-T_1+1} \boldsymbol{x}^t$

---

---

**Algorithm 5:** `y-Averaging-OGDA`

---

**Input:** time interval: $[T_1 : T_2]$, initial policy $\tilde{\boldsymbol{y}} \in \mathcal{Y}$, stepsize: $\eta$

Initialize $\boldsymbol{y}^{T_1} = \tilde{\boldsymbol{y}}$

**for** $t = T_1, \cdots, T_2$ **do**

    play policy $\boldsymbol{y}^t$

    receive $\boldsymbol{r}_y^t$ and $\mathbb{P}_y^t$

    **if** $t == T_1$ **then**

        solve the MDP $\mathcal{M}_y^{T_1} = \left( \mathcal{S}, \mathcal{B}, \mathbb{P}_y^{T_1}, \boldsymbol{r}_y^{T_1}, \gamma \right)$ to compute

        $\overline{V}^{T_1}(s) = \max_{\boldsymbol{y}' \in \mathcal{Y}} V^{\mathcal{M}_y^{T_1}, \boldsymbol{y}'}(s)$ for any $s \in \mathcal{S}$

    **end**

    compute for $(s,b) \in \mathcal{S} \times \mathcal{B}$, $\overline{\boldsymbol{q}}_s^t(b) = \boldsymbol{r}_y^t(s,b) + \gamma \sum_{s' \in \mathcal{S}} \mathbb{P}_y^t(s'|s,b) \overline{V}^t(s')$

    optimistic gradient ascent

$$\widetilde{\boldsymbol{y}}_s^t = \mathbb{I}_{\{t = T_1\}} \cdot \boldsymbol{y}_s^{T_1} + \mathbb{I}_{\{t > T_1\}} \cdot \mathcal{P}_{\Delta_{\mathcal{B}}} \left( \widetilde{\boldsymbol{y}}_s^{t-1} + \eta \overline{\boldsymbol{q}}_s^t \right)$$

$$\boldsymbol{y}_s^{t+1} = \mathcal{P}_{\Delta_{\mathcal{B}}} \left( \widetilde{\boldsymbol{y}}_s^t + \eta \overline{\boldsymbol{q}}_s^t \right)$$

    update value function $\overline{V}^{t+1}(s) = \max_{b \in \mathcal{B}} \sum_{j=T_1}^t \alpha_{t-T_1+1}^{j-T_1+1} \overline{\boldsymbol{q}}_s^j(b)$

**end**

Compute the average policy $\widehat{\boldsymbol{y}}^{[T_1:T_2]} = \sum_{t=T_1}^{T_2} \alpha_{T_2-T_1+1}^{t-T_1+1} \boldsymbol{y}^t$

---

● **Equivalence between Averaging OGDA (6) and Algorithm 4, 5**

Firstly, it follows by definition that

$$V^{\dagger, \boldsymbol{y}^t}(s) = \min_{\boldsymbol{x}'' \in \mathcal{X}} V^{\boldsymbol{x}'', \mathcal{M}_x^t}(s), \ V^{\boldsymbol{x}^t, \dagger}(s) = \max_{\boldsymbol{y}'' \in \mathcal{Y}} V^{\mathcal{M}_y^t, \boldsymbol{y}''}(s). \tag{96}$$

Thus, the initiation steps in Averaging OGDA (6) and Algorithm 4, 5 are equivalent. Thus, $\underline{V}^{T_1}$ in Averaging OGDA (6) equals that in Algorithm 4.

Consider the variable $\underline{\boldsymbol{q}}_s^t(a)$ defined in Algorithm 4,

$$\underline{\boldsymbol{q}}_s^t(a) = \boldsymbol{r}_x^t(s,a) + \gamma \sum_{s' \in \mathcal{S}} \mathbb{P}_x^t(s'|s,a) \underline{V}^t(s'). \tag{97}$$

By substituting (95) into (97) and combining the definition of the Bellman target operator in the introduction, we have

$$\underline{\boldsymbol{q}}_s^t(a) = \sum_{b \in \mathcal{B}} \boldsymbol{R}_s(a,b) \boldsymbol{y}_s^t(b) + \sum_{b \in \mathcal{B}} \underline{V}^t(s') \mathbb{P}(s'|s,a,b) \boldsymbol{y}_s^t(b) = \left\langle \mathbf{1}_a, \boldsymbol{Q}_s[\underline{V}^t] \boldsymbol{y}_s^t \right\rangle, \tag{98}$$

The RHS of (98) is exactly our definition for $\underline{\boldsymbol{q}}_s^t$ in Averaging OGDA (6) in Section 3.2. Analogously, the definition for $\overline{\boldsymbol{q}}_s^t$ equals in (6) and Algorithm 5.

Then, by induction, $\left\{ \underline{\boldsymbol{q}}_s^t, \overline{\boldsymbol{q}}_s^t, \underline{V}^t(s), \overline{V}^t(s) \right\}_{t \in [T_1:T_2], s \in \mathcal{S}}$ has the same value in Averaging OGDA (6) and Algorithm 4, 5. This gives the equivalence of Averaging OGDA (6) and Algorithm 4, 5.

●**Decentralized implementation of Homotopy-PO**

Recall that we have shown in Section 4 and Section 5 that Averaging OGDA (6) and OGDA (8) can serve as the base algorithm `Global-Slow` and `Local-Fast` in the meta algorithm `Homotopy-PO`, respectively. Thus, we can interpolate Averaging OGDA as `Global-Slow` and OGDA as `Local-Fast` in the meta algorithm `Homotopy-PO` to obtain a globally linearly convergence algorithm.

We have shown that Averaging OGDA can be implemented in a decentralized manner (Algorithm 4, 5), and OGDA is also a decentralized algorithm (Algorithm 2, 3). In addition, the inputs of Algorithm 2, 3 and Algorithm 4, 5 only need local information (the min-player only needs $\boldsymbol{x}^{T_1}$, the

max-player only needs $\boldsymbol{y}^{T_1}$) with no requirement for knowledge of its opponent's policies. Thus, the algorithm constructed by interpolating Averaging OGDA and OGDA into `Homotopy-PO` is naturally a decentralized algorithm. The pseudocodes in the min-player's and the max-player's perspectives are illustrated in Algorithm 6 and Algorithm 7, respectively.

We make final remarks that our instantiation for `Homotopy-PO` is symmetric and rational. Since the min-player and the max-player use equal stepsize $\eta$ for OGDA and equal stepsize $\eta'$ for Averaging OGDA, the players have symmetric roles in our algorithms.

Rationality means one player can converge to the best response set when its opponent chooses a stationary policy. This property is naturally possessed by decentralized and symmetric algorithms. Similar arguments for rationality can also be found in some existing decentralized algorithms, see for instance Sayin et al. (2021); Wei et al. (2021). We attach the proof for rationality here for completeness. In addition, since our instantiation of `Homotopy-PO` has linear convergence, it is not only rational but also able to guarantee the linear convergence to the best response set.

**Theorem 6** *(Rationality) If the max-player chooses a stationary policy $\widehat{\boldsymbol{y}} = \{\widehat{\boldsymbol{y}}_s\}_{s\in\mathcal{S}} \in \mathcal{Y}$ and the min-player runs the instantiation of `Homotopy-PO` (Algorithm 6), then $\boldsymbol{x}^t$ will converge to the best response set $\{\boldsymbol{x} \in \mathcal{X} : V^{\boldsymbol{x},\widehat{\boldsymbol{y}}}(s) = V^{\dagger,\widehat{\boldsymbol{y}}}(s), \forall s \in \mathcal{S}\}$ at a linear rate. Analogously, if the min-player chooses a stationary policy $\widehat{\boldsymbol{x}} = \{\widehat{\boldsymbol{x}}_s\}_{s\in\mathcal{S}} \in \mathcal{X}$ and the max-player runs the instantiation of `Homotopy-PO` (Algorithm 7), then $\boldsymbol{y}^t$ will converge to the best response set $\{\boldsymbol{y} \in \mathcal{Y} : V^{\widehat{\boldsymbol{x}},\boldsymbol{y}}(s) = V^{\widehat{\boldsymbol{x}},\dagger}(s), \forall s \in \mathcal{S}\}$ at a linear rate.*

*Proof of Theorem 6.* Since the min-player and the max-player are symmetric, without loss of generality, we let the max-player chooses a stationary policy $\widehat{\boldsymbol{y}} = \{\widehat{\boldsymbol{y}}_s\}_{s\in\mathcal{S}} \in \mathcal{Y}$.

Then, we define a new Markov game $\mathcal{MG}' = (\mathcal{S}, \mathcal{A}, \widehat{\mathcal{B}}, \widehat{\mathbb{P}}, \widehat{\boldsymbol{R}}, \gamma)$, where $\mathcal{S}, \mathcal{A}, \gamma$ have the same meaning as in the original Markov game. Now, the action set of the max-player only has one action $\widehat{\mathcal{B}} = \{1\}$. $\widehat{\mathbb{P}}(s'|s, a, 1) = \sum_{b\in\mathcal{B}} \mathbb{P}(s'|s, a, b)\widehat{\boldsymbol{y}}_s(b)$ represents the transition probability to state $s'$ when the min-player takes action $a$ and the max-player plays the stationary policy $\widehat{\boldsymbol{y}}$. Similarly, define $\widehat{\boldsymbol{R}}_s(a, 1) = \sum_{b\in\mathcal{B}} \boldsymbol{R}_s(a, b)\widehat{\boldsymbol{y}}_s(b)$ as the marginal reward function that the min-player will receive when its opponent chooses the stationary policy $\widehat{\boldsymbol{y}}$.

Denote the one-sided NE set of the min-player in the new Markov game $\mathcal{MG}'$ by $\mathcal{X}^*(\mathcal{MG}')$. By definition, the minimax game values $\widehat{v}^*$ of $\mathcal{MG}'$ are $\widehat{v}^*(s) = V^{\dagger,\widehat{\boldsymbol{y}}}(s)$. Then, for any $\boldsymbol{x}^* \in \mathcal{X}^*(\mathcal{MG}')$, $V^{\boldsymbol{x}^*,\widehat{\boldsymbol{y}}}(s) = V^{\dagger,\widehat{\boldsymbol{y}}}(s)$ for any $s \in \mathcal{S}$. Equivalently, $\mathcal{X}^*(\mathcal{MG}')$ is the best response set of $\widehat{\boldsymbol{y}}$.

By applying Theorem 2 to the new Makov game $\mathcal{MG}'$, we have that the policy $\boldsymbol{x}^t$ played by the min-player will converge at a global linear rate to $\mathcal{X}^*(\mathcal{MG}')$ that is the best response set of $\widehat{\boldsymbol{y}}$. Similar arguments also hold for the max-player. This gives the rationality. $\square$

---

**Algorithm 6:** Instantiation of `Homotopy-PO` with Averaging OGDA and OGDA (min-player's perspective)

---

**Input:** iterations: $[0 : T]$, initial policy: $\boldsymbol{x}^0 \in \mathcal{X}$, stepsizes: $\eta, \eta' > 0$
set $k = 1, \widetilde{\mathcal{I}}_{\text{lf}}^0 = -1, \boldsymbol{x}^{-1} = \boldsymbol{x}^0$
**while** $\widetilde{\mathcal{I}}_{\text{lf}}^{k-1} < T$ **do**

    $\mathcal{I}_{\text{gs}}^k = \widetilde{\mathcal{I}}_{\text{lf}}^{k-1} + 1, \widetilde{\mathcal{I}}_{\text{gs}}^k = \min\{\mathcal{I}_{\text{gs}}^k + 2^k - 1, T\}, \mathcal{I}_{\text{lf}}^k = \widetilde{\mathcal{I}}_{\text{gs}}^k + 1, \widetilde{\mathcal{I}}_{\text{lf}}^k = \min\{\mathcal{I}_{\text{lf}}^k + 4^k - 1, T\}$

    during time interval $[\mathcal{I}_{\text{gs}}^k : \widetilde{\mathcal{I}}_{\text{gs}}^k]$, run `x-Averaging-OGDA`$([\mathcal{I}_{\text{gs}}^k : \widetilde{\mathcal{I}}_{\text{gs}}^k], \boldsymbol{x}^{\widetilde{\mathcal{I}}_{\text{lf}}^{k-1}}, \eta')$ and

     compute an average policy $\widehat{\boldsymbol{x}}^{[\mathcal{I}_{\text{gs}}^k : \widetilde{\mathcal{I}}_{\text{gs}}^k]}$ (Algorithm 4)

    during time interval $[\mathcal{I}_{\text{lf}}^k : \widetilde{\mathcal{I}}_{\text{lf}}^k]$, run `x-OGDA`$([\mathcal{I}_{\text{lf}}^k : \widetilde{\mathcal{I}}_{\text{lf}}^k], \widehat{\boldsymbol{x}}^{[\mathcal{I}_{\text{gs}}^k : \widetilde{\mathcal{I}}_{\text{gs}}^k]}, \eta)$ (Algorithm 2)

    $k \leftarrow k + 1$

**end**

---

---

**Algorithm 7:** Instantiation of `Homotopy-PO` with Averaging OGDA and OGDA (max-player's perspective)

---

**Input:** iterations: $[0:T]$, initial policy: $\boldsymbol{y}^0 \in \mathcal{Y}$, stepsizes: $\eta, \eta' > 0$

set $k = 1$, $\widetilde{\mathcal{I}}_{\text{lf}}^0 = -1$, $\boldsymbol{y}^{-1} = \boldsymbol{y}^0$

**while** $\widetilde{\mathcal{I}}_{\text{lf}}^{k-1} < T$ **do**

    $\mathcal{I}_{\text{gs}}^k = \widetilde{\mathcal{I}}_{\text{lf}}^{k-1} + 1$, $\widetilde{\mathcal{I}}_{\text{gs}}^k = \min\{\mathcal{I}_{\text{gs}}^k + 2^k - 1, T\}$, $\mathcal{I}_{\text{lf}}^k = \widetilde{\mathcal{I}}_{\text{gs}}^k + 1$, $\widetilde{\mathcal{I}}_{\text{lf}}^k = \min\{\mathcal{I}_{\text{lf}}^k + 4^k - 1, T\}$

    during time interval $[\mathcal{I}_{\text{gs}}^k : \widetilde{\mathcal{I}}_{\text{gs}}^k]$, run `y-Averaging-OGDA`$([\mathcal{I}_{\text{gs}}^k : \widetilde{\mathcal{I}}_{\text{gs}}^k], \boldsymbol{y}^{\widetilde{\mathcal{I}}_{\text{lf}}^{k-1}}, \eta')$ and

      compute an average policy $\widehat{\boldsymbol{y}}^{[\mathcal{I}_{\text{gs}}^k : \widetilde{\mathcal{I}}_{\text{gs}}^k]}$ (Algorithm 5)

    during time interval $[\mathcal{I}_{\text{lf}}^k : \widetilde{\mathcal{I}}_{\text{lf}}^k]$, run `y-OGDA`$([\mathcal{I}_{\text{lf}}^k : \widetilde{\mathcal{I}}_{\text{lf}}^k], \widehat{\boldsymbol{y}}^{[\mathcal{I}_{\text{gs}}^k : \widetilde{\mathcal{I}}_{\text{gs}}^k]}, \eta)$ (Algorithm 3)

    $k \leftarrow k + 1$

**end**

---

# H  NATURAL GENERALIZATION OF GLOBAL-SLOW WITH MORE EXAMPLE

In this section, we mainly (1) show the convergence results of `Homotopy-PO` when Global-Slow base algorithm has different rates on the RHS of (2); (2) provide another example of Global-Slow base algorithm with generalized global convergence rates by proving the geometric boundedness of Algorithm 1 in Wei et al. (2021) with a slightly modified initialization.

## H.1  CONVERGENCE RESULT OF HOMOTOPY-PO WHEN GLOBAL-SLOW HAS DIFFERENT CONVERGENCE RATES

To avoid abuse of notations, we call the `Global-Slow` algorithm with more general global convergence rates by *Generalized Global-Slow base algorithm*.

**Generalized Global-Slow base algorithm:** by calling `Gen-Global-Slow`$([T_1 : T_2], \tilde{\boldsymbol{z}}, \eta')$ during time interval $[T_1 : T_2]$ where $\tilde{\boldsymbol{z}} = (\tilde{\boldsymbol{x}}, \tilde{\boldsymbol{y}})$ is the initial policy pair, the players play policy pair $\boldsymbol{z}^t = (\boldsymbol{x}^t, \boldsymbol{y}^t)$ for each iteration $t \in [T_1 : T_2]$, and compute a policy pair $\widehat{\boldsymbol{z}}^{[T_1:T_2]} = (\widehat{\boldsymbol{x}}^{[T_1:T_2]}, \widehat{\boldsymbol{y}}^{[T_1:T_2]})$ at the end of iteration $T_2$ such that $\boldsymbol{z}^t, \widehat{\boldsymbol{z}}^{[T_1:T_2]}$ satisfy the following two properties:

• **global convergence**: there is a problem-dependent constant $\widehat{C}' > 0$ and real numbers $p_1 > 0$ and $p_2, p_3 \geq 0$ such that

$$\text{dist}(\widehat{\boldsymbol{z}}^{[T_1:T_2]}, \mathcal{Z}^*) \leq \frac{\widehat{C}' \log^{p_3}(T_2 - T_1 + 1)}{\eta'^{p_2}(T_2 - T_1 + 1)^{p_1}}, \tag{99}$$

• **geometric boundedness**: there exists a problem-dependent constant $\widehat{D}_0 > 0$ (possibly $\widehat{D}_0 > 1$) such that if $\eta' \leq 1$, then for any $t \in [T_1 : T_2]$,

$$\text{dist}^2(\boldsymbol{z}^t, \mathcal{Z}^*) \leq \widehat{D}_0^{t-T_1} \cdot \text{dist}^2(\tilde{\boldsymbol{z}}, \mathcal{Z}^*),$$
$$\text{dist}^2(\widehat{\boldsymbol{z}}^{[T_1:T_2]}, \mathcal{Z}^*) \leq \widehat{D}_0^{T_2-T_1} \cdot \text{dist}^2(\tilde{\boldsymbol{z}}, \mathcal{Z}^*).$$

The main difference between `Gen-Global-Slow` and `Global-Slow` is that (1) the RHS of (99) in the definition of `Gen-Global-Slow` add more flexibility in the power numbers then the condition (2) in the definition of `Global-Slow`; (2) $\widehat{\boldsymbol{z}}^{[T_1:T_2]}$ need not to be an average policy. In the example (104) below, we can simply set $\widehat{\boldsymbol{z}}^{[T_1:T_2]} = \boldsymbol{z}^{T_2}$.

By similar arguments with Theorem 1, we have the following convergence rates for `Homotopy-PO` with generalized `Global-Slow`.

**Theorem 7** *Let $\{\boldsymbol{z}^t = (\boldsymbol{x}^t, \boldsymbol{y}^t)\}_{t \in [0:T]}$ be the policy pairs played when running `Homotopy-PO` (Algorithm 1) where `Global-Slow` is replaced by `Gen-Global-Slow`. Then, for any $t \in [0 :$*

$T$], *we have*

$$\text{dist}^2(\boldsymbol{z}^t, \mathcal{Z}^*) \le 2S \max\{\Gamma_0, 1\} \cdot \left(1 - \frac{c_0\eta^2}{48}\right)^{t - 128\widehat{M}^*/3}, \tag{100}$$

*where the value of* $\widehat{C}', c_0, \delta_0, \Gamma_0$ *can be found in the definitions of* `Gen-Global-Slow` *and* `Local-Fast` *and*

$$\widehat{M}^* = O\left(\left(\frac{\left(\widehat{C}'\right)^2 \log^{2p_3}(\widehat{C}'/(\delta_0\eta\eta'))}{\delta_0\eta^4\eta'^{2p_2}}\right)^{\frac{1}{p_1}} + \frac{\log^2(\widehat{D}_0 + 1)}{c_0^2\eta^4}\right). \tag{101}$$

*Proof of Theorem 7.* Let $c_0, \delta_0$ be defined in the local linear convergence of `Local-Fast`, $D_0$ defined in the geometric boundedness of `Gen-Global-Slow`, $\widehat{C}'$ defined in the global convergence of `Gen-Global-Slow`.

Define

$$\widehat{M}_1^* = \min\left\{t \ge 1 : \frac{\widehat{C}' \log^{p_3}(t)}{\eta'^{p_2} t^{p_1}} \le \sqrt{\delta_0\eta^4}\right\}, \tag{102}$$

$M_2^*$ and $M_3^*$ are defined the same as in the proof of Theorem 1 in Appendix E.

Analogous to the proof of Theorem 1, we also let $\widehat{M}^* = \max\left\{(M_1^*)^2, M_2^*, (M_3^*)^2\right\}$. This gives the order of $\widehat{M}^*$ in (101).

Notice that the global linear rate only depends on the local linear rate of `Local-Fast` and the geometric boundedness of `Global-Slow`. The global convergence rate of `Gen-Global-Slow` is only relevant to the length of Hidden Phase I, i.e., $\widehat{M}_1^*$ will only affect the length of Hidden Phase I. Then the rest of this proof follows from Theorem 1 directly. Analogously to (88), we also have

$$\text{dist}^2(\boldsymbol{z}^t, \mathcal{Z}^*) \le 2S \max\{\Gamma_0, 1\} \cdot \left(1 - \frac{c_0\eta^2}{48}\right)^{t - 128\widehat{M}^*/3}.$$

This gives the convergence result of `Homotopy-PO` when equipped with `Gen-Global-Slow` and `Local-Fast`. □

## H.2 ANOTHER EXAMPLE OF GLOBAL-SLOW BASE ALGORITHM

Next, we show that the algorithm in Wei et al. (2021) with a slightly modified initialization can serve as an example of `Gen-Global-Slow`. It is shown in Theorem 2 of Wei et al. (2021) that Algorithm 1 therein has a sub-linear last-iterate global convergence rate which satisfies the RHS of (99) with $p_1 = \frac{1}{2}, p_2 = 2, p_3 = 0$. To instantiate that Algorithm 1 in Wei et al. (2021) can be an example of `Gen-Global-Slow`, it suffices to prove its geometric boundedness. We remark that geometric boundedness may not hold for the original Algorithm 1 in Wei et al. (2021) since its initialization $V^0(s) = 0$ may cause the policy gradients in the first step to deviate largely. However, this problem can be fixed simply by changing the initialization to $V^0(s) = V^{\boldsymbol{x}^1, \boldsymbol{y}^1}(s)$.

When running Algorithm 1 of Wei et al. (2021) in the full-information setting (with the different initialization discussed above) during the time interval $[T_1 : T_2]$, the min-player and the max-player initialize $\widetilde{\boldsymbol{x}}^{T_1} = \boldsymbol{x}^{T_1} = \tilde{\boldsymbol{x}}, \widetilde{\boldsymbol{y}}^{T_1} = \boldsymbol{y}^{T_1} = \tilde{\boldsymbol{y}}$ and

$$V^{T_1 - 1}(s) = V^{\boldsymbol{x}^{T_1}, \boldsymbol{y}^{T_1}}(s) \tag{103}$$

and update for $t \geq T_1$ and any $s \in \mathcal{S}$

$$\widetilde{\boldsymbol{x}}_s^{t+1} = \mathcal{P}_{\mathcal{X}}\left(\widetilde{\boldsymbol{x}}_s^t - \eta \widehat{\boldsymbol{Q}}_s^t \boldsymbol{y}_s^t\right), \tag{104a}$$

$$\boldsymbol{x}_s^{t+1} = \mathcal{P}_{\mathcal{X}}\left(\widetilde{\boldsymbol{x}}_s^{t+1} - \eta \widehat{\boldsymbol{Q}}_s^t \boldsymbol{y}_s^t\right), \tag{104b}$$

$$\widetilde{\boldsymbol{y}}_s^{t+1} = \mathcal{P}_{\mathcal{Y}}\left(\widetilde{\boldsymbol{y}}_s^t + \eta \left(\widehat{\boldsymbol{Q}}_s^t\right)^\top \boldsymbol{x}_s^t\right), \tag{104c}$$

$$\boldsymbol{y}_s^{t+1} = \mathcal{P}_{\mathcal{Y}}\left(\widetilde{\boldsymbol{y}}_s^{t+1} + \eta \left(\widehat{\boldsymbol{Q}}_s^t\right)^\top \boldsymbol{x}_s^t\right), \tag{104d}$$

$$V^t(s) = (1 - \beta_{t-T_1+1}) V^{t-1}(s) + \beta_{t-T_1+1} \left\langle \boldsymbol{x}_s^t, \widehat{\boldsymbol{Q}}_s^t \boldsymbol{y}_s^t \right\rangle, \tag{104e}$$

where $\widehat{\boldsymbol{Q}}_s^t = \boldsymbol{Q}_s[V^{t-1}]$ and $\beta_t = \frac{H_0+1}{H_0+t}$ with $H_0 = \left\lceil \frac{2}{1-\gamma} \right\rceil$. Recall that $\boldsymbol{Q}_s[\cdot]$ is the Bellman target operator defined in the introduction.

When using the algorithm (104) with initialization (103), the output policy can be set as

$$\widehat{\boldsymbol{x}}^{[T_1:T_2]} = \boldsymbol{x}^{T_2}, \quad \widehat{\boldsymbol{y}}^{[T_1:T_2]} = \boldsymbol{y}^{T_2}.$$

We also denote $\boldsymbol{z}^t = (\boldsymbol{x}^t, \boldsymbol{y}^t)$, $\widehat{\boldsymbol{z}}^{[T_1:T_2]} = (\widehat{\boldsymbol{x}}^{[T_1:T_2]}, \widehat{\boldsymbol{y}}^{[T_1:T_2]})$, $\boldsymbol{x}^{t*} = \mathcal{P}_{\mathcal{X}^*}(\boldsymbol{x}^t)$, $\boldsymbol{y}^{t*} = \mathcal{P}_{\mathcal{Y}^*}(\boldsymbol{y}^t)$, $\boldsymbol{z}^{t*} = \mathcal{P}_{\mathcal{Z}^*}(\boldsymbol{z}) = (\boldsymbol{x}^{t*}, \boldsymbol{y}^{t*})$ in the analysis below.

Next, we proceed to show the geometric boundedness of the algorithm of Wei et al. (2021) with the slightly modified initialization in a similar way with Appendix D.2. We first provide mutual bounds among $\{\text{dist}(\boldsymbol{z}^t, \mathcal{Z}^*)\}$ and $\{\|V^t(s) - v^*(s)\|_\infty\}$ in Lemma 23 and Lemma 24 below.

**Lemma 23** *Let $\{\boldsymbol{z}^t, V^t\}$ be generated from* (104) *with initialization* (103). *For any $t \geq T_1$,*

$$\left\|V^t - v^*\right\|_\infty \leq \max_{j \in [T_1:t]} \frac{\sqrt{A+B}\,\text{dist}(\boldsymbol{z}^j, \mathcal{Z}^*)}{1-\gamma} + \max_{j \in [T_1-1:t-1]} \left\|V^j - v^*\right\|_\infty.$$

*Proof of Lemma 23.* Firstly, define $\beta_t^j = \beta_j \Pi_{k=j+1}^t (1 - \beta_k)$ for $0 \leq j \leq t-1$ and $\beta_t^t = \beta_t$. Since $\beta_t^0 = 0$, by (104e), for any $t \geq T_1$

$$V^t(s) = \sum_{j=T_1}^t \beta_{t-T_1+1}^{j-T_1+1} \left\langle \boldsymbol{x}_s^j, \widehat{\boldsymbol{Q}}_s^j \boldsymbol{y}_s^j \right\rangle.$$

By the definition of $\widehat{\boldsymbol{Q}}_s^j$, we have

$$\max_{s,a,b}\left|\widehat{\boldsymbol{Q}}_s^j(a,b) - \boldsymbol{Q}_s^*(a,b)\right| = \max_{s,a,b}\left|\widehat{\boldsymbol{Q}}_s^j(a,b) - \boldsymbol{Q}_s[v^*](a,b)\right| \leq \left\|V^{j-1} - v^*\right\|_\infty. \tag{105}$$

By Lemma 19, $v^*(s) = \left\langle \boldsymbol{x}_s^{j*}, \boldsymbol{Q}_s[v^*]\boldsymbol{y}_s^{j*} \right\rangle$. Thus, for any $t \geq T_1$ and $s \in \mathcal{S}$, by combining the above equations, we have

$$\left|V^t(s) - v^*(s)\right| \leq \sum_{j=T_1}^t \beta_{t-T_1+1}^{j-T_1+1} \left|\left\langle \boldsymbol{x}_s^j, \widehat{\boldsymbol{Q}}_s^j \boldsymbol{y}_s^j \right\rangle - \left\langle \boldsymbol{x}_s^{j*}, \boldsymbol{Q}_s[v^*]\boldsymbol{y}_s^{j*} \right\rangle\right|$$

$$\leq \sum_{j=T_1}^t \beta_{t-T_1+1}^{j-T_1+1} \left\|\boldsymbol{x}_s^{j*}\right\|_1 \cdot \max_{(a,b)\in\mathcal{A}\times\mathcal{B}} |\boldsymbol{Q}_s[v^*](a,b)| \cdot \left\|\boldsymbol{y}_s^j - \boldsymbol{y}_s^{j*}\right\|_1$$

$$+ \sum_{j=1}^t \beta_{t-T_1+1}^{j-T_1+1} \left\|\boldsymbol{x}_s^{j*}\right\|_1 \cdot \max_{(a,b)\in\mathcal{A}\times\mathcal{B}} \left|\widehat{\boldsymbol{Q}}_s^j(a,b) - \boldsymbol{Q}_s[v^*](a,b)\right| \cdot \left\|\boldsymbol{y}_s^j\right\|_1$$

$$+ \sum_{j=T_1}^t \beta_{t-T_1+1}^{j-T_1+1} \left\|\boldsymbol{x}_s^j - \boldsymbol{x}_s^{j*}\right\|_1 \left\|\widehat{\boldsymbol{Q}}_s^j \boldsymbol{y}_s^j\right\|_\infty$$

$$\leq \sum_{j=T_1}^t \beta_{t-T_1+1}^{j-T_1+1} \left(\frac{\left\|\boldsymbol{z}_s^j - \boldsymbol{z}_s^{j*}\right\|_1}{1-\gamma} + \left\|V^{j-1} - v^*\right\|_\infty\right).$$

Then, the proof is completed by the fact that $\max_{s \in \mathcal{S}} \left\| z_s^j - z_s^{j*} \right\|_1 \le \sqrt{A+B} \text{dist}(z^j, \mathcal{Z}^*)$ and $\sum_{j=T_1}^t \beta_{t-T_1+1}^{j-T_1+1} = 1$. $\qquad\square$

**Lemma 24** *Let* $\{z^t, V^t\}$ *be generated from* (104) *with initialization* (103). *Then, for any* $t \ge T_1$,

$$\text{dist}^2(\widetilde{z}^{t+1}, \mathcal{Z}^*) \le 18\text{dist}^2(\widetilde{z}^t, \mathcal{Z}^*) + 8\eta^2 S(A+B) \left\| V^{t-1} - v^* \right\|_\infty^2$$
$$+ 8\eta^2 \frac{\max\{A, B\}^2}{(1-\gamma)^2} \text{dist}^2\left(z^t, \mathcal{Z}^*\right),$$

$$\text{dist}^2(z^{t+1}, \mathcal{Z}^*) \le 324\text{dist}^2(\widetilde{z}^t, \mathcal{Z}^*) + 152\eta^2 S(A+B) \left\| V^{t-1} - v^* \right\|_\infty^2$$
$$+ 152\eta^2 \frac{\max\{A, B\}^2}{(1-\gamma)^2} \text{dist}^2\left(z^t, \mathcal{Z}^*\right).$$

*Proof of Lemma 24.* By applying Lemma 2 to (104) and substituting (105), we have

$$\left\| \widetilde{z}^{t+1} - \widetilde{z}^t \right\|^2 \le 8\text{dist}^2\left(\widetilde{z}^t, \mathcal{Z}^*\right) + 4\eta^2 S(A+B) \left\| V^{t-1} - v^* \right\|_\infty^2$$
$$+ 4\eta^2 \frac{\max\{A, B\}^2}{(1-\gamma)^2} \text{dist}^2\left(z^t, \mathcal{Z}^*\right).$$

$$\left\| z^{t+1} - \widetilde{z}^{t+1} \right\|^2 \le 8\text{dist}^2\left(\widetilde{z}^{t+1}, \mathcal{Z}^*\right) + 4\eta^2 S(A+B) \left\| V^{t-1} - v^* \right\|_\infty^2$$
$$+ 4\eta^2 \frac{\max\{A, B\}^2}{(1-\gamma)^2} \text{dist}^2\left(z^t, \mathcal{Z}^*\right).$$

The bound of $\text{dist}^2(\widetilde{z}^{t+1}, \mathcal{Z}^*)$ follows by the fact that $\text{dist}^2(\widetilde{z}^{t+1}, \mathcal{Z}^*) \le 2\text{dist}^2(\widetilde{z}^t, \mathcal{Z}^*) + 2\left\| \widetilde{z}^{t+1} - \widetilde{z}^t \right\|^2$. The bound of $\text{dist}^2(z^{t+1}, \mathcal{Z}^*)$ follows by the fact that $\text{dist}^2\left(z^{t+1}, \mathcal{Z}^*\right) \le 2\text{dist}^2\left(\widetilde{z}^{t+1}, \mathcal{Z}^*\right) + 2\left\| z^{t+1} - \widetilde{z}^{t+1} \right\|^2$ and substituting the bound for $\text{dist}^2(\widetilde{z}^{t+1}, \mathcal{Z}^*)$. $\qquad\square$

Next, we show the geometric boundedness of Algorithm 1 in Wei et al. (2021) with the initialization (103).

**Theorem 8** *(Geometric Boundedness of Algorithm 1 in Wei et al. (2021)) Let* $\{z^t\}_{t \in [T_1:T_2]}$ *be the policy pairs played by running the algorithm* (104) *with initialization* (103). *If* $\eta \le 1$, *then there is a problem-dependent constant* $\widehat{D}_0 = O(\frac{S(A+B)^2}{(1-\gamma)^4})$ *(possibly* $\widehat{D}_0 > 1$*) such that for any* $t \in [T_1 : T_2]$,

$$\text{dist}^2(z^t, \mathcal{Z}^*) \le \widehat{D}_0^{t-T_1} \cdot \text{dist}^2(\tilde{z}, \mathcal{Z}^*). \tag{106}$$

*Proof of Theorem 8.* We will show (106) by proving (112) inductively.

Firstly, we define some constants which are used in the definition of $\widehat{D}_0$. By Lemma 23, for $t \ge T_1$,

$$\left\| V^t - v^* \right\|_\infty^2 \le \max_{j \in [T_1:t]} C_1' \cdot \text{dist}^2(z^j, \mathcal{Z}^*) + 2 \max_{j \in [T_1-1:t-1]} \left\| V^j - v^* \right\|_\infty^2, \tag{107}$$

where

$$C_1' = \frac{2(A+B)}{(1-\gamma)^2}.$$

By Lemma 24 and the condition $\eta \le 1$, for $t \ge T_1$,

$$\text{dist}^2(\widetilde{z}^{t+1}, \mathcal{Z}^*) \le D_1'\text{dist}^2(\widetilde{z}^t, \mathcal{Z}^*) + C_2' \left\| V^{t-1} - v^* \right\|_\infty^2$$
$$+ C_3'\text{dist}^2\left(z^t, \mathcal{Z}^*\right), \tag{108}$$

$$\text{dist}^2(z^{t+1}, \mathcal{Z}^*) \le D_2'\text{dist}^2(\widetilde{z}^t, \mathcal{Z}^*) + C_4' \left\| V^{t-1} - v^* \right\|_\infty^2$$
$$+ C_5'\text{dist}^2\left(z^t, \mathcal{Z}^*\right), \tag{109}$$

where

$$D_1' = 18, \ C_2' = 8S(A+B), \ C_3' = \frac{8\max\{A,B\}^2}{(1-\gamma)^2},$$

$$D_2' = 324, \ C_4' = 152S(A+B), \ C_5' = \frac{152\max\{A,B\}^2}{(1-\gamma)^2}.$$

For the initialization (103), by (91),

$$\left\|V^{T_1-1} - v^*\right\|_\infty^2 \le C_6' \cdot \mathrm{dist}^2(\boldsymbol{z}^{T_1}, \mathcal{Z}^*), \tag{110}$$

where

$$C_6' = \frac{A+B}{(1-\gamma)^4}.$$

Define

$$\widehat{D}_0 = \max\{C_1' + 2, C_6', D_1' + C_2' + C_3', D_2' + C_4' + C_5'\}. \tag{111}$$

By definition, $\widehat{D}_0 \le O\left(\frac{S(A+B)^2}{(1-\gamma)^4}\right)$. Now, we proceed to prove (112) by induction.

$$\max\left\{\mathrm{dist}^2(\boldsymbol{z}^j, \mathcal{Z}^*), \mathrm{dist}^2(\widetilde{\boldsymbol{z}}^j, \mathcal{Z}^*), \left\|V^{j-1} - v^*\right\|_\infty^2\right\} \le \widehat{D}_0^{j-T_1} \cdot \mathrm{dist}^2(\widetilde{\boldsymbol{z}}, \mathcal{Z}^*). \tag{112}$$

The case of $j = T_1$ follows by (110) and the initialization $\boldsymbol{z}^{T_1} = \widetilde{\boldsymbol{z}}^{T_1} = \widetilde{\boldsymbol{z}}$.

Suppose we have shown (112) for $j \in [T_1 : t]$. Then, by (107), the fact $\widehat{D}_0 \ge 1$ and induction hypothesis,

$$\left\|V^t - v^*\right\|_\infty^2 \le (C_1' + 2)\widehat{D}_0^{t-T_1} \cdot \mathrm{dist}^2(\widetilde{\boldsymbol{z}}, \mathcal{Z}^*) \le \widehat{D}_0^{t+1-T_1} \cdot \mathrm{dist}^2(\widetilde{\boldsymbol{z}}, \mathcal{Z}^*).$$

By (108) and induction hypothesis

$$\mathrm{dist}^2(\widetilde{\boldsymbol{z}}^{t+1}, \mathcal{Z}^*) \le (D_1' + C_2' + C_3') \cdot \widehat{D}_0^{t-T_1} \cdot \mathrm{dist}^2(\widetilde{\boldsymbol{z}}, \mathcal{Z}^*) \le \widehat{D}_0^{t+1-T_1} \cdot \mathrm{dist}^2(\widetilde{\boldsymbol{z}}, \mathcal{Z}^*).$$

Analogously, by (109) and induction hypothesis

$$\mathrm{dist}^2(\boldsymbol{z}^{t+1}, \mathcal{Z}^*) \le (D_2' + C_4' + C_5') \cdot \widehat{D}_0^{t-T_1} \cdot \mathrm{dist}^2(\widetilde{\boldsymbol{z}}, \mathcal{Z}^*) \le \widehat{D}_0^{t+1-T_1} \cdot \mathrm{dist}^2(\widetilde{\boldsymbol{z}}, \mathcal{Z}^*).$$

Thus, we have shown (112) for $j = t + 1$. By induction, (112) holds for any $j \in [T_1 : T_2]$, which implies (106) directly.

This completes the proof for the geometric boundedness of the algorithm (104) with the initialization (103). □

**Remark 3** *When the meta algorithm* `Homotopy-PO` *switches between Algorithm 1 of Wei et al. (2021) (with the slightly modified initialization) and OGDA (8), then by Theorem 7 and (57),*

$$\frac{c_0}{48} = O\left(\frac{(1-\gamma)^4 c_+^2}{S^3}\right).$$

*Then, if* $\eta = O(\frac{(1-\gamma)^{\frac{5}{2}}}{\sqrt{S}(A+B)})$ *for OGDA, the linear rate is*

$$1 - \frac{c_0\eta^2}{48} = 1 - O\left(\frac{(1-\gamma)^9 c_+^2}{S^4(A+B)^2}\right).$$

*As in Algorithm 1 of Wei et al. (2021), the stepsize $\eta$ therein has to satisfy $\eta \le \frac{(1-\gamma)^{\frac{5}{2}}}{10^4\sqrt{S}}$. By combining Theorem 1 of Wei et al. (2021) with (100), (101), (57), (58), if $\eta = O(\frac{(1-\gamma)^{\frac{5}{2}}}{\sqrt{S}(A+B)})$ for OGDA and $\eta' = O(\frac{(1-\gamma)^{\frac{5}{2}}}{\sqrt{S}})$ for Algorithm 1 in Wei et al. (2021), then the length of Hidden Phase I is of order $128\widehat{M}^*/3 = \widetilde{O}\left(\frac{S^{32}(A+B)^{10}}{c_+^{16}(1-\gamma)^{74}}\right).$*

## I  NUMERICAL EXPERIMENTS

In this section, we evaluate the numerical performance of `Homotopy-PO` where `Local-Fast` and `Global-Slow` are instantiated with OGDA and Averaging OGDA respectively.

**Markov game model.**  We generate a sequence of zero-sum Markov games randomly and independently in the way described below and test the performance of `Homotopy-PO` on each of the games. In each Markov game generated below, the number of states is $S = 10$, the min-player and max-player have $A = B = 10$ actions respectively, and the discount factor $\gamma = 0.99$. The reward functions $\{\boldsymbol{R}_s(a,b)\}_{s\in\mathcal{S},a\in\mathcal{A},b\in\mathcal{B}}$ are generated from uniform distribution on $[0,1]$ independently. To generate the transition kernel, for each $(s,a,b)$, we first choose an integer $i_{s,a,b}$ uniformly at random from $[S]$. Then, we choose a random subset $M_{s,a,b} \subseteq \mathcal{S}$ with $|M_{s,a,b}| = i_{s,a,b}$. Then for each $s' \in M_{s,a,b}$, we set $\widehat{\mathbb{P}}(s'|s,a,b)$ from uniform distribution on $[0,1]$ independently, and for $s' \in \mathcal{S} \backslash M_{s,a,b}$, we set $\widehat{\mathbb{P}}(s'|s,a,b) = 0$. Finally, we normalize $\mathbb{P}(s'|s,a,b) = \widehat{\mathbb{P}}(s'|s,a,b)/\sum_{s''\in\mathcal{S}}\widehat{\mathbb{P}}(s''|s,a,b)$ for each $(s,a,b)$ to get the transition kernel. For the initial policies, we first generate $\{\boldsymbol{u}_s\}_{s\in\mathcal{S}}$ with $\boldsymbol{u}_s(a)$ chosen from uniform distribution on $[0,1]$ for each $s \in \mathcal{S}$, $a \in \mathcal{A}$. Then, we normalize $\boldsymbol{x}_s^0 = \boldsymbol{u}_s/\|\boldsymbol{u}_s\|_1$ for each $s \in \mathcal{S}$. The initial policy $\{\boldsymbol{y}_s^0\}_{s\in\mathcal{S}}$ of the max-player is generated independently in the same way.

**Algorithm implementation.**  In all the experiments below, we set the stepsizes $\eta = 0.1$ in OGDA and also $\eta' = 0.1$ in Averaging OGDA. We find our algorithm has linear convergence in all the experiments with these stepsizes.

**Performance metric.**  We measure the closeness of $\boldsymbol{z}^t$ to the Nash equilibria set by the Nash gap $\max_{s\in\mathcal{S}} V^{\boldsymbol{x}^t,\dagger}(s) - V^{\dagger,\boldsymbol{y}^t}(s)$. By combining Lemma 7 and Corollary 1 with the fact that $\mathrm{dist}(\boldsymbol{z},\mathcal{Z}^*) \le \sqrt{S}\max_{s\in\mathcal{S}}\mathrm{dist}(\boldsymbol{z}_s,\mathcal{Z}_s^*)$, we have the following relation between the Nash gap $\max_{s\in\mathcal{S}} V^{\boldsymbol{x},\dagger}(s) - V^{\dagger,\boldsymbol{y}}(s)$ and the distance to the NE set $\mathrm{dist}(\boldsymbol{z},\mathcal{Z}^*)$: for any $\boldsymbol{z} = (\boldsymbol{x},\boldsymbol{y}) \in \mathcal{Z}$,

$$\frac{c_+}{\sqrt{S}} \cdot \mathrm{dist}(\boldsymbol{z},\mathcal{Z}^*) \le \max_{s\in\mathcal{S}} V^{\boldsymbol{x},\dagger}(s) - V^{\dagger,\boldsymbol{y}}(s) \le \frac{\max\{\sqrt{2A},\sqrt{2B}\}}{(1-\gamma)^2} \cdot \mathrm{dist}(\boldsymbol{z},\mathcal{Z}^*). \tag{113}$$

Thus, the linear convergence of $\mathrm{dist}(\boldsymbol{z}^t,\mathcal{Z}^*)$ is equivalent to the linear convergence of the Nash gap $\max_{s\in\mathcal{S}} V^{\boldsymbol{x}^t,\dagger}(s) - V^{\dagger,\boldsymbol{y}^t}(s)$ up to problem-dependent constants. In the figures below, $y$-axis represents the logarithmic of the Nash gap $\log\big(\max_{s\in\mathcal{S}} V^{\boldsymbol{x}^t,\dagger}(s) - V^{\dagger,\boldsymbol{y}^t}(s)\big)$, $x$-axis represents the iteration number.

**Remark 4** *As we can see, there are discontinuities when switching from Averaging OGDA to OGDA in the figures below. This is because Averaging OGDA is an averaging style method. Recall that the $y$-axis represents $\log\big(\max_{s\in\mathcal{S}} V^{\boldsymbol{x}^t,\dagger}(s) - V^{\dagger,\boldsymbol{y}^t}(s)\big)$. However, the initial policy pair of the $k$-th call of OGDA is the average policy $\widehat{\boldsymbol{z}}^{[\mathcal{I}_{\mathrm{gs}}^k:\widetilde{\mathcal{I}}_{\mathrm{gs}}^k]} = \sum_{t=\mathcal{I}_{\mathrm{gs}}^k}^{\widetilde{\mathcal{I}}_{\mathrm{gs}}^k} \alpha_{2^k}^{t-\mathcal{I}_{\mathrm{gs}}^k+1} \boldsymbol{z}^t$. Since it is quite possible that $\widehat{\boldsymbol{z}}^{[\mathcal{I}_{\mathrm{gs}}^k:\widetilde{\mathcal{I}}_{\mathrm{gs}}^k]} \ne \boldsymbol{z}^{\widetilde{\mathcal{I}}_{\mathrm{gs}}^k}$, there can be some discontinuities in the figures below when switching from Averaging OGDA to OGDA. On the other hand, our theoretical bound in Figure 1 is continuous because by setting $t = T_2$ in (3), theoretically $\mathrm{dist}^2(\boldsymbol{z}^{T_2},\mathcal{Z}^*) \le D_0^{T_2-T_1} \cdot \mathrm{dist}^2(\tilde{\boldsymbol{z}},\mathcal{Z}^*)$ whose bound equals the bound for $\mathrm{dist}^2(\widehat{\boldsymbol{z}}^{[T_1:T_2]},\mathcal{Z}^*)$ on the RHS of (4). We remark that in practice, it is predictable that $\boldsymbol{z}^{\widetilde{\mathcal{I}}_{\mathrm{gs}}^k} \ne \widehat{\boldsymbol{z}}^{[\mathcal{I}_{\mathrm{gs}}^k:\widetilde{\mathcal{I}}_{\mathrm{gs}}^k]}$ in most cases.*

**Numerical performance.**  We validate the linear convergence of our instantiation of `Homotopy-PO`, where `Global-Slow` and `Local-Fast` are instantiated by Averaging OGDA and OGDA respectively.

Figure 2 shows the performance when the min-player and max-player run Algorithm 6 and Algorithm 7, respectively. We do 10 random and independent trials and the algorithm exhibits linear convergence in every trial. The plot shows the average trajectory and standard deviation of the 10 random and independent trials. The vertical dotted line is at the end of 7-th call to OGDA (iteration $t = 22098$). As we can see, on the RHS of the dotted line (after $t > 22098$), the algorithm converges linearly and the Nash gap is less than $10^{-5}$ after $2 \times 10^5$ iterations. The standard deviation of the 10

random trials is illustrated by the shadow area. Since the switching pattern is $2^k$ iterations of Averaging OGDA followed by $4^k$ iterations of OGDA, Averaging OGDA is only run for 1022 iterations in the total $2 \times 10^5$ iterations. Thus, Averaging OGDA is hardly seen in Figure 2. We magnify the trajectory of the 9-th call to Averaging OGDA as a subfigure in Figure 2. We can find that Averaging OGDA increases in its 9-th call. This has been predicted in our theoretical bounds (see segment $\overline{BC}$ in Figure 1). The 8-th call to OGDA has $4^8$ iterations, while the 9-th call to Averaging OGDA only has $2^9$ iterations. We have $4^8/2^9 = 128$, i.e., the iterations of OGDA are hundreds of times more than those in the successive call to Averaging OGDA. Then the increase caused by Averaging OGDA can be naturally "omitted" compared with the decrease from OGDA. This aligns with our theoretical bounds in Figure 1 (see the relation between the segments $\overline{AB}$ and $\overline{BC}$ in Figure 1).

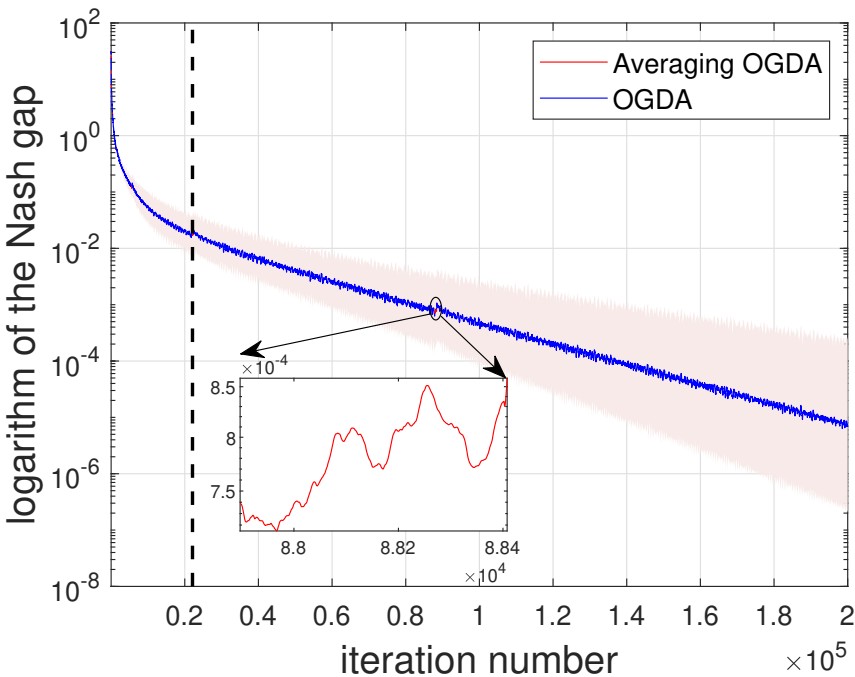

Figure 2: The numerical performance of `Homotopy-PO` when `Global-Slow` and `Local-Fast` are instantiated by Averaging OGDA and OGDA. The trajectory is the average of 10 random and independent trials. The $x$-axis represents the iteration number, while the $y$-axis represents the logarithm of the Nash gap. The shadow area shows the standard deviations of these trials. The vertical dotted line is drawn at the end of the 7-th call to OGDA (iteration $t = 22098$). On the RHS of the dotted line (equivalently, after $t > 22098$), the algorithm exhibits fast linear convergence. In our switching pattern, $2^k \ll 4^k$ when $k$ is large. Thus, Averaging OGDA is almost "invisible". We magnify the 9-th call to Averaging OGDA as a subfigure. Though Averaging OGDA can increase, its increase is negligible by the decrease from hundreds of times more steps of OGDA. This aligns with our theoretical guarantees (see the relation between segments $\overline{AB}$ and $\overline{BC}$ in Figure 1).

To avoid the problem that the iterations of Averaging OGDA is too few to be "visible", we do another group of trials by generalizing the switching pattern slightly. Recall that in Algorithm 1, the $k$-th call to `Global-Slow` has $2^k$ iterations while the $k$-th call to `Local-Fast` has $4^k$ iterations. It is worth noting that the choices of $2^k$ and $4^k$ in Algorithm 1 is only for simplicity. The proofs for linear convergence of `Homotopy-PO` can be directly generalized to the case when the $k$-th call to `Global-Slow` and `Local-Fast` has $\lceil u^k \rceil$ and $\lceil v^k \rceil$ iterations respectively whenever $u, v$ are real numbers satisfying $v > u > 1$. Then to see how `Homotopy-PO` switches between Averaging OGDA and OGDA and see the performance difference between Averaging OGDA and OGDA separately, we test the performance of `Homotopy-PO` where the $k$-th call to `Global-Slow` and `Local-Fast` has $2^k$ and $\lceil 2.1^k \rceil$ iterations respectively. We do another 10 random and independent trials in this switching pattern. The average trajectory and standard deviation are illustrated in

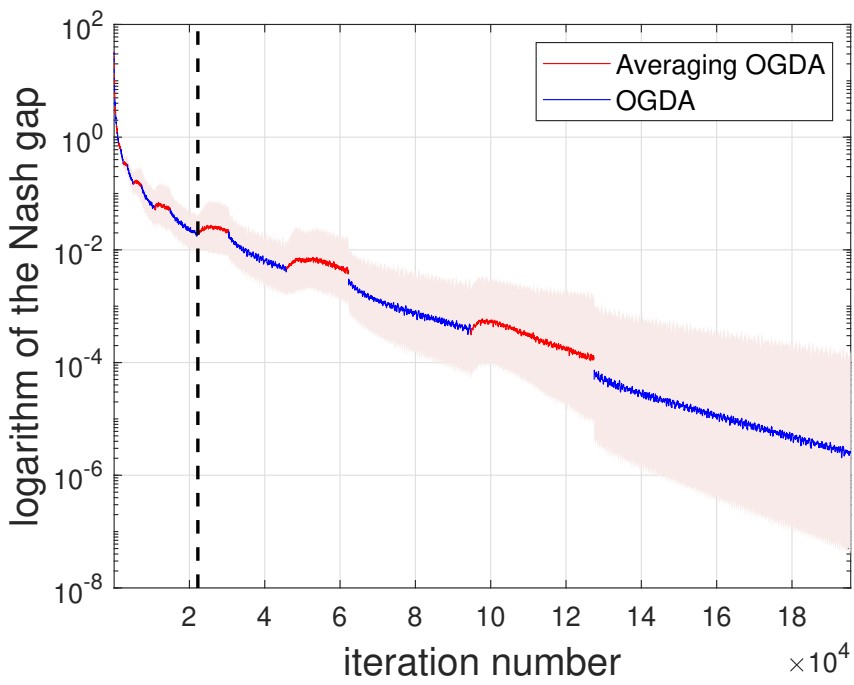

Figure 3: The numerical performance of `Homotopy-PO` with a slightly generalized switching scheme. In the new switching scheme, the $k$-th call to Averaging OGDA has $2^k$ steps and the $k$-th call to OGDA has $\lceil 2.1^k \rceil$ steps. In this way, there are more iterations of Averaging OGDA so that the switches between them can be seen more clearly. The trajectory is the average of 10 random and independent trials with this switching pattern. The shadow area shows the standard deviation of these trials. The $x$-axis represents the iteration number, while the $y$-axis represents the logarithm of the Nash gap. We show the trajectories of the first 15 calls of Averaging OGDA and OGDA (iterations $1 \leq t \leq 195592$) in this figure. The discontinuity in the trajectory is because Averaging OGDA is an averaging style method where $\widehat{z}^{[\mathcal{I}_{\mathrm{gs}}^k : \widetilde{\mathcal{I}}_{\mathrm{gs}}^k]}$ may not equal $z^{\widetilde{\mathcal{I}}_{\mathrm{gs}}^k}$ (see Remark 4). A vertical dotted line is drawn at the end of the 12-th call to OGDA (iteration $t = 22237$). It can be observed that on the RHS of the dotted line (iteration $t > 22237$), the algorithm exhibits linear convergence. This aligns with our theoretical bounds illustrated in Figure 1, where Averaging OGDA can increase but its increase can be "omitted" compared with the decrease from the more steps of OGDA so that the algorithm still has linear convergence.

Figure 3, where the iterations of Averaging OGDA are drawn in red while those of OGDA are drawn in blue. We show the trajectories of the first 15 calls of Averaging OGDA and OGDA (iterations $1 \leq t \leq 195592$) in Figure 3. The discontinuity of the trajectory is because Averaging OGDA is an averaging style method and OGDA uses the average policy $\widehat{z}^{[\mathcal{I}_{\mathrm{gs}}^k : \widetilde{\mathcal{I}}_{\mathrm{gs}}^k]}$ rather than $z^{\widetilde{\mathcal{I}}_{\mathrm{gs}}^k}$ as the initial policy (see Remark 4). We draw a vertical dotted line at the end of the 12-th call to OGDA (iteration $t = 22237$). It can be observed that on the RHS of the dotted line (after $t > 22237$), the algorithm exhibits linear convergence. On the RHS of the dotted line, the performance of Averaging OGDA is generally inferior to OGDA. Averaging OGDA can even increase in some iterations. This coincides with our theoretical bounds (see the segment $\overline{BC}$ in Figure 1). Thanks to the fast and efficient linear convergence of OGDA together with the fact that the iterations of Averaging OGDA take up less and less proportion in the total iterations, the algorithm can exhibit linear convergence on the RHS of the vertical dotted line. This also aligns with our theoretical bounds illustrated in Figure 1.

To see the switches between Averaging OGDA and OGDA clearly in each trial, in Figure 4 and Figure 5 below, we present the 10 random trials of the changed switching pattern ($2^k$ iterations of Averaging OGDA followed by $\lceil 2.1^k \rceil$ iterations of OGDA). We illustrate the trajectories of the first 15 calls of Averaging OGDA and OGDA (iterations $1 \leq t \leq 195592$) in Figure 4 and Figure 5.

In each subplots, we draw a vertical dotted line at the end of the 12-th call to OGDA (iteration $t = 22237$). It can be observed that on the RHS of the dotted line (after $t > 22237$), the algorithm has linear convergence in each trial. In some of the trials, Averaging OGDA can increase in some iterations. This is predicted (see segment $\overline{BC}$ in Figure 1). Since OGDA converges linearly and Averaging OGDA takes less and less proportion in the total iterations, the algorithm can still exhibit linear convergence on the RHS of the dotted line ($t > 22237$). This aligns with our theoretical bounds (see the relation between segments $\overline{AB}$ and $\overline{BC}$ in Figure 1). Even in the worst case (the 8-th trial), the Nash gap is less than $10^{-3}$ after $2 \times 10^5$ iterations. And in some fast cases such as the 3-rd, 4-th, 5-th, 9-th, 10-th trials, the Nash gap can be less than $10^{-6}$ or even $10^{-8}$ in about $2 \times 10^5$ iterations.

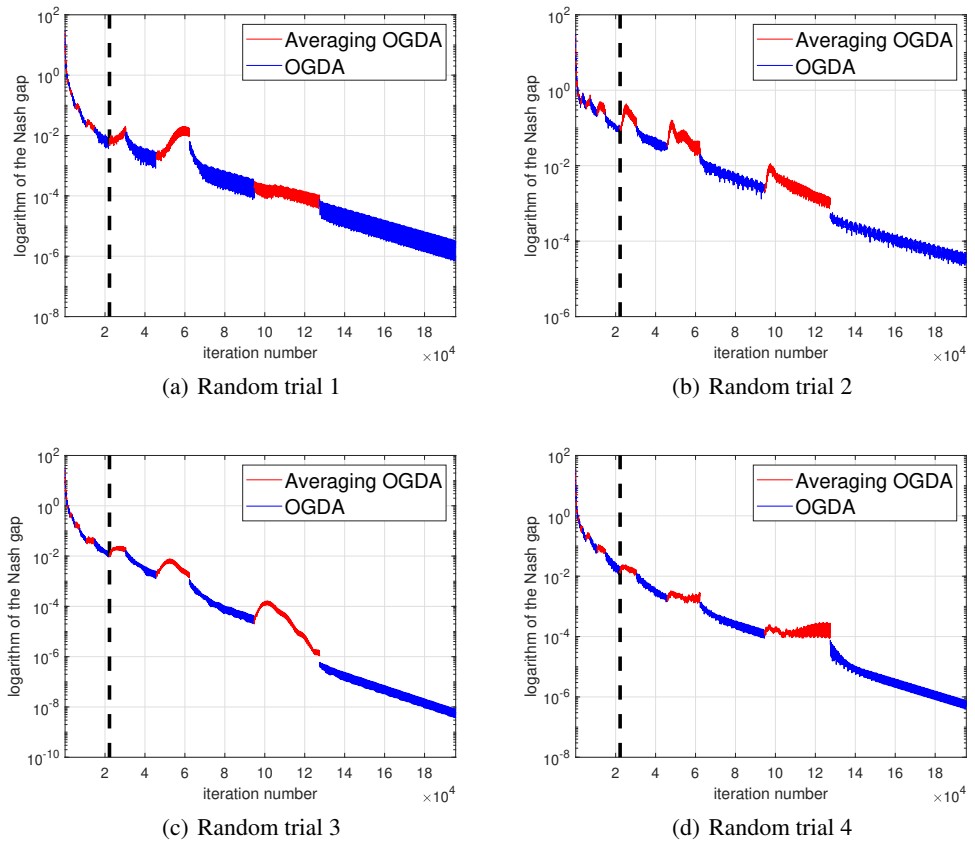

(a) Random trial 1                    (b) Random trial 2

(c) Random trial 3                    (d) Random trial 4

Figure 4: The first 4 trajectories of 10 random and independent trials with the switching pattern described for Figure 3. The rest 6 trajectories are illustrated in Figure 5 below. In these trials, the $k$-th call to Averaging OGDA and OGDA have $2^k$ and $\lceil 2.1^k \rceil$ iterations respectively so that the switches between them can be seen more clearly. The $x$-axis represents the iteration number, while the $y$-axis represents the logarithm of the Nash gap. We show the trajectories of the first 15 calls of Averaging OGDA and OGDA (iterations $1 \leq t \leq 195592$) in these subfigures. The vertical dotted line is drawn at the end of the 12-th call to OGDA (iteration $t = 22237$). As we can see, on the RHS of the vertical dotted line ($t > 22237$), all trajectories have linear convergence. The discontinuity is because Averaging OGDA is an averaging style method (see Remark 4). The trajectories coincides with our theoretical bounds in Figure 1 where although Averaging OGDA can cause increase, its increase can be "omitted" by the more steps of decrease from OGDA.

We also compare our algorithm with Alg. 1 in Wei et al. (2021). We choose the stepsizes of both our `Homotopy-PO` and Alg. 1 in Wei et al. (2021) to be $0.1$. We choose the discount factor $\gamma = 0.5$, and the rest settings are the same with those in the experiments above. The switching scheme is chosen to be the same with that in Figure 3 above. The comparison between `Homotopy-PO` and Alg. 1 in Wei et al. (2021) is illustrated in Figure 6, where the curves are drawn by taking the average over

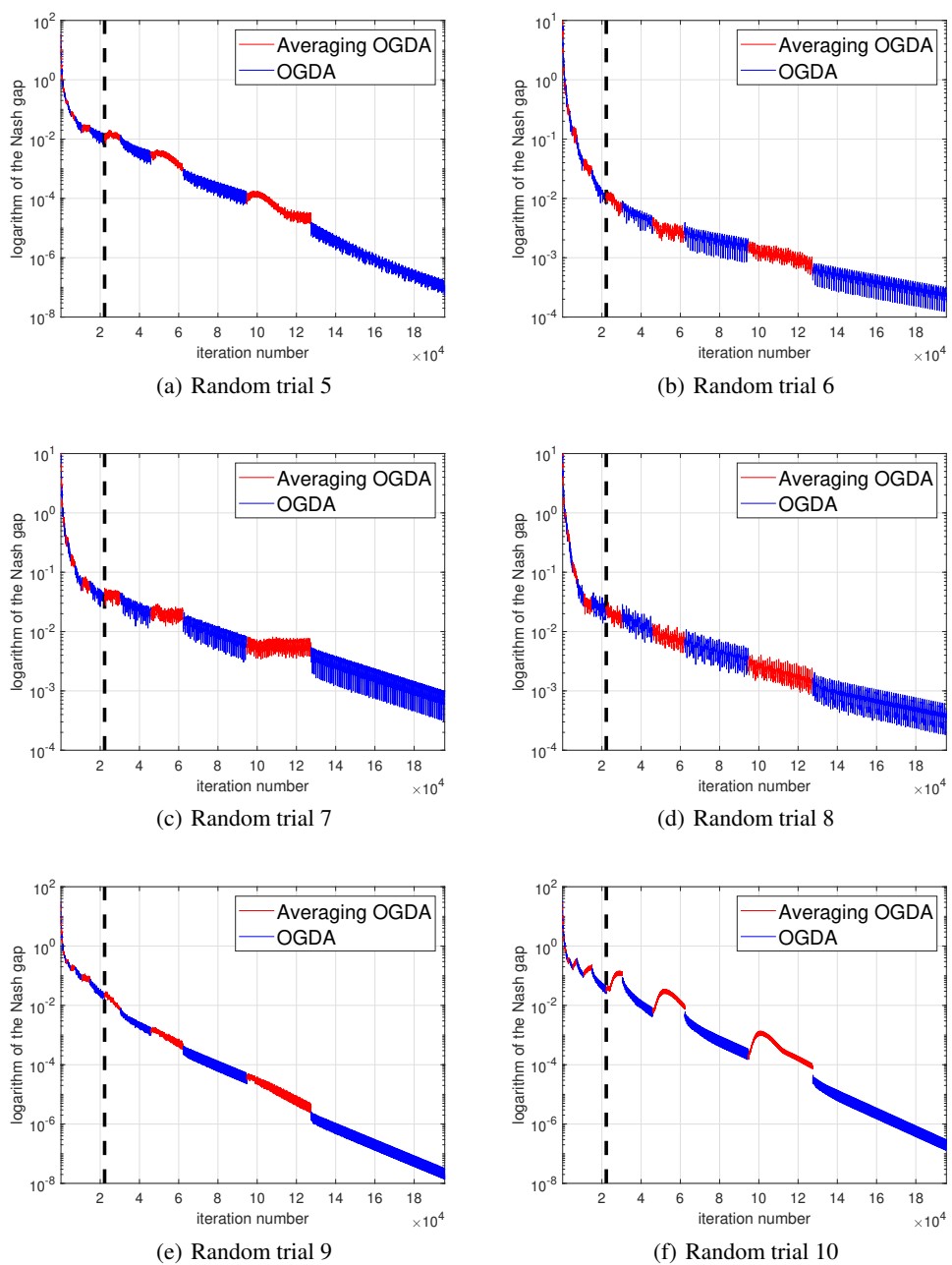

Figure 5: As complement to Figure 4, this figure shows the rest 6 trajectories of the 10 random and independent trials with the switching pattern described for Figure 3. The caption of this figure has been integrated into that of Figure 4.

5 random trajectories and connecting the points at the time points when `Homotopy-PO` switches between Averaging OGDA and OGDA. As we can see in Figure 6, `Homotopy-PO` can converge to the NE set faster than Alg. 1 in Wei et al. (2021).

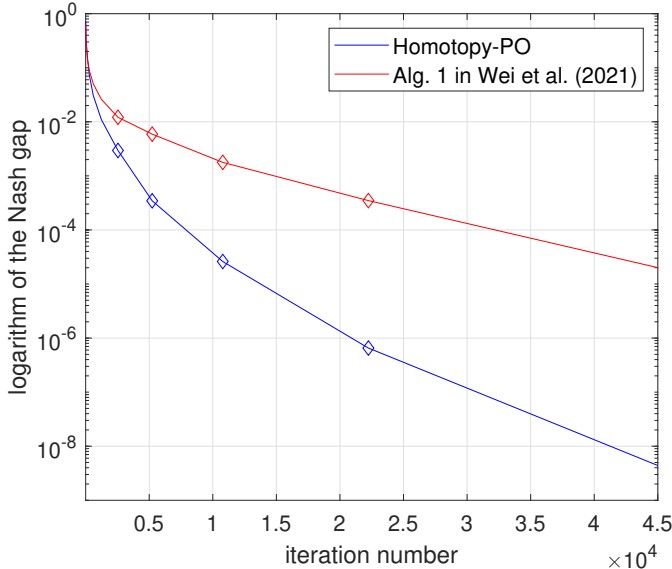

Figure 6: Comparison between `Homotopy-PO` with a slightly generalized switching scheme with Alg. 1 in Wei et al. (2021). In the new switching scheme, the $k$-th call to Averaging OGDA has $2^k$ steps and the $k$-th call to OGDA has $\lceil 2.1^k \rceil$ steps. The curves are computed from the average of 5 random and independent trials. The curves are the segments connecting the points at the time points when `Homotopy-PO` switches between Averaging OGDA and OGDA.

## J   DISCUSSIONS

### J.1   POSSIBLE TRANSLATION TO SAMPLE-BASED ALGORITHMS

We remark that it is possible to translate our algorithm into sample-based algorithms. Here, we tentatively discuss the analogues of local linear convergence of OGDA under the following two cases and give an intuitive analysis for each case. The analogues of global convergence and geometric boundedness of Averaging OGDA can be discussed similarly.

• **Case 1: Assuming access to a simulator (generative model).** If there is a simulator (generative model) and the players can draw lots of samples in one iteration, then it is possible to get linear convergence against the iteration number. More specifically, at iteration $t$, for each $s$, $N_t$ samples are drawn from the distributions $\mathbb{P}(\cdot|s, a^{t,j}, b^{t,j})$, where $1 \leq j \leq N_t$ and $a^{t,j} \sim x_s^t$, $b^{t,j} \sim y_s^t$. We define a very small variable $\delta_t = O(c_0 \eta^2 (1 - c_0 \eta^2)^t)$. Define the truncated simplex $\Delta_{\mathcal{A}}^t = \{x \in \mathbb{R}^A : x(a) \geq \delta_t, \sum x(a) = 1\}$. $\Delta_{\mathcal{B}}^t$ is defined analogously. At iteration $t$, we replace the projection operator $\mathcal{P}_{\Delta_{\mathcal{A}}}(\cdot)$ and $\mathcal{P}_{\Delta_{\mathcal{B}}}(\cdot)$ with $\mathcal{P}_{\Delta_{\mathcal{A}}^t}(\cdot)$ and $\mathcal{P}_{\Delta_{\mathcal{B}}^t}(\cdot)$. This guarantees that each action is taken with probability at least $\delta_t$. Then by Hoeffding's inequality, each action $a$ is taken by the min-player for at least $O(N_t \delta_t)$ times with high probability (w.h.p). Then, the empirical marginal reward $\widehat{r}_x^t$ and marginal transition kernel $\widehat{\mathbb{P}}_x^t$ observed by the min-player satisfy the following relation w.h.p.,

$$\|\widehat{r}_x^t - r_x^t\|_\infty \leq \widetilde{O}\left(\sqrt{\frac{1}{N_t \delta_t}}\right), \quad \|\widehat{\mathbb{P}}_x^t(\cdot|s, a) - \mathbb{P}_x^t(\cdot|s, a)\|_1 \leq \widetilde{O}\left(\sqrt{\frac{1}{N_t \delta_t}}\right).$$

In this remark, $\widetilde{O}(\cdot)$ suppresses logarithmic terms and problem parameters such as $S, A, B, 1/(1-\gamma)$ for simplicity. Thus, we have $\|\widehat{V}^{x^t, y^t} - V^{x^t, y^t}\|_\infty \leq \widetilde{O}(\sqrt{\frac{1}{N_t \delta_t}})$, $\|\widehat{V}^{\dagger, y^t} - V^{\dagger, y^t}\|_\infty \leq \widetilde{O}(\sqrt{\frac{1}{N_t \delta_t}})$, $\|\widehat{V}^{x^t, \dagger} - V^{x^t, \dagger}\|_\infty \leq \widetilde{O}(\sqrt{\frac{1}{N_t \delta_t}})$, $|\widehat{Q}_s^{x^t, y^t}(a, b) - Q_s^{x^t, y^t}(a, b)| \leq \widetilde{O}(\sqrt{\frac{1}{N_t \delta_t}})$ for any $(s, a, b)$. Here, we use $\widehat{\cdot}$ overhead to indicate the empirical quantities. And the replacement of $\mathcal{P}_{\Delta_{\mathcal{A}}}(\cdot)$ by $\mathcal{P}_{\Delta_{\mathcal{A}}^t}(\cdot)$ will add an error term whose $\ell_\infty$-norm is at most $\delta_t$. Thus in each iteration $t$, new error terms of order $\widetilde{O}(\sqrt{\frac{1}{N_t \delta_t}} + \delta_t)$ are added. At iteration $t$, let $\bar{x}^{t+1}$ be the ideal variable computed from $\{x^j\}_{j \leq t}$

with exact value of the marginal information $r_x^t$ and $\mathbb{P}_x^t$. Let $x^{t+1}$ be the real variable computed in the learning process. $\bar{y}^{t+1}$ and $y^{t+1}$ are defined similarly. Then, by Cauchy-Schwartz inequality, $dist^2(z^{t+1}, \mathcal{Z}^*) \leq (1 + c_0\eta^2/2)dist^2(\bar{z}^{t+1}, \mathcal{Z}^*) + (1 + 1/(2c_0\eta^2))\widetilde{O}(\frac{1}{N_t\delta_t} + \delta_t^2)$. After adding these error terms to the proof of Theorem 5, the bounds for the potential functions $\Lambda^t$ defined in (27) will be

$$\Lambda^{t+1} \leq (1 - c_0\eta^2)(1 + c_0\eta^2/2)\Lambda^t + \widetilde{O}\left(\frac{1}{c_0\eta^2}\left(\frac{1}{N_t\delta_t} + \delta_t^2\right)\right).$$

Then by setting $N_t = O(c_0^3\eta^6(1 - c_0\eta^2)^{2t})$, we can show by induction that $\Lambda^t \leq O((1 - c_0\eta^2/3)^t)$. This gives the local linear convergence of OGDA when the players can draw lots of samples in one iteration.

• **Case 2: Using an ergodic assumption.** When no simulator is available, we consider translating our algorithm into a sample-based algorithm under an ergodic assumption. The assumption is that there exists a constant $L_0 > 0$ such that for any policy pair $z = (x, y)$, if the min-player and the max-player play policy $x$ and $y$ respectively in $L_0$ successive iterations $t \in [T_0 : T_0 + L_0 - 1]$, then, for any initial state $s_{T_0}$ and state $s \in \mathcal{S}$, there exists a $t' \in [T_0 : T_0 + L_0 - 1]$ such that $s_{t'} = s$. Briefly, this assumption requires that when the players choose a stationary policy pair for successive $L_0$ iterations, then every state must be visited at least once in these $D_0$ iterations. Under this assumption, our strategy is to regard successive $L_0 \times N_k$ iterations as a virtual iteration $k$. In this way, we divide $[1 : T]$ into

$$[1 : T] = [T_1 : T_2] \cup [T_3 : T_4] \cup \cdots \cup [T_{2k-1} : T_{2k}] \cup \cdots$$

where $T_{2k} - T_{2k-1} = L_0 \times N_k$. Then, in the time interval $[T_{2k-1} : T_{2k}]$, each state $s$ has been visited for at least $N_k$ times. This is similar to the case when we have a simulator and $N_k$ samples are drawn for each state $s$ in iteration $k$. In this way, by applying our algorithm and analysis for the simulator case (Case 1 above), we can show the local linear convergence with respect to the virtual iteration number $k$.

### J.2 DISCUSSIONS ABOUT THE INTUITIONS BEHIND AVERAGING OGDA AND OGDA

• **In Markov games, the main challenge of finding an NE is to estimate the minimax game values** $\{v^*(s)\}$. If $\{v^*(s)\}$ are already known, the players can use $\boldsymbol{Q}_s[v^*]\boldsymbol{y}_s^t$ and $\boldsymbol{Q}_s[v^*]^\top\boldsymbol{x}_s^t$ as policy gradients to do optimistic gradient descent/ascent. Then finding an NE is reduced to solving $S$ matrix games $\min_{\boldsymbol{x}_s \in \Delta_\mathcal{A}} \max_{\boldsymbol{y}_s \in \Delta_\mathcal{B}} \boldsymbol{x}_s^\top \boldsymbol{Q}_s[v^*]\boldsymbol{y}_s$ separately. Approximating $\{v^*(s)\}_{s \in \mathcal{S}}$ is difficult because (1) zero-sum Markov games are nonconvex-nonconcave problems, and then, solving $\{v^*(s)\}$ is almost as hard as our goal which is finding a NE; (2) the players only know marginal reward and marginal transition kernel, so in each player's perspective, the marginal MDP observed is dynamic. More specifically, the errors in approximations for $\{v^*(s)\}$ will cause inaccuracy in approximate Q-functions and policy gradients. The inaccuracy in policy gradients will make $z^t$ far from the NE set and then induce errors in approximations for $\{v^*(s)\}$. This will easily generate a vicious circle and make the errors blow up.

• **Averaging OGDA and OGDA employ different ways to approximate $\{v^*(s)\}_{s \in \mathcal{S}}$ in each iteration.** OGDA directly uses $V^{x^t, y^t}(s)$ to approximate $v^*(s)$. Thus, the min-player uses $Q_s^t y_s^t = Q_s[V^{x^t, y^t}]y_s^t$ as approximate Q-functions to do optimistic gradient descent, while the max-player uses $(Q_s^t)^\top x_s^t = Q_s[V^{x^t, y^t}]^\top x_s^t$ as approximate Q-functions to do optimistic gradient ascent. On the other hand, in Averaging OGDA, the min-player and max-player use $\underline{V}^t(s)$ and $\overline{V}^t(s)$ respectively to approximate $\{v^*(s)\}$. Thus, the min-player uses $\underline{q}_s^t = Q_s[\underline{V}^t(s)]y_s^t$ as approximate Q-functions to do optimistic gradient descent, while the max-player uses $\overline{q}_s^t = Q_s[\overline{V}^t(s)]^\top x_s^t$ as approximate Q-functions to do optimistic gradient ascent. $\underline{V}^t(s)$ and $\overline{V}^t(s)$ are computed through the averaging technique. Take the min-player as an example, where $\underline{V}^t(s) = \min_{a \in \mathcal{A}} \sum_{j=T_1}^{t-1} \alpha_{t-T_1}^{j-T_1+1} \underline{q}_s^j(a)$ is a step of value iteration on the average of past Q-function approximations. We will elaborate on the intuition of $\underline{V}^t(s)$ below, but before that, we first show why the averaging step is needed to achieve global convergence (equivalently, the disadvantage of using $V^{x^t, y^t}(s)$ to approximate $v^*(s)$).

• **Technical challenge in the analysis of OGDA.** The main difficulty in the analysis of OGDA is the nonconvex-nonconcave essence of zero-sum Markov games. As discussed in Section 5.1

of Daskalakis et al. (2020), the failure of the Minty Variational Inequality (MVI) property in zero-sum Markov games poses challenges for the last-iterate convergence of extragradient methods/optimistic gradient methods. More specifically, given the objective function $f(z)$ with $z = (x, y)$ and $F(z) = (\nabla_x f(z), -\nabla_y f(z))$, the MVI property means that there exists a point $z^* = (x^*, y^*)$ such that $\langle F(z), z - z^* \rangle \geq 0$ for any $z$. Proposition 2 of Daskalakis et al. (2020) proves that when setting $f(x, y) = V^{x,y}(s)$ for some state $s \in \mathcal{S}$, the MVI property can fail in arbitrarily small neighborhoods of the NE set.

More specifically, for the OGDA method (8), it may happen that there exists some $s \in \mathcal{S}$ such that

$$\left\langle \boldsymbol{x}_s^{t+1} - \widetilde{\boldsymbol{x}}_s^{t*}, \boldsymbol{Q}_s^{t+1} \boldsymbol{y}_s^{t+1} \right\rangle + \left\langle \widetilde{\boldsymbol{y}}_s^{t*} - \boldsymbol{y}_s^{t+1}, \left( \boldsymbol{Q}_s^{t+1} \right)^\top \boldsymbol{x}_s^{t+1} \right\rangle < 0, \tag{114}$$

where $\widetilde{\boldsymbol{x}}_s^{t*} = \mathcal{P}_{\mathcal{X}^*}(\widetilde{\boldsymbol{x}}^t)$, $\widetilde{\boldsymbol{y}}_s^{t*} = \mathcal{P}_{\mathcal{Y}^*}(\widetilde{\boldsymbol{y}}^t)$ are projections. We also denote $\widetilde{\boldsymbol{z}}_s^{t*} = \mathcal{P}_{\mathcal{Z}^*}(\widetilde{\boldsymbol{z}}^t)$. The troublesome case (114) implies that going in the directions of policy gradients may deviate from rather than get close to the NE set. A naive bound to evaluate how worse the policy gradients can be is: $\left\langle \boldsymbol{x}_s^{t+1} - \widetilde{\boldsymbol{x}}_s^{t*}, \boldsymbol{Q}_s^{t+1} \boldsymbol{y}_s^{t+1} \right\rangle + \left\langle \widetilde{\boldsymbol{y}}_s^{t*} - \boldsymbol{y}_s^{t+1}, \left( \boldsymbol{Q}_s^{t+1} \right)^\top \boldsymbol{x}_s^{t+1} \right\rangle \geq -2 \max_{(a,b) \in \mathcal{A} \times \mathcal{B}} \left| \boldsymbol{Q}_s^{t+1}(a,b) - \boldsymbol{Q}_s^*(a,b) \right|$.

The troublesome error term in the naive bound is of order $2 \max_{(a,b) \in \mathcal{A} \times \mathcal{B}} \left| \boldsymbol{Q}_s^{t+1}(a,b) - \boldsymbol{Q}_s^*(a,b) \right| = O(\text{dist}(\boldsymbol{z}^t, \mathcal{Z}^*))$. On the other hand, as we show in Appendix C.2, projected optimistic gradient descent/ascent can only provide progress of order $O(\text{dist}^2(\boldsymbol{z}^t, \mathcal{Z}^*))$. When $\boldsymbol{z}^t$ is close to the NE set, the error term can be much larger than the progress, i.e., $2 \max_{(a,b) \in \mathcal{A} \times \mathcal{B}} \left| \boldsymbol{Q}_s^{t+1}(a,b) - \boldsymbol{Q}_s^*(a,b) \right| << O(\text{dist}^2(\boldsymbol{z}^t, \mathcal{Z}^*))$. This prevents us from even showing the local convergence of OGDA.

● **Intuition of Averaging OGDA.** Averaging OGDA tackles the problem (114) by using $\underline{V}^t$, $\overline{V}^t$ instead of $V^{x^t, y^t}$ to approximate $v^*$. The corresponding policy gradients $Q_s[\underline{V}^t] y_s^t, Q_s[\underline{V}^t]^\top x_s^t$ are good directions in the sense that we can provide a good lower bound for $(x_s^t - x_s^{t*})^\top Q_s[\underline{V}^t] y_s^t + (y_s^{t*} - y_s^t)^\top Q_s[\overline{V}^t]^\top x_s^t$. More specifically, by Lemma 19 and Fact 2, we have

$$(x_s^t - x_s^{t*})^\top Q_s[\underline{V}^t] y_s^t + (y_s^{t*} - y_s^t)^\top Q_s[\overline{V}^t]^\top x_s^t \geq -\|\overline{V}^t - \underline{V}^t\|_\infty.$$

As in Appendix D.1, the term $\|\overline{V}^t - \underline{V}^t\|_\infty$ is relatively easy to control. Thus, $Q_s[\underline{V}^t] y_s^t, Q_s[\underline{V}^t]^\top x_s^t$ are "good" directions.

● **More discussions on OGDA.** Although Averaging OGDA use the averaging technique to achieve global convergence, the averaging technique will make the errors from past iterations fail to decrease exponentially fast. This prevents Averaging OGDA from achieving linear convergence. To get local linear convergence, OGDA is a natural candidate because $V^{x^t, y^t}$ depends only on the most recent policies, thus, it avoids errors from the past steps. The problem (114) mentioned above necessitates novel analysis for OGDA. Our strategy for the local linear convergence of OGDA starts from considering a weighted sum of $\left\langle \boldsymbol{x}_s^{t+1} - \widetilde{\boldsymbol{x}}_s^{t*}, \boldsymbol{Q}_s^{t+1} \boldsymbol{y}_s^{t+1} \right\rangle$ and $\left\langle \widetilde{\boldsymbol{y}}_s^{t*} - \boldsymbol{y}_s^{t+1}, \left( \boldsymbol{Q}_s^{t+1} \right)^\top \boldsymbol{x}_s^{t+1} \right\rangle$.

Let $\boldsymbol{\rho}_0$ denote the uniform distribution on $\mathcal{S}$. As $(\widetilde{\boldsymbol{x}}^{t*}, \widetilde{\boldsymbol{y}}^{t*})$ attains a Nash equilibrium,

$$V^{\boldsymbol{x}^{t+1}, \widetilde{\boldsymbol{y}}^{t*}}(\boldsymbol{\rho}_0) - V^{\widetilde{\boldsymbol{x}}^{t*}, \boldsymbol{y}^{t+1}}(\boldsymbol{\rho}_0) \geq 0.$$

Thus, $0 \leq V^{\boldsymbol{x}^{t+1}, \widetilde{\boldsymbol{y}}^{t*}}(\boldsymbol{\rho}_0) - V^{\widetilde{\boldsymbol{x}}^{t*}, \boldsymbol{y}^{t+1}}(\boldsymbol{\rho}_0) = V^{\boldsymbol{x}^{t+1}, \widetilde{\boldsymbol{y}}^{t*}}(\boldsymbol{\rho}_0) - V^{\boldsymbol{x}^{t+1}, \boldsymbol{y}^{t+1}}(\boldsymbol{\rho}_0) + V^{\boldsymbol{x}^{t+1}, \boldsymbol{y}^{t+1}}(\boldsymbol{\rho}_0) - V^{\widetilde{\boldsymbol{x}}^{t*}, \boldsymbol{y}^{t+1}}(\boldsymbol{\rho}_0)$. Then, by applying performance difference lemma (Lemma 20), we have a variant of the MVI property with time-varying coefficients which is as in (16), which can be regarded as a variant of the MVI property with time-varying coefficients.

In order to utilize (16) to get local linear convergence, we still need to tackle the following two problems:

- whether we can find a neighborhood of the NE set such that the time-varying coefficients $\boldsymbol{d}_x^t(s)$, $\boldsymbol{d}_y^t(s)$ in (16) are "stable"?

- if the time-varying coefficients $\boldsymbol{d}_x^t(s)$, $\boldsymbol{d}_y^t(s)$ in (16) can be "stable" in a small neighborhood of the NE set, will the difference between $\boldsymbol{Q}_s^t$ and $\boldsymbol{Q}_s^*$ prevent the local linear convergence?

To address the above questions, we mainly use the two geometric observations in Section 5. Observation I guarantees the progress of projected gradient descent/ascent is substantial. This means that the difference between $Q_s^t$ and $Q_s^*$ will not be troublesome in deriving the local linear convergence. Observation II implies the stability of state visitation distribution. Thus, the time-varying coefficients $d_x^t(s)$, $d_y^t(s)$ will be "stable" when $z^t, \widetilde{z}^t$ are approaching the NE set. In other words, we can find a problem-dependent neighborhood where the time-varying coefficients $d_x^t(s)$, $d_y^t(s)$ will possess some "stability". We remark that since $d_x^t(s)$, $d_y^t(s)$ can vary rapidly when $z^t$ is far from the NE set, our analysis for OGDA only hold in a small neighborhood of the NE set. Whether OGDA has global convergence is still an open problem.

