# OpenReview forum: "Can We Find Nash Equilibria at a Linear Rate in Markov Games?"
_ICLR.cc/2023/Conference — ICLR 2023 notable top 25%_

### Official Review · Reviewer_5eyy · 2022-10-23

**Confidence:** 4
**Clarity, Quality, Novelty And Reproducibility:** Both the algorithm and analysis seem …
**Correctness:** 3
**Technical Novelty And Significance:** 3
**Empirical Novelty And Significance:** Not applicable
**Recommendation:** 8

**Strength And Weaknesses:**

Pros:
- The presentation is very clear, and the writing is also flawless.
- The proposed framework is general, and the convergence result is important.

Cons:
- No experiments.

**Summary Of The Paper:**

This paper studies policy optimization for Markov Games to find the Nash equilibrium. The authors parameterize the policy by directly considering the probability for each state and action. The key contribution of this work is to propose a slow-fast framework with a series of conditions, and prove that any pair of slow and fast optimization algorithms that satisfy the above conditions, can enjoy a linear convergence rate to the Nash equilibrium.

**Summary Of The Review:**

In general, this paper did a good job of showing a global linear convergence to the Nash equilibrium for Markov Games. The idea to use hidden phases to analyze the convergence of slow and fast algorithms is quite interesting, and I believe this can be extended to other policy optimization problems considered in reinforcement learning. The only shortcoming of this paper is the lack of experiments, and I suggest the authors do some at least on some simulated datasets, especially to see how the large constant ($(1-c\eta^2)^{-poly}$ in (9)) affect the final convergence behavior.

---

> ### Author Response · Authors · 2022-11-17
> **Response to Reviewer 5eyy (Part 2/2)**
>
> (continuing with our numerical results)
>
> To see the switches between Averaging OGDA and OGDA more clearly, we do another group of experiments by generalizing the switch pattern slightly. Notice that for arbitrary $v > u > 1$, if the $k$-th call to Averaging OGDA and OGDA have $\lceil u^k \rceil$ and $\lceil v^k \rceil$ iterations respectively, we can prove the linear convergence in a similar way to Theorem 1 directly. This is because after $k\geq O(\frac{1}{c_0\eta^2}\log_{v/u}(D_0\Gamma_0))$, the increase of Averaging OGDA can be "omitted" by the decrease of OGDA (see the relation between the segments $\overline{AB}$ and $\overline{BC}$ in our theoretical bounds illustrated in Figure 1). And $u^k$ increases exponentially fast so that the length of Hidden Phase I is polynomial in problem parameters. Specifically, in Figure 3, Figure 4 and Figure 5, we set $u = 2$, $v = 2.1$, which means that the $k$-th call to Averaging OGDA and OGDA have $2^k$ and $\lceil 2.1^k \rceil$ iterations respectively. The average trajectory and standard deviation are illustrated in Figure 3. And the trajectories of the 10 trials are illustrated in Figure 4 and Figure 5. We remark that since Averaging OGDA is an averaging style method, it is quite possible that the average policy $\widehat{z}^{[T_1:T_2]} \neq z^{T_2}$. Meanwhile, we use $\widehat{z}^{[T_1:T_2]}$ as the initial policy of the next call to OGDA. This naturally causes discontinuity in the trajectories when switching from Averaging OGDA to OGDA (see Remark 4 in the revision).
> We draw a vertical dotted line at the end of the 12-th call to OGDA ($t = 22237$). Then we can see all the trajectories exhibit linear convergence on the RHS of the dotted line (after $t > 22237$). In the trajectories, Averaging OGDA can cause increase but its increase is still negligible compared with the decrease from lots of steps of OGDA. This again aligns with our theoretical bounds for Hidden Phase II (see the relation between the segments $\overline{AB}$ and $\overline{BC}$ in Figure 1).

---

> ### Author Response · Authors · 2022-11-17
> **Response to Reviewer 5eyy (Part 1/2)**
>
> Thank you for your valuable feedback! Here are our responses.
> >  The only shortcoming of this paper is the lack of experiments, and I suggest the authors do some at least on some simulated datasets, especially to see how the large constant ($(1 - c\eta^2)^{t - poly(S, A, B, 1/(1-\gamma), 1/c_+)}$ in (9)) affect the final convergence behavior.
>
> We have added numerical experiments in Appendix I of the revision.
>
> We briefly summarize our numerical experiments here, and more details can be found in Appendix I of the revision.
>
> ### **Numerical experiments**
>
> - **Markov Game Model** We generate a sequence of zero-sum Marov games randomly and independently and run our algorithm on each game. In each game, there are 10 states, the players have $A = B = 10$ actions. The discount factor $\gamma = 0.99$. The rewards are drawn uniformly at random from $[0, 1]$. To generate the transition probability, for each tuple $(s,a,b)$, we first select a random subset $M_{s,a,b}$ of $\mathcal{S}$ and sample a value in $[0, 1]$ randomly for each element in $M_{s,a,b}$. Then we normalize these values to get the transition kernel. The initial policies are also chosen randomly.
>
> - **Algorithm Implementation** In all the experiments (10 random games, we test  Homotopy-PO and a variant of Homotopy-PO with a generalization in the switching pattern on each game, then there are 20 trials in total), we set the stepsize $\eta = 0.1$ for OGDA and $\eta' = 0.1$ for Averaging OGDA. We find our algorithm has linear convergence in all the experiments with these stepsizes.
>
> - **Performance Metric** By Eq(113) in the revision, the linear convergence of $dist(z^t, \mathcal{Z}^*)$ is equivalent to the linear convergence of the Nash gap $\max_{s\in \mathcal{S}} V^{x^t, \dagger}(s) - V^{\dagger, y^t}(s)$. Thus we use the Nash gap to measure the closeness of the policy pair $z^t$ to the NE set. More specifically, the $y$-axes of  all the figures are the logarithm of the Nash gap and all $x$-axes represent iteration numbers.
> $$
>     \text{y-axis}: \log(\max_{s\in \mathcal{S}} V^{x^t, \dagger}(s) - V^{\dagger, y^t}(s)), \quad \text{x-axis: iteration number}
> $$
>
> **Results** We generate 10 random games in the way described above. We test our instantiation of Homotopy-PO on each of the game with the stepsize stated above. Our algorithm can exhibit linear convergence in each of the trials. The average trajectory and standard deviation of these trials are illustrated in Figure 2 of the revision, where after the 7-th call to OGDA ($t> 22098$), the average trajectory starts to show linear convergence and the Nash gap is less than $10^{-5}$ after about $2\times 10^5$ iterations.
>
> In Homotopy-PO, $2^k$ iterations of Averaging OGDA is followed by $4^k$ iterations of OGDA. In the first $2\times 10^5$ iterations, Averaging OGDA is only run for 1022 iterations which makes it almost invisible in the trajectory. We magnify the 9-th call to Averaging OGDA as a subfigure in Figure 2. It is observed that Averaging OGDA can cause the Nash gap to increase. However, the 9-th call to Averaging OGDA has $2^9$ iterations, while the 8-th call to OGDA has $4^8$ iterations. $4^8/2^9 = 128$. Thus the increase from Averaging OGDA is negligible compared with the decrease from lots of steps of OGDA. This aligns with with our theoretical bounds for Hidden Phase II (see the relation between the segments $\overline{AB}$ and $\overline{BC}$ in Figure 1).

---

### Official Review · Reviewer_p6su · 2022-10-24

**Confidence:** 3
**Correctness:** 4
**Technical Novelty And Significance:** 4
**Empirical Novelty And Significance:** Not applicable
**Recommendation:** 10

**Clarity, Quality, Novelty And Reproducibility:**

As discussed above, this paper is well-written and contains high novelty. Meanwhile, there are also several questions as the following.

### Questions
- What is the intuition of using $\underline{\mathbf{q}}_s^{t}$ (and correspondingly $\overline{\mathbf{q}}_s^{t}$) for update in Averaging OGDA? More specifically, what is the intuitive difference between the update rules of Averaging OGDA and OGDA?
- Is that possible to extend this idea to a learning algorithm (in which we cannot compute $\mathbf{Q}$ exactly) with linear convergence?
- Why the linear convergence of $\mathsf{LOCAL}$-$\mathsf{FAST}$ requires the initialization $\hat{\mathbf{z}}$ to be close enough to $\mathcal{Z}^*$? Is this requirement fundamental?
- What is the rough magnitude of $c_+$ in terms other problem parameters?
- Is that possible to translate the convergence results in this paper to the version in terms of Nash gap (i.e. $\max_{s\in\mathcal{S}}\left(V^{\mathbf{x}^t, \dagger}(s)-V^{\dagger, \mathbf{y}^t}(s)\right)$)?

### Suggestions on Writing
- $\eta'$->$\eta$ in equation (2).
- $k\geq k^*$->$k\geq k_1^*$ after **Hidden Phase II** at the bottom of page 5.
- It may be better to also explicitly state the rationality of $\mathsf{Homotopy}$-$\mathsf{PO}$ as a theorem.

**Strength And Weaknesses:**

### Strengths
This paper proposes the first decentralized algorithm with global linear convergence rate, which is considered to be significant. Meanwhile, the construction of the meta-algorithm is based on a very clear and novel intuition, which has potential wider application since it only requires the base algorithms to satisfy certain properties. Furthermore, proving that OGDA has local linear convergence itself is also considered to be significant and novel.

### Weaknesses
Based on the decentralized implementation of the two base algorithms OGDA and Averaging OGDA, it seems translating the current optimization algorithm to a learning algorithm contains some inherent difficulty. In particular, the initialization needs to solve the marginal MDP completely, which looks hard to approximate during the learning process. Is there any approach to relax it so that we can translate it to a learning algorithm easily?

Nevertheless, from my perspective, it is okay to leave this as future work for now.

---

**Post-rebuttal update:** concerns and questions have been well-addressed.

**Summary Of The Paper:**

This paper investigates decentralized policy optimization algorithm for two-player zero-sum discounted Markov games. In particular, it proposes a decentralized meta-algorithm called $\mathsf{Homotopy}$-$\mathsf{PO}$ that has global linear convergence rate if the its two base algorithms have the desired properties. Then, it proves that Averaging OGDA and OGDA can serve as the two base algorithms for $\mathsf{Homotopy}$-$\mathsf{PO}$, resulting a concrete decentralized algorithm with linear convergence rate.

**Summary Of The Review:**

This paper proposes the first decentralized algorithm with linear convergence rate in discounted two-player zero-sum Markov games, which is considered to be novel and significant. Meanwhile, it also contains some novel techniques may be of independent interest. Although its practicability looks limited now, it is okay to leave it as future work.

---

> ### Author Response · Authors · 2022-11-17
> **Response to Reviewer p6su (Part 5/5)**
>
> > Why the linear convergence of Local-Fast requires the initialization $\widehat{z}$ to be close enough to $\mathcal{Z}^*$? Is this requirement fundamental?
>
> The initialization of Local-Fast does not need to be within a small neighborhood of the NE set. We only require that Local-Fast will converge linearly in a small neighbor of the NE set, but do not require that the initialization is in this neighborhood when Local-Fast is called by Homotopy-PO. It is still possible that when Homotopy-PO calls Local-Fast, the current iterate is outside of such a benign neighborhood and we do not require the convergence of Local-Fast when its initialization is outside this benign neighborhood.
>
>   However, we can prove that in Hidden Phase II, the initialization of Local-Fast always stays in the benign neighborhood. More specifically, by the geometric boundedness of Global-Slow, when $2^k \geq O(\frac{1}{c_0\eta^2}\log(D_0 \Gamma_0))$, the increase of Global-Slow can be offset by the fast decrease of Local-Fast (please the relation between segments $\overline{BC}$ and $\overline{CD}$ in Figure 1). Then we can prove by induction that in Hidden Phase II, $z^t$ always stays in the small neighborhood where Local-Fast has linear convergence. Thus, even if Local-Fast has cycles or does not converge outside this neighborhood, Homotopy-PO can still have global linear convergence.
>
> > What is the rough magnitude of $c_+$ in terms of other problem parameters?
>
> The constant $c_+$ only depends on the matrices $\\{Q^*_s\\}\_{s\in \mathcal{S}}$.
>
> More specifically, Lemma 4 of [R3] proves that for any matrix $G$, there exists $\varphi(G) > 0$ such that for any $x\in \Delta_{\mathcal{A}}$, $y\in \Delta_{\mathcal{B}}$, $z = (x,y)$,
> $$
>     \max_{y'\in \Delta_{\mathcal{B}}} x^{\top}Gy' - \min_{x' \in \Delta_{\mathcal{A}}}(x')^{\top}Gy \geq \varphi(G)\cdot dist(z, \mathcal{Z}^*(G)),
> $$
> where $\mathcal{Z}^*(G)$ is the NE set of the matrix game $\min_x\max_y x^{\top}Gy$.
>
> Then as in Corollary 2 of the revision, $c_+$ is defined as
> $$
>   c_+ = \min_{s\in \mathcal{S}} \varphi(Q^*_s).
> $$
>
> The constant $\varphi(G)$ or $c_+$ usually occurs as an independent parameter in the complexities of the algorithms for matrix games and Markov games, while how $\varphi(G)$ scales with the size of $G$ or how $c_+$ scales with $S, A, B$ is still unclear till now and is worth future study. We take [R1, R3, R4] as examples.
>
> [R3] is the first algorithm that finds a NE at a linear rate for zero-sum matrix games. Its linear rate is $(1 - O(\frac{\varphi(G)}{\|G\|}))^t$.
>
> [R1] proves that the OGDA method therein can find a NE for matrix games at a linear rate of $(1 - O(\frac{\varphi(G)^2}{\|G\|^2}))^t$.
>
> [R4] studies zero-sum Markov games and gets a sub-linear algorithm whose convergence rate is $O(\frac{S}{\eta^2c_+^2(1-\gamma)^2\sqrt{T}})$.
>
> > Is that possible to translate the convergence results in this paper to the version in terms of Nash gap (i.e. $\max_{s\in \mathcal{S}} V^{x^t, \dagger}(s) - V^{\dagger, y^t}(s)$)?
>
>
> Our results for $dist^2(z^t, \mathcal{Z}^*)$ can be translated to the Nash gap directly. By Eq(113) in the revision, we have
>     $$
>         \frac{c_+}{\sqrt{S}}\cdot dist(z^t, \mathcal{Z}^*) \leq \max_{s\in \mathcal{S}} V^{x^t, \dagger}(s) - V^{\dagger, y^t}(s) \leq \frac{\max\{\sqrt{2A}, \sqrt{2B}\}}{(1-\gamma)^2}\cdot dist(z^t, \mathcal{Z}^*).
>     $$
>   Thus, the linear convergence of $dist(z^t, \mathcal{Z}^*)$ is equivalent to the linear convergence of the Nash gap up to problem-dependent constants.
>
> > $\eta \rightarrow \eta'$ in equation (2).
>
> > $k \geq k^* \rightarrow k \geq k^*_1$ after Hidden Phase II at the bottom of page 5.
>
> > It may be better to also explicitly state the rationality of Homotopy-PO as a theorem.
> - Thank you for your careful review! We have fixed all the problems in the revision. We state and prove the rationality in Theorem 6 of the revision.
>
>
>
> **References**
>
> [R1] Wei, C. Y., Lee, C. W., Zhang, M., & Luo, H. (2020). Linear Last-iterate Convergence in Constrained Saddle-point Optimization. In International Conference on Learning Representations.
>
> [R2] Daskalakis, C., Foster, D. J., & Golowich, N. (2020). Independent policy gradient methods for competitive reinforcement learning. Advances in neural information processing systems, 33, 5527-5540.
>
> [R3] Gilpin, A., Pena, J., & Sandholm, T. W. (2012). First-Order Algorithm with $O (\ln (1/\epsilon))$ Convergence for $\epsilon$-Equilibrium in Two-Person Zero-Sum Games. Mathematical Programming. 133:279-298.
>
> [R4] Wei, C. Y., Lee, C. W., Zhang, M., & Luo, H. (2021). Last-iterate convergence of decentralized optimistic gradient descent/ascent in infinite-horizon competitive markov games. In Conference on learning theory (pp. 4259-4299). PMLR.

---

> > ### Comment · Reviewer_p6su · 2022-11-28
> > **Response**
> >
> > Thank you very much for your detailed explanations! My concerns have all been well-addressed and I have also increased my score!

---

> > > ### Author Response · Authors · 2022-11-29
> > > **Thank you for the update!**
> > >
> > > Thank you for the update!

---

> ### Author Response · Authors · 2022-11-17
> **Response to Reviewer p6su (Part 4/5)**
>
> (continuing with the intuition of Averaging OGDA and OGDA)
>
>    (3) **Technical challenge in the analysis of global convergence of OGDA.**
>    OGDA uses $V^{x^t, y^t}$ to approximate $v^*$ directly. Then, OGDA uses $Q_s[V^{x, y}]y_s$, $(Q_s[V^{x, y}])^{\top}x_s$ as gradient policies, which may not be descent directions. More specifically, there exists $z_s = (x_s,y_s)\in \mathcal{Z}_s$ such that for any $z^*_s = (x^*_s, y^*_s) \in \mathcal{Z}^*_s$,
>    \begin{align}
>         ( x_s - x^*_s )^{\top} Q_s [ V^{x, y} ]  y_s + ( y^*_s - y_s )^{\top} ( Q_s[ V^{x, y} ] )^{\top} x_s < 0.
>    \end{align}
>    This is identified as a critical challenge in the analysis of OGDA or extragradient methods (see Section 5 of [R2]).
>
>    (4) **Intuition of Averaging OGDA.**
>    Averaging OGDA tackles the problem mentioned in subsection(3) above by using $\underline{V}^t$, $\overline{V}^t$ instead of $V^{x^t, y^t}$ to approximate $v^*$. The corresponding policy gradients $Q_s[\underline{V}^t]y^t_s$, $Q_s[\underline{V}^t]^{\top}x^t_s$ are good directions in the sense that we can provide a good lower bound for $(x_s^t - x^{t*}_s)^{\top}Q_s[\underline{V}^t]y^t_s + (y^{t*}_s - y^t_s)^{\top}Q_s[\overline{V}^t]^{\top}x^t_s$. More specifically,
>
>    $$
>     (x_s^t - x^{t*}_s)^{\top} Q_s [ \underline{V}^t ] y^t_s + ( y^{t*}_s - y^t_s )^{\top} Q_s [ \overline{V}^t ]^{\top} x^t_s \geq - ||\overline{V}^t - \underline{V}^t||\_{\infty}.
> $$
>
> We will show later that the term $|| \overline{V}^t - \underline{V}^t ||\_{\infty}$ is easy to control. The above inequality is derived from the following facts: (i) $\underline{V}^t(s)$ and $\overline{V}^t(s)$ are kept as lower and upper bounds for $v^*(s)$. Then, in entry-wise sense, $Q_s[\underline{V}^t] \leq Q_s[v^*]$ and $Q_s[\overline{V}^t] \leq Q_s[v^*]$. (ii) By Shapley's theorem (Lemma 19),
>    $$ {y^{t*}_s}^{\top}(Q^*_s)^{\top}x_s^t - {x^{t*}_s}^{\top}Q^*_sy_s^t \geq 0.
>    $$
>
>    The term $|| \overline{V}^t - \underline{V}^t ||\_{\infty}$ is relatively easy to control. More specifically, in Lemma 12, we use the contraction property of the operator $Q_s[\cdot]$ and the fact $\sum_{t=j}^{\infty}\alpha^j_t = 1 + \frac{1}{H}$ of the classical stepsize choice $\alpha_t = (H+1)/(H+t)$ and $\alpha^j_t = \alpha_j\prod_{k=j+1}^t(1-\alpha_k)$. By letting $H = O(1/(1 - \gamma))$, we can have $\gamma(1+\frac{1}{H}) < 1$. Intuitively, this indicates that the error accumulation is slower than the decrease caused by the contraction property of $Q_s[\cdot]$. Then we can bound the term $||\overline{V}^t - \underline{V}^t||\_{\infty}$ by weighted sum of "regrets". Then we use standard analysis in normal form games to bound the "regrets".
>
>    (5) **More discussions on the intuition of OGDA.**
>    Although Averaging OGDA use the averaging technique to achieve global convergence, the averaging technique will make the errors from past iterations fail to decrease exponentially fast. This prevents Averaging OGDA from achieving linear convergence. To get local linear convergence, OGDA is a natural candidate because $V^{x^t, y^t}$ depends only on the most recent policies so it avoids errors from the past steps. As we have discussed in subsection(3) above, using $V^{x^t, y^t}$ to approximate $v^*$ can make the corresponding policy gradients not good directions. This will let the error bounding become very hard. We propose a novel analysis for OGDA to tackle this. We avoid the trouble discussed in subsection(3) above by Eq(16) of the manuscript. But a new problem occurs which is the time-varying and unstable coefficients $d^t_x$, $d^t_y$. We deal with instability of $d^t_x$, $d^t_y$ using Observation II in Section 5 and Appendix B, Lemma 10 of the manuscript. Intuitively, we show that when $z^t$ is approaching the NE set, the time-varying coefficients $d^t_x$, $d^t_y$ will become more and more stable. We also prove that gradient descent/ascent can give progress of order $O(dist^2(z^t, \mathcal{Z}^*))$ by Observation I (Lemma 7). Then intuitively when the coefficients $d^t_x$, $d^t_y$ are "stable" enough and projected gradient descent/ascent can provide progress proportional to $O(dist^2(z^t, \mathcal{Z}^*))$, the linear convergence begins.
>
>
>
>
> > Is that possible to extend this idea to a learning algorithm (in which we cannot compute Q  exactly) with linear convergence?
>
> (1) Consider sample complexity. For MDP which is a special case of zero-sum Markov games when the reward and transition kernel is invariant with one player's policy, we need at least $\widetilde{O}(1/\epsilon^2)$ samples to get $\epsilon$-optimal policy. So we may not have linear convergence against the number of samples.
>
>    (2) If there is a simulator (generative model), then it is possible to get linear convergence against the iteration number. Please see the section "Translation to learning algorithms" in our response.

---

> ### Author Response · Authors · 2022-11-17
> **Response to Reviewer p6su (Part 3/5)**
>
> > What is the intuition of using $\underline{q}^t_s$ (and correspondingly $\overline{q}^T_s$) for update in Averaging OGDA? More specifically, what is the intuitive difference between the update rules of Averaging OGDA and OGDA?
>
> - Thank you for this critical comment. We will explain the intuition of Averaging OGDA and OGDA step-by-step in the following subsections (1)-(5).
>
>    (1) **In Markov games, the main challenge of finding an NE is to estimate the minimax game value** $v^*(s)$.
>
>    In matrix games $\min_x\max_y x^{\top}Gy$, there is no need to approximate the minimax game value because the players can use $Gy^t$ and $G^{\top}x^t$ as gradients to do optimistic gradient descent/ascent and get linear convergence, see [R1].
>
>
>    In zero-sum Markov games, the players have to approximate $\\{v^*(s)\\}\_{s\in \mathcal{S}}$ to compute policy gradients.
>
>    Imagine that if the players know the exact minimax game values $\\{v^*(s)\\}\_{s\in \mathcal{S}}$ in advance, then they can compute the ideal policy gradient $Q_s[v^*]y^t_s$ and $Q_s[v^*]^{\top}x^t_s$ (because for any $v\in \mathbb{R}^S$, the min-player only needs marginal reward and marginal transition kernel to compute $Q_s[v]y^t_s$). Then the problem will reduce to solving the following $S$ matrix games  separately
>   $$
>      \\{\min\_{x_s}\max\_{y_s} x^{\top}_sQ_s[v^*]y_s\\}\_{s\in \mathcal{S}}
>   $$
>
>   However, the challenge in solving zero-sum Markov games is that we do not know $\\{v^*(s)\\}\_{s\in \mathcal{S}}$.
>
>
>   Approximating $\\{v^*(s)\\}\_{s\in \mathcal{S}}$ is difficult because (1) zero-sum Markov games are nonconvex-nonconcave problems, and then, solving $\{v^*(s)\}$ is almost as hard as our goal which is finding a NE; (2) the players only know marginal reward and marginal transition kernel, so in each player's perspective, the marginal MDP observed is dynamic. More specifically, the errors in approximations for $\{v^*(s)\}$ will cause inaccuracy in approximate Q-functions and policy gradients. The inaccuracy in policy gradients will make $z^t$ far from the NE set and then induce errors in approximations for $\\{v^*(s)\\}$. This will easily generate a vicious circle and make the errors blow up. The problem is harder considering the players only have marginal information.
>
>
>    (2) **Averaging OGDA and OGDA employ different ways to approximate** $\\{v^*(s)\\}\_{s\in \mathcal{S}}$ **in each iteration.**
>
>
>    OGDA directly uses $V^{x^t, y^t}(s)$ to approximate $v^*(s)$. So the min-player uses $Q^t_sy^t_s = Q_s[V^{x^t, y^t}]y^t_s$ as approximate Q-functions to do optimistic gradient descent, while the max-player uses $(Q^t_s)^{\top}x^t_s = Q_s[V^{x^t, y^t}]^{\top}x^t_s$ as approximate Q-functions to do optimistic gradient ascent.
>
>
>    On the other hand, in Averaging OGDA, the min-player and max-player use $\underline{V}^t(s)$ and $\overline{V}^t(s)$ respectively to approximate $\{v^*(s)\}$. So the min-player uses $\underline{q}^t_s = Q_s[\underline{V}^t(s)]y^t_s$ as approximate Q-functions to do optimistic gradient descent, while the max-player uses $\overline{q}^t_s = Q_s[\overline{V}^t(s)]^{\top}x^t_s$ as approximate Q-functions to do optimistic gradient ascent. $\underline{V}^t(s)$ and $\overline{V}^t(s)$ are computed through the averaging technique. Take the min-player as an example, where $\underline{V}^t(s) = \min_{a\in \mathcal{A}}\sum_{j=T_1}^{t-1}\alpha_{t-T_1}^{j-T_1+1}\underline{q}^j_s(a)$ is a step of value iteration on the average of past Q-function approximations. We will elaborate on the intuition of $\underline{V}^t(s)$ below. But before that, we first show why the averaging step is needed to achieve global convergence (equivalently, the disadvantage of using $V^{x^t, y^t}(s)$ to approximate $v^*(s)$).

---

> ### Author Response · Authors · 2022-11-17
> **Response to Reviewer p6su (Part 2/5)**
>
> (continuing with the translation to learning algorithms)
>
>   2. **Using an ergodic assumption.** When no simulator is available, we consider translating our algorithm into a learning algorithm under an ergodic assumption. The assumption is that there exists a constant $D_0 > 0$ such that for any policy pair $z = (x, y)$, if the min-player and the max-player play policy $x$ and $y$ respectively in $D_0$ successive iterations $t\in [T_0:T_0+D_0-1]$, then, for any initial state $s_{T_0}$ and state $s\in \mathcal{S}$, there exists a $t'\in [T_0:T_0+D_0 - 1]$ such that $s_{t'} = s$. Briefly, this assumption requires that when the players choose a stationary policy pair for successive $D_0$ iterations, then every state must be visited at least once in these $D_0$ iterations. Under this assumption, our strategy is to regard successive $D_0\times N_k$ iterations as a virtual iteration $k$. In this way, we divide $[1:T]$ into
> $$
> [1:T] = [T_1:T_2]\cup [T_3:T_4] \cup \cdots \cup [T_{2k-1}:T_{2k}] \cup \cdots
> $$
> where $T_{2k} - T_{2k-1} = D_0\times N_k$.
> Then, in the time interval $[T_{2k-1}:T_{2k}]$, each state $s$ has been visited for at least $N_k$ times. This is similar to the case when we have a simulator and $N_k$ samples are drawn for each state $s$ in iteration $k$. In this way, by applying our algorithm and analysis for the simulator case (Case 1 above), we can show the linear convergence with respect to the virtual iteration number $k$.

---

> ### Author Response · Authors · 2022-11-17
> **Response to Reviewer p6su (Part 1/5)**
>
> Thank you for your valuable feedback! Here are our responses.
>
> > Based on the decentralized implementation of the two base algorithms OGDA and Averaging OGDA, it seems translating the current optimization algorithm to a learning algorithm contains some inherent difficulty. In particular, the initialization needs to solve the marginal MDP completely, which looks hard to approximate during the learning process. Is there any approach to relax it so that we can translate it to a learning algorithm easily?
> Nevertheless, from my perspective, it is okay to leave this as future work for now.
>
> - **Translation to learning algorithms.**
> We think it is possible to translate our algorithm into learning algorithms. Here, we tentatively consider the following two cases and give a rough analysis for each case. More details are left for future study.
>   1. **Assuming access to a simulator (generative model).** If there is a simulator (generative model) and the players can draw lots of samples in one iteration, then it is possible to get linear convergence against the iteration number. More specifically, at iteration $t$, for each $s$, $N_t$ samples $s^{'j}\sim \mathbb{P}(\cdot|s, a^{t,j}, b^{t,j})$ are drawn, where $1\leq j\leq N_t$ and $a^{t,j}\sim x^t_s$, $b^{t,j}\sim y^t_s$. We define a very small variable $\delta_t = O(c_0\eta^2(1-c_0\eta^2)^t)$. Define the truncated simplex $\Delta^t_{\mathcal{A}} = \\{x\in\mathbb{R}^A: x(a) \geq \delta_t, \sum x(a) = 1 \\}$. $\Delta^t_{\mathcal{B}}$ is defined analogously.
>
>        At iteration $t$, we replace the projection operator $\mathcal{P}\_{\Delta_{\mathcal{A}}}(\cdot)$ and $\mathcal{P}\_{\Delta_{\mathcal{B}}}(\cdot)$ with $\mathcal{P}\_{\Delta^t_{\mathcal{A}}}(\cdot)$ and $\mathcal{P}\_{\Delta^t_{\mathcal{B}}}(\cdot)$. This guarantees that each action is taken with probability at least $\delta_t$. Then by Hoeffding's inequality, each action $a$ is taken by the min-player for at least $O(N_t\delta_t)$ times with high probability (w.h.p). Then, the empirical marginal reward $\widehat{r}_x^t$ and marginal transition kernel $\mathbb{\widehat{P}}_x^t$ observed by the min-player satisfy the following relation w.h.p.,
>    $$
>        ||\widehat{r}_x^t - r^t_x||\_{\infty} \leq \widetilde{O}(\sqrt{\frac{1}{N_t\delta_t}}),\quad ||\mathbb{\widehat{P}}_x^t(\cdot|s, a) - \mathbb{P}_x^t(\cdot|s, a)||\_1 \leq \widetilde{O}(\sqrt{\frac{1}{N_t\delta_t}}).
>    $$
>
>    In our responses, $\widetilde{O}(\cdot)$ suppresses logarithmic terms and problem parameters such as $S, A, B, 1/(1-\gamma)$ for simplicity.
>    Thus, we have $||\widehat{V}^{x^t, y^t} - V^{x^t, y^t}||\_{\infty} \leq \widetilde{O}(\sqrt{\frac{1}{N_t\delta_t}})$, $||\widehat{V}^{\dagger, y^t} - V^{\dagger, y^t}||\_{\infty} \leq \widetilde{O}(\sqrt{\frac{1}{N_t\delta_t}})$, $||\widehat{V}^{x^t, \dagger} - V^{x^t, \dagger}||\_{\infty} \leq \widetilde{O}(\sqrt{\frac{1}{N_t\delta_t}})$,
>    $| \widehat{Q}^{x^t, y^t}_s(a,b) - Q^{x^t, y^t}_s(a,b) | \leq \widetilde{O}(\sqrt{\frac{1}{N_t\delta_t}})$
> for any $(s,a,b)$. Here, we use $\widehat{\cdot}$ overhead to indicate the empirical quantities. And the replacement of $\mathcal{P}\_{\Delta_{\mathcal{A}}}(\cdot)$ by $\mathcal{P}\_{\Delta^t_{\mathcal{A}}}(\cdot)$ will add an error term whose $\ell_{\infty}$-norm is at most $\delta_t$. Thus in each iteration $t$, new error terms of order $\widetilde{O}(\sqrt{\frac{1}{N_t\delta_t}} + \delta_t)$ are added. At iteration $t$, let $\bar{x}^{t+1}$ is the ideal variable computed from $\\{x^j\\}\_{j\leq t}$ with exact value of the marginal information $r^t_x$ and $\mathbb{P}^t_x$. Let $x^{t+1}$ be the real variable computed in the learning process. $\bar{y}^{t+1}$ and $y^{t+1}$ are defined similarly. Then, by Cauchy-Schwartz inequality, $dist^2(z^{t+1}, \mathcal{Z}^*) \leq (1 + c_0\eta^2/2)dist^2(\bar{z}^{t+1}, \mathcal{Z}^*) + (1+1/(2c_0\eta^2))\widetilde{O}(\frac{1}{N_t\delta_t}+\delta_t^2)$ let us go into the proof of Theorem 5 in Appendix C.4. Then after adding these error terms, the bounds for the potential functions $\Lambda^t$ defined in Eq.(27) will be
>    $$
>        \Lambda^{t+1} \leq (1 - c_0\eta^2)(1+c_0\eta^2/2)\Lambda^t + \widetilde{O}(\frac{1}{c_0\eta^2}(\frac{1}{N_t\delta_t}+\delta_t^2)).
>    $$
>    Then by setting $N_t = O(c_0^3\eta^6(1 - c_0\eta^2)^{2t})$, we can show by induction that $\Lambda^t \leq O((1 - c_0\eta^2/3)^t)$. This gives the local linear convergence of OGDA when the players can draw lots of samples in one iteration. Similarly, we can also show the global convergence and geometric boundedness of Averaging OGDA. Thus, we think it is possible to get linear convergence against iteration number when there is a generative model and lots of samples can be drawn within one iteration.

---

### Official Review · Reviewer_SKfD · 2022-10-24

**Confidence:** 2
**Clarity, Quality, Novelty And Reproducibility:** High quality, clarity and originality.
**Correctness:** 4
**Technical Novelty And Significance:** 4
**Empirical Novelty And Significance:** Not applicable
**Recommendation:** 8

**Strength And Weaknesses:**

# Strengths
- Finding Nash equilibria in 2-player Markov games is an important problem. Existing methods are restricted to special classes of games (matrix games); have sublinear rates of convergence; or are approximations of Nash equilibria (such as quantal response equilibria).
- The symmetric, decentralized setting, in which the algorithm is symmetric wrt the agents and the agents observe dynamic, local information without being aware of the policy of the opponent.
- The methods proposed are novel; the idea of using a slow, global algorithm used to guide a fast, local algorithm that only works on a neighborhood of a Nash equilibrium is very interesting.
- The fast, local algorithm proposed is of independent interest.

# Weaknesses
- Unclear how practical the algorithms are, as no implementation or empirical validation is provided. It would be nice if the global, linear convergence actually translates into a fast and practical algorithm.

**Summary Of The Paper:**

This paper consider the problem of finding a Nash equilibrium in 2-player, zero-sum Markov games. The contributions are 1) a decentralized, meta-algorithm Homotopy-PO that converges to a Nash equilibrium at a global linear rate. It relies on two subroutines, Global-Slow and Local-Fast. 2) An instantiation of Homotopy-PO with choices for Global-Slow and Local-Fast that achieves the global linear rate of convergence. Further, the algorithms provided are symmetric and rational.

**Summary Of The Review:**

Based on the above comments, the paper is a clear accept.

---

> ### Author Response · Authors · 2022-11-17
> **Response to Reviewer SKfD (Part 2/2)**
>
> (continuing with our numerical results)
>
> To see the switches between Averaging OGDA and OGDA more clearly, we do another group of experiments by generalizing the switch pattern slightly. Notice that for arbitrary $v > u > 1$, if the $k$-th call to Averaging OGDA and OGDA have $\lceil u^k \rceil$ and $\lceil v^k \rceil$ iterations respectively, we can prove the linear convergence in a similar way to Theorem 1 directly. This is because after $k\geq O(\frac{1}{c_0\eta^2}\log_{v/u}(D_0\Gamma_0))$, the increase of Averaging OGDA can be "omitted" by the decrease of OGDA (see the relation between the segments $\overline{AB}$ and $\overline{BC}$ in our theoretical bounds illustrated in Figure 1). And $u^k$ increases exponentially fast so that the length of Hidden Phase I is polynomial in problem parameters. Specifically, in Figure 3, Figure 4 and Figure 5, we set $u = 2$, $v = 2.1$, which means that the $k$-th call to Averaging OGDA and OGDA have $2^k$ and $\lceil 2.1^k \rceil$ iterations respectively. The average trajectory and standard deviation are illustrated in Figure 3. And the trajectories of the 10 trials are illustrated in Figure 4 and Figure 5. We remark that since Averaging OGDA is an averaging style method, it is quite possible that the average policy $\widehat{z}^{[T_1:T_2]} \neq z^{T_2}$. Meanwhile, we use $\widehat{z}^{[T_1:T_2]}$ as the initial policy of the next call to OGDA. This naturally causes discontinuity in the trajectories when switching from Averaging OGDA to OGDA (see Remark 4 in the revision).
> We draw a vertical dotted line at the end of the 12-th call to OGDA ($t = 22237$). Then we can see all the trajectories exhibit linear convergence on the RHS of the dotted line (after $t > 22237$). In the trajectories, Averaging OGDA can cause increase but its increase is still negligible compared with the decrease from lots of steps of OGDA. This again aligns with our theoretical bounds for Hidden Phase II (see the relation between the segments $\overline{AB}$ and $\overline{BC}$ in Figure 1).

---

> ### Author Response · Authors · 2022-11-17
> **Response to Reviewer SKfD (Part 1/2)**
>
> Thank you for your valuable feedback! Here are our responses.
>
> > Unclear how practical the algorithms are, as no implementation or empirical validation is provided. It would be nice if the global, linear convergence actually translates into a fast and practical algorithm.
>
> We have added numerical experiments in Appendix I of the revision.
>
> We briefly summarize our numerical experiments here, and more details can be found in Appendix I of the revision.
>
> ### **Numerical experiments**
>
> - **Markov Game Model** We generate a sequence of zero-sum Marov games randomly and independently and run our algorithm on each game. In each game, there are 10 states, the players have $A = B = 10$ actions. The discount factor $\gamma = 0.99$. The rewards are drawn uniformly at random from $[0, 1]$. To generate the transition probability, for each tuple $(s,a,b)$, we first select a random subset $M_{s,a,b}$ of $\mathcal{S}$ and sample a value in $[0, 1]$ randomly for each element in $M_{s,a,b}$. Then we normalize these values to get the transition kernel. The initial policies are also chosen randomly.
>
> - **Algorithm Implementation** In all the experiments (10 random games, we test  Homotopy-PO and a variant of Homotopy-PO with a generalization in the switching pattern on each game, then there are 20 trials in total), we set the stepsize $\eta = 0.1$ for OGDA and $\eta' = 0.1$ for Averaging OGDA. We find our algorithm has linear convergence in all the experiments with these stepsizes.
>
> - **Performance Metric** By Eq(113) in the revision, the linear convergence of $dist(z^t, \mathcal{Z}^*)$ is equivalent to the linear convergence of the Nash gap $\max_{s\in \mathcal{S}} V^{x^t, \dagger}(s) - V^{\dagger, y^t}(s)$. Thus we use the Nash gap to measure the closeness of the policy pair $z^t$ to the NE set. More specifically, the $y$-axes of  all the figures are the logarithm of the Nash gap and all $x$-axes represent iteration numbers.
> $$
>     \text{y-axis}: \log(\max_{s\in \mathcal{S}} V^{x^t, \dagger}(s) - V^{\dagger, y^t}(s)), \quad \text{x-axis: iteration number}
> $$
>
> **Results** We generate 10 random games in the way described above. We test our instantiation of Homotopy-PO on each of the game with the stepsize stated above. Our algorithm can exhibit linear convergence in each of the trials. The average trajectory and standard deviation of these trials are illustrated in Figure 2 of the revision, where after the 7-th call to OGDA ($t> 22098$), the average trajectory starts to show linear convergence and the Nash gap is less than $10^{-5}$ after about $2\times 10^5$ iterations.
>
> In Homotopy-PO, $2^k$ iterations of Averaging OGDA is followed by $4^k$ iterations of OGDA. In the first $2\times 10^5$ iterations, Averaging OGDA is only run for 1022 iterations which makes it almost invisible in the trajectory. We magnify the 9-th call to Averaging OGDA as a subfigure in Figure 2. It is observed that Averaging OGDA can cause the Nash gap to increase. However, the 9-th call to Averaging OGDA has $2^9$ iterations, while the 8-th call to OGDA has $4^8$ iterations. $4^8/2^9 = 128$. Thus the increase from Averaging OGDA is negligible compared with the decrease from lots of steps of OGDA. This aligns with with our theoretical bounds for Hidden Phase II (see the relation between the segments $\overline{AB}$ and $\overline{BC}$ in Figure 1).

---

### Official Review · Reviewer_Uzv1 · 2022-10-25

**Confidence:** 3
**Correctness:** 4
**Technical Novelty And Significance:** 4
**Empirical Novelty And Significance:** Not applicable
**Recommendation:** 8

**Clarity, Quality, Novelty And Reproducibility:**

The writing is clear and concise and the organization of the paper is very helpful especially for non-expert readers who seek to understand the high level ideas first before going deeper into the analysis. As discussed, the approach and results are to the best of my knowledge novel. No reproducibility concerns since there is no experimental evaluation.

**Strength And Weaknesses:**

Regarding strengths, to the best of my knowledge the proposed algorithms have state of the art convergence rates for the problem of zero sum Markov games. At the same time, the algorithm remains symmetric and rational and does not rely on regularization, making the results even more surprising.

Regarding the techniques used, they are also to the best of my knowledge novel. While two phase style techniques (global slow rates and local fast rates)  are pretty common even in the case of matrix games, these results are typically analyzing the performance of a single update rule that becomes faster closer to the equilibrium. Here the authors combine two distinct algorithms where one is responsible for the global convergence and the other for the fast rates. These two algorithms do not only have different rates, they are also qualitatively different as one relies on averaging and the other has last-iterate style guarantees. The instantiation of the local algorithm using OGDA may be of independent interest.

Regarding the minor weaknesses of this work, it is still unclear to me why an averaging style algorithm is required for the global step is unclear to me. Since fast convergence is not required, why cannot we employ last iterate results from prior work for this step. If the authors could explain the reasoning, it would be helpful.



**Summary Of The Paper:**

The authors propose a symmetric and rational algorithm for finding Nash policies for a two player zero sum Markov game that converges in linear time. Additionally the algorithm proposed does not rely on computing the equilibria of a regularized Markov game.

The key idea of this work is to combine a slow algorithm that enjoys global convergence to the Nash equilibrium and a fast algorithm whose linear convergence rate is only guaranteed close to the Nash equilibrium. The authors not only show how to instantiate the proposed components of their work but also how to combine them in a single unified algorithm. Importantly, this unification of the two components works despite the fact that the algorithm cannot detect if the iterates are in the linear convergence region of the fast algorithm.

**Summary Of The Review:**

In summary, based on the strong and novel results I am recommending acceptance of this work.

---

> ### Author Response · Authors · 2022-11-17
> **Response to Reviewer Uzv1 (Part 2/2)**
>
> - **Finally, even though we can use [R1] with modified initialization as Global-Slow to achieve linear convergence and decentralized implementation simultaneously, using Average OGDA as Global-Slow has its own merits – it has a faster convergence rate, which reduces the length of Hidden Phase I.** More specifically, as in Theorem 7 (Appendix H.1 of the revision), if the RHS of Eq(2) in the definition of Global-Slow is extended to $\widetilde{O}(1/(\eta'^{p_2}(T_2 - T_1 + 1)^{p_1}))$, then if $p_1$ is bigger or $p_2$ is smaller, the length of Hidden Phase I will be shorter.  The resulting convergence result of $\widehat{z}^{[T_1:T_2]}$ of [R1] with modified initialization is $O(1/(\eta'^2\sqrt{T_2 - T_1+1}))$. Meanwhile, Averaging OGDA has the faster $\widetilde{O}(1/(\eta'(T_2 - T_1+1))$ global convergence rate of of $\widehat{z}^{[T_1:T_2]}$. Faster global convergence rate of Averaging OGDA can reduce the length of Hidden Phase I. In other words, given the new finding in our revision that [R1] with modified initialization can have geometric boundedness, Averaging OGDA is still meaningful in the sense that it can help reduce the length of Hidden Phase I.
>
>
>
> **References**
>
> [R1] Wei, C. Y., Lee, C. W., Zhang, M., & Luo, H. (2021). Last-iterate convergence of decentralized optimistic gradient descent/ascent in infinite-horizon competitive markov games. In Conference on learning theory (pp. 4259-4299). PMLR.
>
> [R2] Cen, S., Wei, Y., & Chi, Y. (2021). Fast policy extragradient methods for competitive games with entropy regularization. Advances in Neural Information Processing Systems, 34, 27952-27964.
>
> [R3] Zeng, S., Doan, T. T., & Romberg, J. (2022). Regularized Gradient Descent Ascent for Two-Player Zero-Sum Markov Games. arXiv preprint arXiv:2205.13746.

---

> ### Author Response · Authors · 2022-11-17
> **Response to Reviewer Uzv1 (Part 1/2)**
>
> Thank you for your valuable feedback! Here are our responses.
>
> > Regarding the minor weaknesses of this work, it is still unclear to me why an averaging style algorithm is required for the global step is unclear to me. Since fast convergence is not required, why cannot we employ last iterate results from prior work for this step. If the authors could explain the reasoning, it would be helpful.
>
> - **For Global-Slow, we need (a) global convergence and (b) geometric boundedness which means that the iterates $\mathbf{z}^t$ and the average iterates do not diverge faster than geometric rates.** The geometric boundedness property is essential in our proof for the global linear convergence of Homotopy-PO. And to the best of our knowledge, there are no discussions about the geometric boundedness of the algorithms in prior works of MARL.
>
> - **Existing works with last-iterate global convergence can be divided into two cases:**
>   (1). they do not have geometric boundedness;  (2). although they might have geometric boundedness, their algorithms require the knowledge of the opponent’s policy (or its entropy), and thus cannot be implemented in a (fully) decentralized fashion. **Thus, using existing works as Global-Slow will make Homotopy-PO either fail to converge linearly or fail to be decentralized.** We take last-iterate convergent algorithms in [R1, R2, R3] as examples.
>
>   For case (1), the main algorithm in [R1] does not have geometric boundedness. Our proofs of geometric boundedness rely on the observation in Appendix B which intuitively shows that for any $z = \\{z_s\\}\_{s \in \mathcal{S}}$ if there hold $z_s \approx \mathcal{P}_{\mathcal{Z}^*_s}(z_s)$, $Q_s \approx Q_s^*$, then $z^+ = \\{z_s^+\\}\_{s \in \mathcal{S}}$  with $z_s^+ = (x_s - \eta Q_sy_s, y_s + \eta Q_s^{\top}x_s)$ will also be close to the NE set.
>
>   However, in [R1], the initialization therein is $V^0(s) = 0$ for any state $s$. In the first step, the difference between the estimate Q-function $Q_s[V^0]$ and the ideal Q-function  $Q_s^* = Q_s[v^*]$  could be large. This will induce an error term of order $O(||v^*||_{\infty}^2)$ (see Lemma 2 in Appendix B) which cannot be bounded by $dist^2(z^t, \mathcal{Z}^*)$. Thus, [R1] with $V^0(s) = 0$ does not have geometric boundedness.
>
>
>   For case (2), algorithms in [R2, R3] use entropy regularization. Additional information about the entropy of the opponent's policy is needed when running the algorithms in [R2, R3]. Thus, [R2, R3] are not (fully) decentralized algorithms.
>
>
> - **Moreover, it is possible to modify the last-iterate convergent, symmetric and decentralized algorithm in [R1] slightly to satisfy geometric boundedness. See Appendix H in the revised version.** In Appendix H.2, we change the initialization of the main algorithm in [R1] from $V^0(s) = 0$ to $V^0(s) = V^{x^1, y^1}(s)$, and then, prove the geometric boundedness of [R1] with the modified initialization. Since [R1] has last-iterate convergence, we can directly set $\widehat{z}^{[T_1:T_2]} = z^{T_2}$ when we are using the modified [R1] in Homotopy-PO. To analyze the corresponding convergence results when Homotopy-PO is instantiated by [R1] with the modified initialization and OGDA, we add Appendix H.1 in the revision. In Appendix H.1, we give the corresponding global linear rate of Homotopy-PO when the RHS of Eq(2) in the definition of Global-Slow is extended to the more general form $\widetilde{O}(1/(\eta'^{p_2}(T_2 - T_1 + 1)^{p_1}))$ which includes [R1] with modified initialization as a special case.

---

### Decision · Program_Chairs · 2023-01-20

**Decision:**

Accept: notable-top-25%

**Justification For Why Not Higher Score:**

Limitations about the observation structure.

**Justification For Why Not Lower Score:**

All reviewers found the idea in the paper interesting and original, and I also agree with them.

**Metareview: Summary, Strengths And Weaknesses:**

The paper studies decentralized learning in two-player zero-sum discounted Markov games, when the players observation are the marginal reward and transition functions. An algorithm is provided to find approximate Nash-equilibrium policies exponentially fast: both players independently play an algorithm that switches back and forth between a slower global optimization algorithm and a faster local optimization algorithm (which converges locally at a linear rate) according to a doubling-like schedule. It is shown that the global and local algorithms can be instantiated with averaging optimistic gradient descent-ascent and optimistic gradient descent-ascent, respectively. All reviewers found this algorithm original and interesting, and the result is the first algorithm to find a Nash equilibrium in this setting at a linear rate, while at the same time it is also decentralized, symmetric, and rational.

Weaknesses:
- The observation structure in the paper is somewhat unrealistic, which limits its usefulness. In the response the authors addressed this question by describing how one can obtain an actual learning algorithm with real observations -- their response should be incorporated in the final version (response to Reviewer p6su).
- In the response the authors present lots of intuition about OGDA and AOGDA, which should be incorporated in the main body of the paper (I hope an extra page will be available for this for the final version).
- During the rebuttal the authors provided interesting experiments, which demonstrate how the proposed algorithm work. It would be interesting to see how earlier algorithms in the literature behave in these example -- please include these additional experiments.

**Note From Pc:**

if the above contains the word "oral" or "spotlight" please see: "oral" presentation means -> notable-top-5% and "spotlight" means -> notable-top-25%. As stated in our emails, we are disassociating presentation type from AC recommendations